# Autoimmunity gene IRGM suppresses cGAS-STING and RIG-I-MAVS signaling to control interferon response

Kautilya Kumar Jena[1,2,†], Subhash Mehto[1,†], Parej Nath[1,2,†], Nishant Ranjan Chauhan[1], Rinku Sahu[1], Kollori Dhar[1], Saroj Kumar Das[3] (ID), Srinivasa Prasad Kolapalli[1], Krushna C Murmu[4] (ID), Ashish Jain[5,6], Sivaram Krishna[1], Bhabani Sankar Sahoo[7], Soma Chattopadhyay[8], Tor Erik Rusten[5,6] (ID), Punit Prasad[4], Swati Chauhan[7] & Santosh Chauhan[1,*] (ID)

## Abstract

Activation of the type 1 interferon response is extensively connected to the pathogenesis of autoimmune diseases. Loss of function of Immunity Related GTPase M (IRGM) has also been associated to several autoimmune diseases, but its mechanism of action is unknown. Here, we found that IRGM is a master negative regulator of the interferon response. Several nucleic acid-sensing pathways leading to interferon-stimulated gene expression are highly activated in IRGM knockout mice and human cells. Mechanistically, we show that IRGM interacts with nucleic acid sensor proteins, including cGAS and RIG-I, and mediates their p62-dependent autophagic degradation to restrain interferon signaling. Further, IRGM deficiency results in defective mitophagy leading to the accumulation of defunct leaky mitochondria that release cytosolic DAMPs and mtROS. Hence, IRGM deficiency increases not only the levels of the sensors, but also those of the stimuli that trigger the activation of the cGAS-STING and RIG-I-MAVS signaling axes, leading to robust induction of IFN responses. Taken together, this study defines the molecular mechanisms by which IRGM maintains interferon homeostasis and protects from autoimmune diseases.

Keywords autophagy; cGAS-STING; IRGM; mitophagy; RIG-I-MAVS
Subject Categories Autophagy & Cell Death; Immunology; Signal Transduction

## Introduction

Our understanding of the activation of innate immune systems upon exposure to hostile conditions such as microbial infection has grown exponentially (Akira *et al*, 2006; Takeuchi & Akira, 2010; Goubau *et al*, 2013). However, how the innate immune pathways are controlled under steady-state conditions are not well defined. In particular, the mechanisms by which negative regulators of innate immune system restrain the aberrant immune activations under basal conditions need to be characterized to understand the genesis of spontaneous inflammatory diseases, including autoimmune disorders. The type I interferon (IFN) response, constitutes a first line of defense against invading pathogens (esp. viruses), but chronic IFN activation can lead to several autoimmune diseases (Di Domizio & Cao, 2013; Psarras *et al*, 2017; Crow *et al*, 2019). A fine homeostatic balance of type I interferons needs to be maintained to avoid autoimmune diseases, including interferonopathies (Di Domizio & Cao, 2013; Niewold, 2014; Lee-Kirsch *et al*, 2016; Crow *et al*, 2019). The knowledge of the master switches and the mechanisms that suppress the type I IFN response will be beneficial for generating therapeutics against autoimmune diseases.

Pattern recognition receptors (PRRs) sense external (pathogen and PAMPs, pathogen-associated molecular patterns) and internal (DAMPs, danger-associated molecular patterns) cellular threats and mount a strong innate immune response that includes the production of pro-inflammatory cytokines (Takeuchi & Akira, 2010; Roers *et al*, 2016). The presence of PAMPs or DAMPs in the cytosol is sensed by cytosolic PRRs such as RIG-I like receptors (RLR) and

1  Cell Biology and Infectious Diseases Unit, Institute of Life Sciences, Bhubaneswar, India
2  School of Biotechnology, KIIT University, Bhubaneswar, India
3  Centre for Biotechnology, Siksha 'O' Anusandhan (Deemed to be University), Bhubaneswar, India
4  Epigenetic and Chromatin Biology Unit, Institute of Life Sciences, Bhubaneswar, India
5  Centre for Cancer Cell Reprogramming, Institute of Clinical Medicine, Faculty of Medicine, University of Oslo, Oslo, Norway
6  Department of Molecular Cell Biology, Institute for Cancer Research, Oslo University Hospital, Oslo, Norway
7  Institute of Life Sciences, Bhubaneswar, India
8  Molecular Virology Lab, Department of Infectious Disease Biology, Institute of Life Sciences, Bhubaneswar, India
   *Corresponding author. Tel: +0674 2304316; E-mails: schauhan@ils.res.in, chauhan2010santosh@gmail.com
   †These authors contributed equally to this work

NOD-like receptors (NLRs) and also by DNA and RNA sensors such cGAS, IFI16, and ZBP-1 (Unterholzner *et al*, 2010; Wu & Chen, 2014; Radoshevich & Dussurget, 2016; Roers *et al*, 2016; Kuriakose & Kanneganti, 2018). RIG-I or MDA5 senses dsRNA species and activates adaptor protein MAVS, which then acts as a platform for activation of TBK1 and IRF3/IRF7 transcription factors (Hornung *et al*, 2006; Kato *et al*, 2006; Reikine *et al*, 2014). These transcription factors then translocate to the nucleus to increase the production of type I interferons. Similarly, DNA sensor cGAS upon sensing dsDNA of viral, mitochondrial or genomic origin activates adaptor protein STING leading to activation of TBK1-IRF3/IRF7 axis for type I interferon production (Li *et al*, 2013; Sun *et al*, 2013; Rongvaux *et al*, 2014; West *et al*, 2015; Roers *et al*, 2016; Mackenzie *et al*, 2017). The interferons thus produced can activate the JAK-STAT1/2 signaling pathway leading to the production of interferon-stimulated genes (ISGs), which are the powerful effector proteins with a varied function in innate immunity, including antiviral/antibacterial response (Ivashkiv & Donlin, 2014; Roers *et al*, 2016). The imbalance in all of these signaling pathways has been strongly linked with autoimmunity (Di Domizio & Cao, 2013; Rice *et al*, 2014; Gray *et al*, 2015; Kato & Fujita, 2015; Louis *et al*, 2018)

IRGM (Irgm1) deficiency is genetically and functionally associated with several inflammatory and autoimmune diseases including ankylosing spondylitis, autoimmune thyroid diseases, Graves' disease, Sjogren's syndrome, Crohn's disease, experimental autoimmune encephalomyelitis, Hepatic steatosis, NAFLD (non-alcoholic fatty liver disease), and severe sepsis (Parkes *et al*, 2007; Xu *et al*, 2010; Kimura *et al*, 2014; Lin *et al*, 2016; Azzam *et al*, 2017; Bellini *et al*, 2017; Xia *et al*, 2017; Yao *et al*, 2018). Recently, in a knockout mouse model, Irgm1 (the mouse orthologue of IRGM) was shown to control autoimmunity (Azzam *et al*, 2017). The authors show that naive Irgm1 knockout mice, in germ-free conditions, displayed the hallmarks of Sjogren's syndrome, an autoimmune disorder characterized by lymphocytic infiltration of exocrine tissues (Azzam *et al*, 2017). The presence of IRGM/Irgm1 in humans and mice is shown to be largely protective against autoimmune disorders. The connections between IRGM and systemic autoimmune diseases argue a central role of IRGM in innate immune homeostasis. The molecular mechanism by which human IRGM controls innate immune homeostasis in steady-state conditions remains completely undetermined.

Three major types of autophagy, i.e., macroautophagy, microautophagy, and chaperone-mediated autophagy, contribute to cargo degradation in the cell. Macroautophagy (henceforth autophagy) utilizes double-membraned bound vesicles (autophagosomes), to engulf cellular cargoes (e.g., proteins and organelles) for delivery to the lysosome for degradation. Accumulating evidence suggest that autophagy-mediated clearance of defunct mitochondria is a powerful mechanism to keep the inflammation under-check (Tal *et al*, 2009; Oka *et al*, 2012; Sliter *et al*, 2018; Xu *et al*, 2020). Autophagy deficiency results in the accumulation of dysfunctional mitochondria that are the primary source of DAMPs for activation of cGAS-STING and RIG-I/MAVS signaling pathways. Activation of these pathways lead to robust induction of interferon response resulting in antiviral response or autoimmune diseases (Tal *et al*, 2009; Gkirtzimanaki *et al*, 2018; Sliter *et al*, 2018; Xu *et al*, 2020). We and others have found that IRGM is a key autophagy protein that plays a significant role in anti-bacterial autophagy and autophagy of inflammasomes (Singh *et al*, 2006, 2010; Chauhan *et al*, 2015;

Kumar *et al*, 2018; Mehto *et al*, 2019). IRGM was also shown to localize to mitochondria, and overexpression of IRGM induces mitochondrial fission, followed by its depolarization (Singh *et al*, 2010). However, it remains undetermined whether IRGM deficiency perturbs mitophagy and affects the downstream innate immune signaling pathways.

This study uncovers that under homeostatic conditions IRGM is a master suppressor of type I IFN response. Whole transcriptome analysis in human cells and mice shows that IRGM controls the expression of almost all major ISGs. Mechanistically, we show that IRGM suppresses IFN signaling by mediating p62-dependent autophagic degradation of cGAS, RIG-I, and TLR3. Further, we find that IRGM is critical for the removal of damaged mitochondria by macroautophagy. Thus, IRGM deficiency results in defective mitophagy, accumulation of dysfunctional mitochondria, and enhanced mitochondrial DAMPs that stimulate cGAS-STING and RIG-I-MAVS axis to drive robust activation of type I IFN response.

# Results

## IRGM is a master suppressor of the interferon response

To understand the role of IRGM in innate immune homeostasis and autoimmunity, we performed RNA sequencing (RNA-seq) experiments with (i) control and IRGM shRNA knockdown (hereafter, IRGM KD, Fig EV1A, Appendix Fig S1A) human HT29 colon epithelial cell line, (ii) wild-type ($Irgm1^{+/+}$) and *Irgm1* knockout mouse ($Irgm1^{-/-}$) bone marrow-derived macrophages (BMDMs), and (iii) $Irgm1^{+/+}$ and $Irgm1^{-/-}$ brain tissues.

The gene ontology (GO)-based pathway analysis was performed using Ingenuity pathway analysis (IPA, https://analysis.ingenuity.com/), Reactome pathway analysis (Fabregat *et al*, 2018), and Metascape pathway analysis (Tripathi *et al*, 2015) with genes upregulated (1.5-fold, $P < 0.05$, $n = 3$) in IRGM KD HT29 cells. In all of these analyses, the top-enriched pathways were the induction of innate/adaptive immune systems and inflammatory signaling/responses (Figs 1A and EV1B, Appendix Fig S1B), indicating that the primary function of human IRGM under steady-state conditions is to control the cellular inflammation and immunity. A closer look at the genes and the pathways that are upregulated suggest that IRGM deficiency results in the induction of interferon responses or the processes/pathways controlled by the interferon responses (Figs 1A and EV1B, Appendix Fig S1B, Dataset EV1). To our surprise, almost all well-known ISGs including interferon-inducible (IFI) genes, oligoadenylate synthases (OAS) genes, ISG15/20, guanylate-binding proteins (GBPs), apolipoprotein B mRNA-editing catalytic polypeptide-like genes (APOBEC), myxovirus resistance (MX genes), MHC class 1 antigen processing and presentation genes, and tripartite motif (TRIM) genes were upregulated upon knocking down IRGM (Fig 1A, and Dataset EV1). The Interferome database analysis (Rusinova *et al*, 2013) using highly stringent parameters shows that ~ 45% of the genes (392 out of a total of 890) induced in IRGM KD cells are interferon-regulated (Fig 1B). The interferons are the major defense system against viruses and that is the reason why the "defense response to viruses/microbes" are other top-induced functions in the IPA (Fig 1C, Dataset EV2) and Metascape pathway analysis (Appendix Fig S1B). In IPA, cancer, autoimmunity (Psoriasis,

Sjogren's syndrome, age-related macular degeneration) and other inflammatory disorders were the top diseases associated with IRGM deficiency (Appendix Fig S1C, and Dataset EV2). The qRT–PCR was performed with key interferon-inducible genes (*RIG-I, IFI16, MDA5, STAT2, OAS1, MX2, ISG15, TRIM22, and APOBEC3G*) to validate the RNA-seq data (Fig 1D). We observed 2- to 1200-folds induction of ISGs in IRGM-deficient cells suggesting that IRGM is a potent inhibitor of interferon response (Fig 1D). The RNA-seq data were also validated in human THP-1 monocytic cells, and also the expression of few of the ISGs was validated in human peripheral blood mononuclear cells (PBMCs) from three independent human donors, where IRGM was knockdown using siRNA (Fig EV1C).

Next, we performed pathway analysis with RNA-seq data from brain and BMDMs of $Irgm1^{+/+}$ and $Irgm1^{-/-}$ mice ($n = 3$). The reason for performing RNA-seq with brain tissues is that it is a relatively immune-privileged organ and is mostly insusceptible to perturbation in the peripheral immune system due to extraneous irritants, and thus, immune responses are typically cell-intrinsic. There was a remarkable similarity in the upregulated genes and pathways in IRGM-depleted human HT29 cells, the $Irgm1^{-/-}$ mouse brain, and the $Irgm1^{-/-}$ BMDMs (Figs 1E and EV1D, and Dataset EV1). Both in the brain and BMDMs, the pathways that were enriched as a response of $Irgm1$ knockout were related to cytokine response, interferon signaling/response, and antiviral/microbial response (Figs 1E and EV1D and E, Appendix Fig S1D, and E). Remarkably, > 80% of the genes (225 out of 288) that were upregulated in $Irgm1^{-/-}$ brain and > 50% of the genes (314 out of 595) that were induced in $Irgm1^{-/-}$ BMDMs were ISGs (Figs 1F and EV1F). Because of systemic induction of ISGs, again the top functions and diseases associated with $Irgm1$ deficiency in brain and BMDMs were the antiviral response, systemic autoimmune syndrome (systemic lupus erythematosus, Sjogren's syndrome, psoriasis), rheumatic diseases, and other inflammatory disorders (Fig 1G, Appendix Fig S1F, and Dataset EV2). The RNA-seq data validation with qRT–PCR from brain tissues showed robust induction of key ISGs (*Rig-I, Mda5, Ifi16, Irf7, Ifn-β, Stat1, Stat2, Isg15, Apobec3, Oas1, Mx1, and Mx2*) in $Irgm1^{-/-}$ mice (Fig 1H).

The class-I MHC restricted antigen presentation pathway is vital for processing and presentation of microbial (endogenous) and tumor antigens leading to antiviral/bacterial and anti-tumor response (Pamer & Cresswell, 1998; Cresswell *et al*, 2005). The expression of class 1 MHC genes is controlled by the interferon response (Keskinen *et al*, 1997; Zhou, 2009; Coomans de Brachene *et al*, 2018). Several of the genes integral to class-I MHC-mediated antigen processing and presentation pathways required for folding, assembly, and peptide loading (HLA genes, immune-proteasome genes, B2M, and TAP1/2; Fig EV1G) were upregulated in IRGM-depleted human and mouse cells (Fig EV1H). Several of the complement pathway genes were upregulated in IRGM-depleted cells (Fig EV1I, Dataset EV1). Both these pathways are known for their role in antimicrobial defense and pathogenesis of systemic autoimmune diseases (Mitchell *et al*, 1996; Byun *et al*, 2007; Chen *et al*, 2010, 2014; Silk *et al*, 2017). Several of the interferon-inducible TRIM proteins (TRIM5, 6, 12, 14, 20, 22 25, 29, 30, 34) that are known to play a significant role in innate immunity including inflammation and virus restriction (Ozato *et al*, 2008; van Gent *et al*, 2018) were significantly upregulated upon depleting IRGM (Fig EV1J, Dataset EV1). Similarly, GBPs that are the critical

effectors of the immune system against pathogens and are interferon responsive genes (Praefcke, 2018) were induced in IRGM-depleted cells (GBP1 and 4 in HT29 cell and GBP2–10 in mice; Dataset EV1). The GO-based pathway analysis with the genes that were downregulated in IRGM-depleted human or mouse cells showed no immunity or inflammatory-related pathways suggesting that IRGM is a very specific suppressor of the inflammatory responses (Appendix Fig S1G).

Taken together, the transcriptome analyses in human cells and mice suggest that (i) the IRGM-mediated regulation of immune systems and interferon response is systemic (not organ-specific), (ii) the mouse Irgm1 (a 42 kDa protein) and human IRGM (a 21 kDa protein), although biochemically different, functionally are highly similar in the regulation of inflammation especially in the regulation of interferons responses, (iii) IRGM is a master switch that suppresses the interferon responses under steady-state conditions and its deficiency results in robust and systemic induction of type 1 IFN response.

## Constitutively activated nucleic acid-sensing signaling pathways in IRGM-depleted cells and mice

The mRNA expression of several cytoplasmic PRRs, including *RIG-I, MDA5,* and *TLR3,* was significantly increased in *IRGM*-depleted mice and human cells (Fig 2A). These PRRs sense cytoplasmic DNA or dsRNA of self or pathogen origin and induces signaling events leading to the production of type I IFNs, which then activate JAK-STAT signaling pathway for the production of ISGs (Fig EV2A). Through transcriptome analysis, it was explicit that the IRGM controls interferon response. However, what are the possible signaling pathways that induce the ISGs in IRGM-depleted cells were not evident. To understand this, we examined the expressions of proteins of DNA/RNA sensing and signaling pathways leading to ISG activation (Fig EV2A) in human IRGM knockdown and mouse IRGM knockout cells using western blotting.

Even after several attempts, we were not able to generate and/or maintain complete CRISPR/Cas9 knockout of IRGM in THP1 or HT29 cells. The transfected cells were dying after a few days in culture. This could be due to increased inflammation leading to cell death in these stable cells (Mehto *et al*, 2019). However, knockout of a single allele of IRGM was well tolerated in HT29 cells (Clone#7, henceforth IRGM$^{+/-}$, Fig EV2B), which is used in several experiments in this study.

We observed increased protein expression of DNA and RNA sensor proteins RIG-I, TLR3, MDA5, and cGAS in IRGM knockdown cells and $Irgm1^{-/-}$ mice (Fig 2B–D). The TLR4 amount remained unchanged (Fig 2C and D). The cGAS was not induced at mRNA level in RNA-seq data, but at protein levels, an evident increase was observed. The adaptor proteins STING, MAVS, and TRIF transduce the signals from cGAS, RIG-I, and TLR3, respectively, leading to activation of TBK1 (Fig EV2A). The total amounts of STING and TRIF were higher in IRGM-depleted cells; however, MAVS levels were unchanged (Fig 2E). Although the total amount of MAVS was not increased, the MAVS aggregation, which is a hallmark of MAVS activation (Hou *et al*, 2011), was markedly induced in the absence of IRGM in Semi-Denaturing Detergent Agarose Gel Electrophoresis (SDD-AGE) assays (Fig 2F). To ascertain, we also performed immunofluorescence assays with IRGM-depleted cells. The results

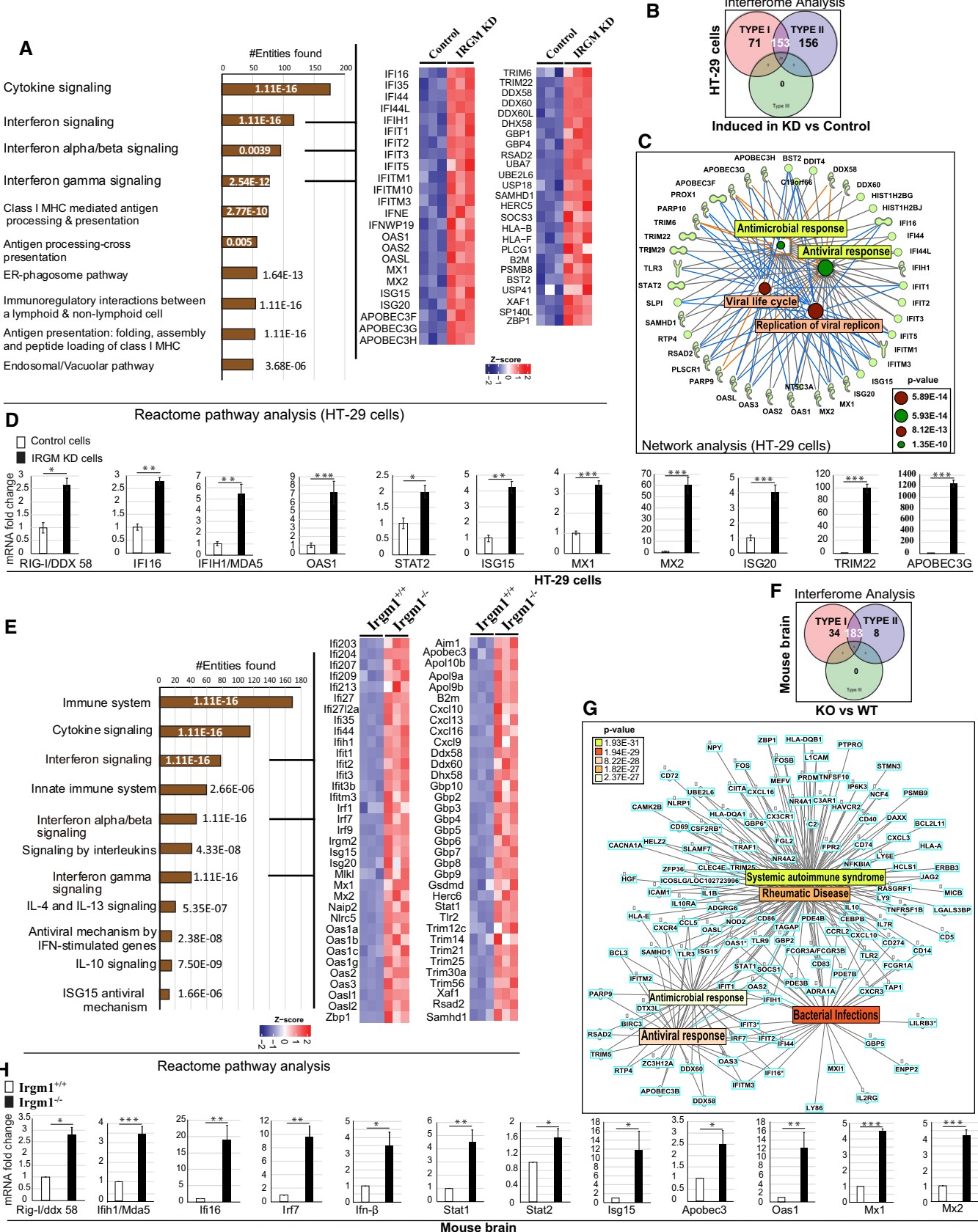

Figure 1.

**Figure 1. IRGM is a master negative regulator of the interferon response.**

A  The bar graph represents top 10 biological pathways upregulated in gene ontology (GO)-based Reactome pathway analysis using a set of genes induced (1.5-fold, $P < 0.05$ Wald test, three biological replicates) in RNA-seq analysis in IRGM shRNA knockdown HT29 cells compared to control shRNA cells. Heatmaps were generated for sentinel interferon-regulated genes (three biological replicates) using "ComplexHeatmap" library using "R" Bioconductor package where the gene expression matrix was transformed into z-score. The heatmap was generated from the common genes present in the three GO terms indicated by the three black lines. The numbers on the bars indicate the $P$-value of that particular GO term.

B  Interferome database analysis with a set of genes induced (1.5-fold, $P < 0.05$ Wald test, three biological replicates) in IRGM shRNA knockdown HT29 cells compared to control shRNA cells. The venn diagram depicts the total number of upregulated type I and type II IFN-regulated genes in IRGM KD cells.

C  Network pathway analysis using IPA. The molecular network of genes connected with the top 4 function-associated genes (1.5-fold, $P < 0.05$ Wald test, three biological replicates) upregulated in IRGM knockdown HT29 cells. The complete list is documented in Dataset EV2.

D  The qRT–PCR validation of RNA-seq data in control and IRGM KD HT29 cells. Mean $\pm$ SD, $n = 3$ (biological replicates), $*P < 0.05$, $**P < 0.005$, $***P < 0.0005$, Student's unpaired $t$-test.

E  The bar graph represents the top pathways upregulated in GO-based Reactome pathway analysis using set of genes induced (1.5-fold, $P < 0.05$ Wald test, three mice each group) in the brain of $Irgm1^{-/-}$ mice compared to $Irgm1^{+/+}$ wild-type mice. Heatmaps were generated for sentinel interferon-regulated genes (three biological replicates) using "ComplexHeatmap" library using "R" Bioconductor package where the gene expression matrix was transformed into z-score. The heatmap was generated from the common genes present in the three GO terms indicated by the three black lines. The numbers on the bars indicate the $P$-value of that particular GO term.

F  Interferome database analysis with a set of genes induced (1.5-fold, $P < 0.05$, 3 biological replicates) in the brain of $Irgm1^{-/-}$ mice compared to $Irgm1^{+/+}$ wild-type mice. The venn diagram depicts the total number of upregulated type I and type 2 IFN-regulated genes in $Irgm1^{-/-}$ mouse brain.

G  Network pathway analysis using IPA. The molecular network of genes connected with the top five functions/diseases associated with genes (1.5-fold, $P < 0.05$ Wald test, three biological replicates) upregulated in $Irgm1$ knockout mouse brain. The complete list is documented in Dataset EV2.

H  The qRT–PCR validation of RNA-seq data in $Irgm1^{+/+}$ and $Irgm1^{-/-}$ mouse brain. Mean $\pm$ SD, $n = 3$ (biological replicates), $*P < 0.05$, $**P < 0.005$, $***P < 0.0005$, Student's unpaired $t$-test.

clearly showed increased aggregation of MAVS in IRGM-depleted mouse BMDMs and THP-1 cells (Fig EV2C and D). Also, these aggregates were co-localized over the mitochondria (Fig EV2C). These data indicate that MAVS is activated in IRGM-depleted cells and mice.

TBK 1 plays a central role in interferon response and serves as an integrator of multiple signals induced by nucleic acid sensors signaling cascades (cGAS, RIG-I, TLR3, and MDA5) leading to the activation of IRF3, and IRF7 transcription factors (Fig EV2A) TBK1 is activated by autophosphorylation at residue Ser172 (Shu et al, 2013). We observed increased phosphorylation of TBK1 in IRGM-deficient human HT29 cells and BMDMs of $Irgm1^{-/-}$ mice (Fig 2G–I). Although the total amount of TBK1 was unchanged in HT29, there was an increased expression of Tbk1 in BMDMs of $Irgm1^{-/-}$ mice compared to controls. Activated TBK1 can increase phosphorylation of IRF3 and IRF7. Consistent with TBK1 activation in IRGM-deficient cells, the activating phosphorylation (Ser396) of IRF3 and IRF7 (Ser477) was increased in IRGM/Irgm1 knockdown/knockout cells (Fig 2J–L). However, the total amount of IRF3 and IRF7 remain unchanged.

When phosphorylated, IRF7 and IRF3 form homodimers or heterodimers, which then translocate to the nucleus and induce the expression of IFN genes (Fig EV2A). The IFNs in autocrine or paracrine manner through the IFN receptors activate the JAK-STAT1/2 pathway (Fig EV2A). Next, we analyzed the status of the JAK-STAT signaling pathway by measuring the phosphorylation status of STAT1 and STAT2, the key events for the activation of this pathway. The total amount, as well as activating phosphorylation of STAT1 (Tyr701) and STAT2 (Tyr690), was substantially increased in IRGM knockout mice and knockdown cells (Figs 2M and N, and EV2E). The expression of IRF9 remained unchanged (Fig 2M and N). As a final step, we determined the protein expression of a few of the ISGs. Heightened levels of MX1, OAS1, and Isg15 proteins were observed in IRGM-deficient cells (Fig 2O–Q). Altogether, the transcriptomic data followed by western blot analysis show that IRGM deficiency results in constitutive expression of several nucleic acid sensor proteins and activation of downstream interferon signaling

pathways leading to increased JAK-STAT1/2 signaling resulting in enduring production of ISGs.

## IRGM interacts with and degrades nucleic acid sensor proteins to control the aberrant activation of the interferon response

Immunity Related GTPase M is an autophagy protein, and its deficiency leads to diminished autophagy flux in immune cells (Singh et al, 2006; Chauhan et al, 2015; Dong et al, 2015; Hansen et al, 2017; Kumar et al, 2018; Mehto et al, 2019). IRGM is known to interact and degrade several PRRs, including NOD1, NOD2, and NLRP3 (Chauhan et al, 2015; Mehto et al, 2019). PRRs are threat sensor proteins, which activate pathways for cytokines responses. Hence, under steady-state conditions, maintaining their low expression is key to preserve the anti-inflammatory state of the cells. The total amount of several of the nucleic acid sensor proteins is controlled by autophagy-mediated degradation (Chen et al, 2016; Liu et al, 2016; Xian et al, 2020). We hypothesized that IRGM, by mounting selective autophagy of nucleic acid sensor proteins restricts the type 1 IFN response under basal conditions. To test this hypothesis, first, we examined whether IRGM interacts with the nucleic acid sensor proteins. In immunoprecipitation assays, a clear interaction between endogenous IRGM and endogenous RIG-I, cGAS, and TLR3 was observed (Fig 3A). However, no interaction was observed with MDA5, AIM2, TLR7, TLR4, and NLRC4 (Fig 3A), suggesting that IRGM specifically interacts with cGAS, RIG-I, and TLR3. We validated these interactions by performing co-immunoprecipitation assays (co-IPs) in HEK293T cells with overexpressed proteins. GFP-IRGM clearly interacted with Flag-tagged cGAS, RIG-I, and TLR3 but not with AIM2 (Fig 3B–D, Appendix Fig S2A). The reverse co-IPs were also performed to further validate these interactions (Fig EV3A). Next, we performed GST pulldown assays to scrutinize IRGM direct interactions with cGAS, RIG-I, and TLR3 using purified GST-IRGM and in vitro translated PRRs. A direct interaction was observed between IRGM and all the three PRRs (Fig 3E). The GST-IRGM strongly interacted with cGAS, but relatively weaker interaction was observed between

IRGM and RIG-I or TLR3 (Fig 3E). Taken together, the results show that IRGM directly interacts with cGAS, RIG-I, and TLR3.

Next, we examined whether IRGM can mediate the degradation of the cytoplasmic sensor proteins to which it interacts. The endogenous levels of RIG-I, cGAS, and TLR3 (but not of MAVS) were reduced by stable (HT-29 cells) or transient (THP1 cells) overexpression (for 4 h) of Flag-IRGM (Figs 3F and EV3B). Previously, we have shown that IRGM does not affect the expression levels of NLRC4 and AIM2 (Mehto et al, 2019). Since cGAS, RIG-I, and TLR3 expression is controlled by IFN response, the reduction of

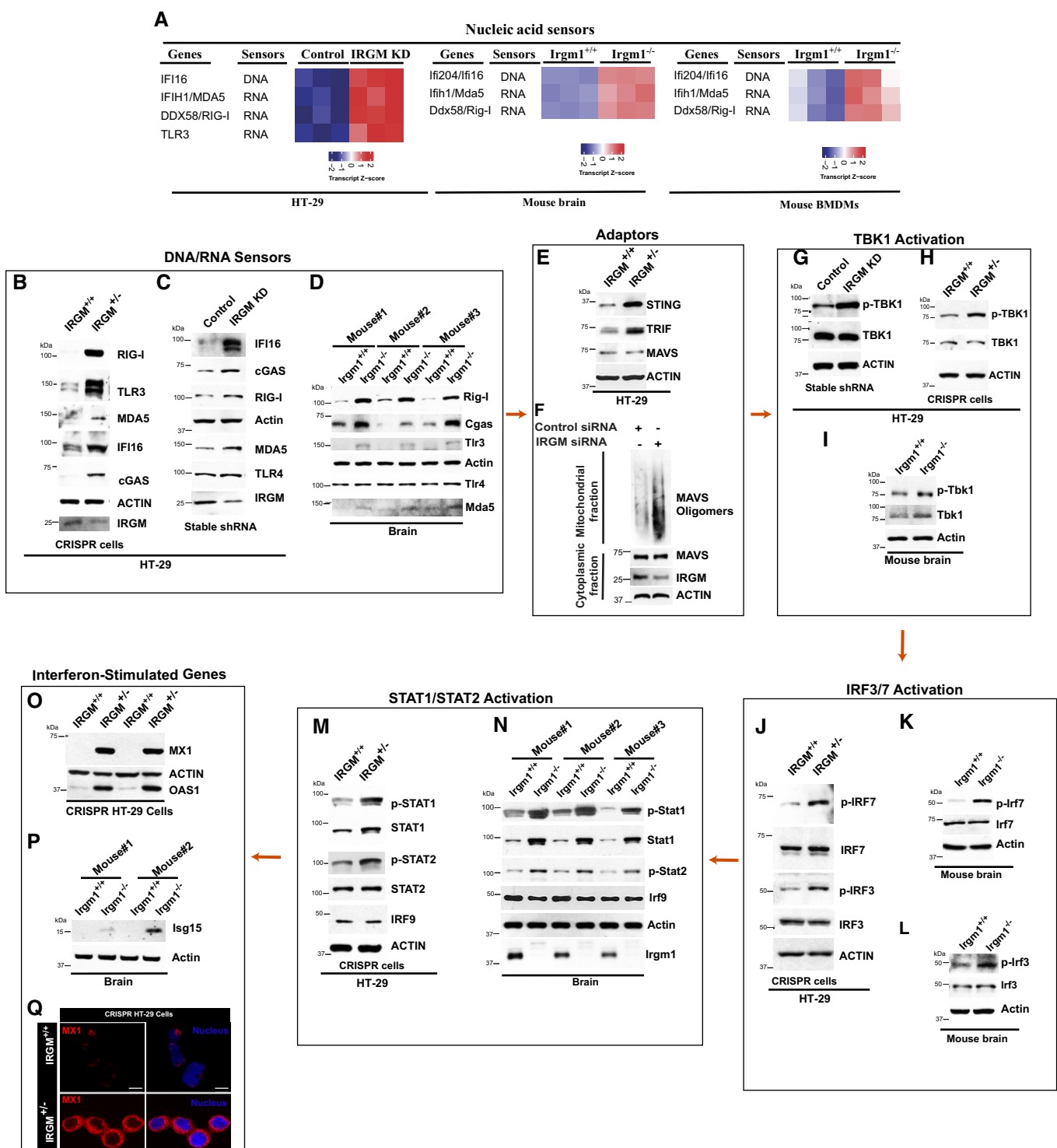

**Figure 2.**

**Figure 2.  The nucleic acid-sensing and ISG production pathways are constitutively active in IRGM-depleted cells and mice.**

A       Heatmap of nucleic acid sensor proteins upregulated in IRGM KD HT29 cells and *Irgm1*$^{-/-}$ mouse brain and BMDMs.

B–D     Western blot analysis to determine levels of nucleic acid sensor proteins with lysates of (B) HT29 control (henceforth IRGM$^{+/+}$) and single allele CRISPR knockout
        IRGM cells (henceforth IRGM$^{+/-}$), (C) HT29 cells stably expressing control shRNA or IRGM shRNA, (D) *Irgm1*$^{+/+}$ and *Irgm1*$^{-/-}$ mouse brain (n = 3 mice).

E       Western blot analysis to determine levels of adaptor proteins in control and IRGM$^{+/-}$ HT29 cells.

F       SDD-AGE followed by western blot analysis with a mitochondrial fraction from control and IRGM siRNA knockdown THP1 cells. Western blot analysis with
        cytoplasmic fraction was also performed.

G–I     Western blot analysis performed with lysates of (G) control and stable IRGM shRNA knockdown HT29 cells (H) IRGM$^{+/+}$ and IRGM$^{+/-}$ HT29 cells, and (I) *Irgm1*$^{+/+}$
        and *Irgm1*$^{-/-}$ mouse brain to determine levels of TBK1 protein.

J–L     Western blot analysis performed with lysates of (J) IRGM$^{+/+}$ and IRGM$^{+/-}$ HT29 cells (K, L) *Irgm1*$^{+/+}$ and *Irgm1*$^{-/-}$ mouse brain to determine levels of IRF proteins.

M, N    Western blot analysis performed with lysates of (M) IRGM$^{+/+}$ and IRGM$^{+/-}$ HT29 cells, (N) *Irgm1*$^{+/+}$ and *Irgm1*$^{-/-}$ mouse brain to determine levels of STAT proteins
        (n = 3 mice).

O, P    Western blot analysis performed with lysates of (O) IRGM$^{+/+}$ and IRGM$^{+/-}$ HT29 cells. 2 biological replicates are shown. (P) *Irgm1*$^{+/+}$ and *Irgm1*$^{-/-}$ mouse brain to
        determine levels of ISG proteins (n = 2 mice).

Q       Representative confocal image of immunofluorescence assay performed with IRGM$^{+/+}$ and IRGM$^{+/-}$ HT29 cells stained with MX1 antibodies. Scale bar, 10 μm.

Source data are available online for this figure.

endogenous levels of these proteins could be an indirect effect of IRGM-mediated suppression of IFN response. To rule out this possibility, we overexpressed both IRGM and the sensor proteins using CMV promoter-driven ORFs in HEK293T cells. The results clearly show that IRGM overexpression can reduce the total amount of RIG-I, cGAS, and TLR3 (Fig 3G–I) but not of AIM2, MAVS, and TLR4 (Appendix Fig S2B and C), suggesting that IRGM is directly involved in the degradation of RIG-I, cGAS, and TLR3 sensor proteins. This phenomenon is further validated by the overexpression of Flag-tagged IRGM (Fig EV3C–E). To further rule out the possibility that the reduction of the endogenous protein levels of these sensors was due to their reduced transcription levels, we blocked the transcription in cells using actinomycin D and chase the Flag-RIG-I protein degradation in the absence and presence of GFP-IRGM. The results show faster protein degradation in the presence of GFP-IRGM in comparison to GFP controls (Appendix Fig S2D), suggesting that indeed IRGM mediates degradation of sensor proteins.

Consistent with these results, the overexpression of IRGM suppressed the levels of the sentinel ISG genes, including *MX2,* and *ISG15* (Fig 3J and K). In agreement with RNA-seq data, the RIG-I and TLR3 (being ISGs) were also suppressed upon IRGM overexpression, but there was no change in mRNA levels of cGAS (Appendix Fig S2E). Furthermore, in ISRE (interferon-stimulated response element) luciferase reporter assays, the overexpression of IRGM reduced the RIG-I, cGAS/STING, and TLR3 induced ISRE-driven promoter transcription (Fig 3L–N). Overall, the data suggest that IRGM interacts and degrades RIG-I, cGAS, and TLR3 to keep type I IFN response under-check.

## IRGM mediates p62- and Beclin1-dependent autophagic degradation of nucleic acid sensors to restrain the activation of the interferon response

Using autophagy and proteasome inhibitors, we next determined the process utilized by IRGM to degrade these proteins. IRGM-mediated degradation of endogenous RIG-I, cGAS, and TLR3 were abrogated by autophagy/lysosomal inhibitors; bafilomycin A1 (BafA1) and chloroquine (Fig 3O and P). The proteasomal inhibitor, MG132, was not able to block the IRGM-mediated degradation of endogenous RIG-I and TLR3, whereas it diminished the degradation of cGAS (Appendix Fig S2F). Similarly, in overexpression

experiments, the GFP-IRGM-mediated degradation of Flag-RIG-I and Flag-TLR3 was abrogated by autophagy/lysosome inhibitors but not by MG132 (Fig EV3F–H), whereas cGAS degradation was reduced by both BafA1 and MG132 (Fig EV3F). These data suggest that IRGM majorly invokes lysosomal degradation of the RIG-I and TLR3. However, cGAS expression is controlled by both lysosomal and proteasomal degradation. If autophagy is the key process employed by IRGM to degrade the nucleic acid sensors, then IRGM-mediated suppression of ISGs should be rescued by lysosomal inhibitors. Indeed, inhibition of autophagy or lysosomal degradation by BafA1 significantly de-repressed the IRGM-mediated inhibition of expression of ISGs (Fig 3Q).

Immunity Related GTPase M facilitates the autophagic degradation of proteins. However itself, it is not degraded by autophagy (Kumar *et al*, 2018; Mehto *et al*, 2019). This is not surprising as several of the core autophagy proteins such as ULK1, ATG16L1, ATG12 (Haller *et al*, 2014; Nazio *et al*, 2016; Scrivo *et al*, 2019) and endolysosomal trafficking proteins such as RAB7A (Mohapatra *et al*, 2019), which facilitates autophagic degradation of cargo proteins but themselves are not degraded by the autophagy.

Above, we used chemical inhibitors to show that IRGM induces autophagy to degrade sensor proteins and control type 1 IFN response. Next, we used a genetic method to validate this finding. BECLIN1 and ATG5 are essential autophagy proteins. The siRNA-mediated knockdown of BECLIN1 and ATG5 in IRGM overexpressing cells abolished the IRGM-mediated degradation of nucleic acid sensor proteins (Fig 4A and B). Further, siRNA-mediated depletion of BECLIN1 and ATG5 in IRGM overexpressing cells considerably restored the expression of ISGs (Fig 4C–H) suggesting that IRGM mounts canonical BECLIN1 and ATG5-dependent autophagy as a key mechanism to maintain low levels of nucleic acid sensor proteins and keeping the type I IFN response under-check. In agreement with this conclusion, the knockdown of BECLIN1 partly phenocopies the depletion of IRGM in terms of induction of IFN response (Fig EV3H, Appendix Fig S2G).

The autophagy adaptor proteins recognize cargoes for selective autophagic degradation (Svenning & Johansen, 2013; Kim *et al*, 2016). We screened the interaction between IRGM and several established autophagy adaptor proteins, including optineurin, TAX1BP1, p62, NDP52, and NBR1, to identify the adaptor protein/s utilized by IRGM to mediate selective autophagic degradation. IRGM

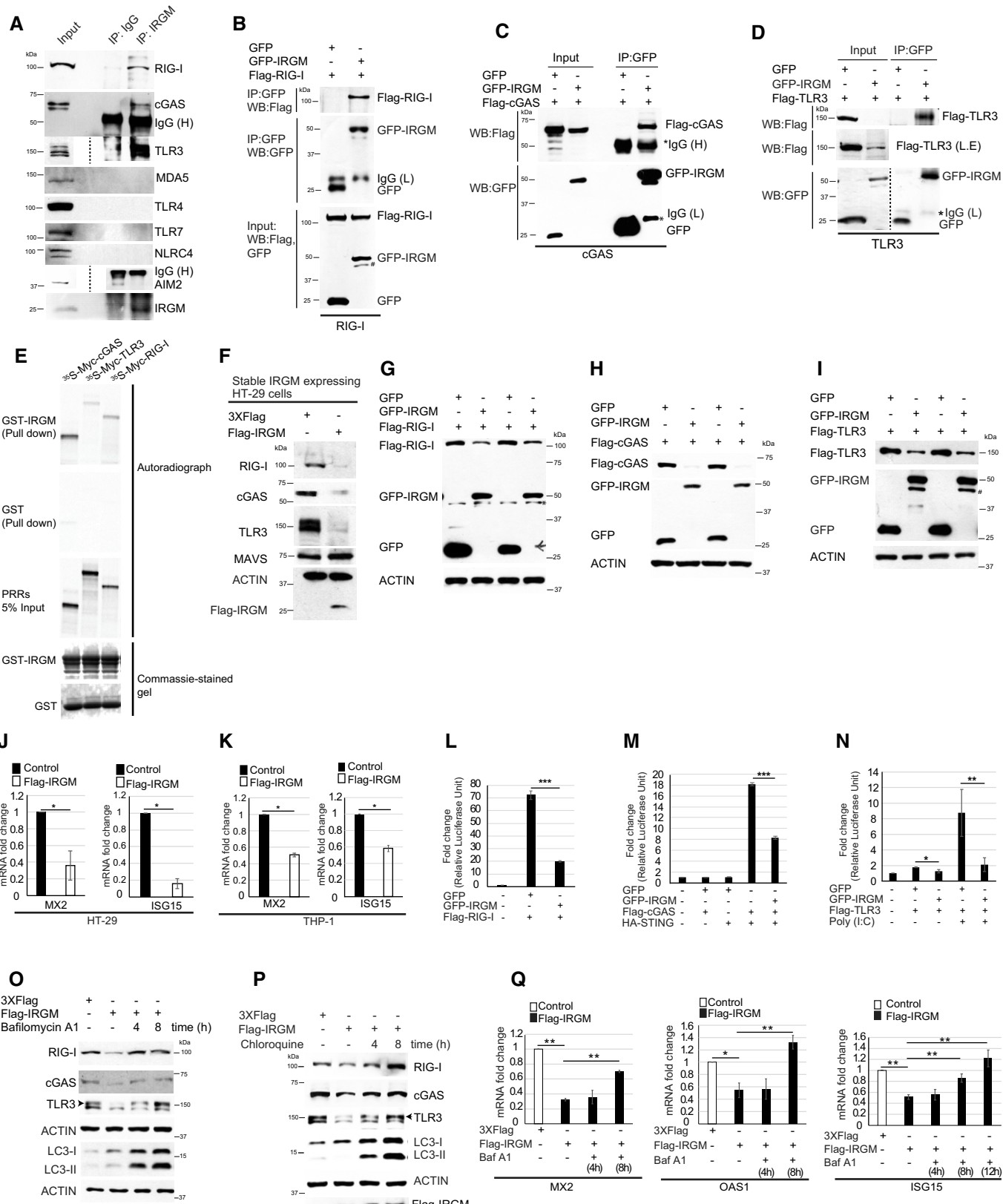

Figure 3.

Figure 3. IRGM interacts with DNA/RNA sensors and utilizes autophagy to degrade them so as to constrain IFN response.

A       Immunoprecipitation (IP) analysis of the interaction between endogenous IRGM and endogenous RIG-I, cGAS, MDA5, TLR3, TLR4, TLR7, NLRC4, and AIM2 in THP-1 cell lysates. IgG (H), IgG heavy chain.

B–D    Co-immunoprecipitation (Co-IP) analysis of the interaction between (B) GFP-IRGM and Flag-RIG-I, (C) GFP-IRGM, and Flag-cGAS, (D) GFP-IRGM and Flag-TLR3 in HEK293T cell lysates. L.E, Long exposure of input blot. IgG (L), IgG Light chain; IgG (H), IgG heavy chain.

E       GST pulldown assay of *in vitro* translated and radiolabelled myc-tagged cGAS, RIG-I, and TLR3 with GST or GST-tagged IRGM.

F       Western blot analysis with the lysates of HT-29 cells stably transfected with control or Flag-IRGM plasmid and probed with the indicated antibodies.

G–I    Western blot analysis with lysates of HEK293T cells transiently expressing GFP or GFP-IRGM and (G) Flag-RIG-I or (H) Flag-cGAS or (I) or Flag-TLR3. Two biological replicates are shown.

J–K    The qRT–PCR analysis with RNA isolated from (J) control and Flag-IRGM stable HT-29 cells (K) THP-1 cells transiently transfected with control or Flag-IRGM plasmid. *n* = 3 (biological replicates). Mean ± SD, *P < 0.05, Student's unpaired *t*-test.

L–N    Luciferase reporter assays in HEK293T cells transiently transfected with ISRE reporter plasmid and other plasmids as indicated. Mean ± SD, *n* = 3 (biological repeats), *P < 0.05, **P < 0.005, ***P < 0.0005, Student's unpaired *t*-test.

O       Western blotting analysis with lysates of control and untreated or bafilomycin A1 (300 nM; 4 and 8 h) treated Flag-IRGM expressing HT29 stable cell lines.

P       Western blotting analysis with lysates of control and untreated or chloroquine (50 μM; 4 and 8 h) treated Flag-IRGM expressing HT29 stable cell lines.

Q       The qRT–PCR analysis with RNA isolated from untreated or bafilomycin A1 (300 nM; 4, 8, and 12 h) treated control or Flag-IRGM expressing HT-29 stable cells as indicated. Mean ± SD, *n* = 3 (biological replicates), *P < 0.05, **P < 0.005 Student's unpaired *t*-test.

Data information: *Non-specific, #degradation products of full-length GFP-IRGM.
Source data are available online for this figure.

interacted strongly with p62, mildly with TAX1BP1, and no interaction was observed with optineurin, NDP52, and NBR1 (Fig 4I and J). Several of the key PRRs (including cGAS and RIG-I) and immune proteins (including STING and TRIF) are shown to be degraded by p62-mediated autophagy (Chen *et al*, 2016; Liu *et al*, 2016; Prabakaran *et al*, 2018; Samie *et al*, 2018; Xian *et al*, 2020). From our IRGM-p62 interaction data and the literature, we envisaged that p62 could be the key adaptor protein employed by IRGM for selective autophagic degradation of nucleic acid sensor proteins. Indeed, the depletion of p62 dramatically reduced the interaction of IRGM with cGAS, RIG-I, and TLR3 (Fig 4K–M, Appendix Fig S2H). Further, the knockdown of p62 abrogated the IRGM-mediated degradation of RIG-I, cGAS, and TLR3 (Fig 4N). Furthermore, p62 knockdown in IRGM overexpressing cells rescued the IRGM-mediated downregulation of ISGs in HT29 and THP-1 cells (Fig 4G, H and O). None of the other adaptor proteins (TAX1BP1, NBR1, or NDP52) was able to rescue the IRGM-mediated suppression of ISGs (Appendix Fig S2I and J). Interestingly, the knockdown of p62 partly phenocopies the depletion of IRGM in terms of induction of IFN response (Fig EV3J, Appendix Fig S2K). Taken together, the results suggest that p62 is the adaptor protein utilized by IRGM for the degradation of PRRs.

Previously we found that IRGM can act as a scaffold protein for increasing interaction between NOD2/NOD1 and autophagy proteins (Chauhan *et al*, 2015). We tested here whether IRGM can potentiate interaction between cGAS/RIG-I/TLR3 and p62 in co-immunoprecipitation assays. We found that p62 immunoprecipitates both IRGM and PRRs indicating a ternary complex (at least) of p62-IRGM-PRRs is formed inside the cells (Fig 4P–R). And indeed, the interaction between p62 and PRRs is increased in the presence of IRGM (Fig 4P–R). Further, we know that the p62 interacts with ubiquitinated (especially K63-linked) cargoes to deliver them to autophagosomes. In order to understand how IRGM might increase the interaction between p62 and PRRs, we tested whether the presence of IRGM increases the K63-linked ubiquitination of PRRs. Indeed, the presence of IRGM increased the K63-linked ubiquitination of PRRs (Fig EV3K). Since overexpression of IRGM degrades the PRRs (Appendix Fig S2L), the ratio of immunoprecipitation samples was run in the ratio of adjusted inputs (Fig EV3K).

Taken together, the data indicate that IRGM supports the ubiquitination of PRRs leading to their p62-mediated selective autophagic degradation resulting in a controlled type I IFN response under basal conditions.

## IRGM deficiency results in defective mitophagy flux and increased mtROS

Immunity Related GTPase M deficiency results in increased expression of nucleic acid sensor proteins and activation of interferon signaling pathways (Figs 1 and 2). However, it was not clear to us, what keeps the sensors in ON state, and fuels the persistent activation of these pathways. Next, we set to determine the cell-intrinsic stimuli (DAMPs) that feeds the enduring IFNs signaling in $Irgm1^{-/-}$ mice. In the absence of external stimuli, the endogenous DAMPs from dysfunctional mitochondria are the major triggers for the type 1 IFN response (Zhang *et al*, 2010; West *et al*, 2015; Angajala *et al*, 2018). We and others have shown that depletion of IRGM in human cells or mice results in defective autophagy (He *et al*, 2012; Dong *et al*, 2015; Mehto *et al*, 2019) (Appendix Fig S3A and B), and this can result in impaired mitophagy and accumulation of defunct mitochondria. Indeed, we observed a significantly higher number of fused, swollen, and clumped mitochondria in $Irgm1^{-/-}$ BMDMs and IRGM siRNA knockdown cells (Figs 5A and B, and EV4A and B). Next, we scrutinized the status of mitophagy in these cells. An E3 ubiquitin ligase, Parkin mediates mitophagy downstream of ubiquitin kinase, PTEN-induced kinase 1 (PINK1) (Narendra *et al*, 2008; Vives-Bauza *et al*, 2010; Jin & Youle, 2012). The PINK1 and PARKIN themselves are degraded by mitophagy, and a defect in productive mitophagy will result in their accumulation over defective mitochondria. In immunofluorescence assays, we observed substantially increased localization of endogenous ubiquitin, PARKIN and PINK1 over the clumped mitochondria in $Irgm1^{-/-}$ BMDMs compared to control cells (Figs 5C–E and EV4C and D). This phenomenon was more pronounced when YFP-Parkin was overexpressed in BMDMs (Fig EV4D). In western blot analysis, we observed an accumulation of PARKIN, PINK1, TOM20, and cytochrome c in IRGM-depleted human and mouse macrophages compared to the control cells (Figs 5F and G, and EV4E). Further, in the presence of autophagy

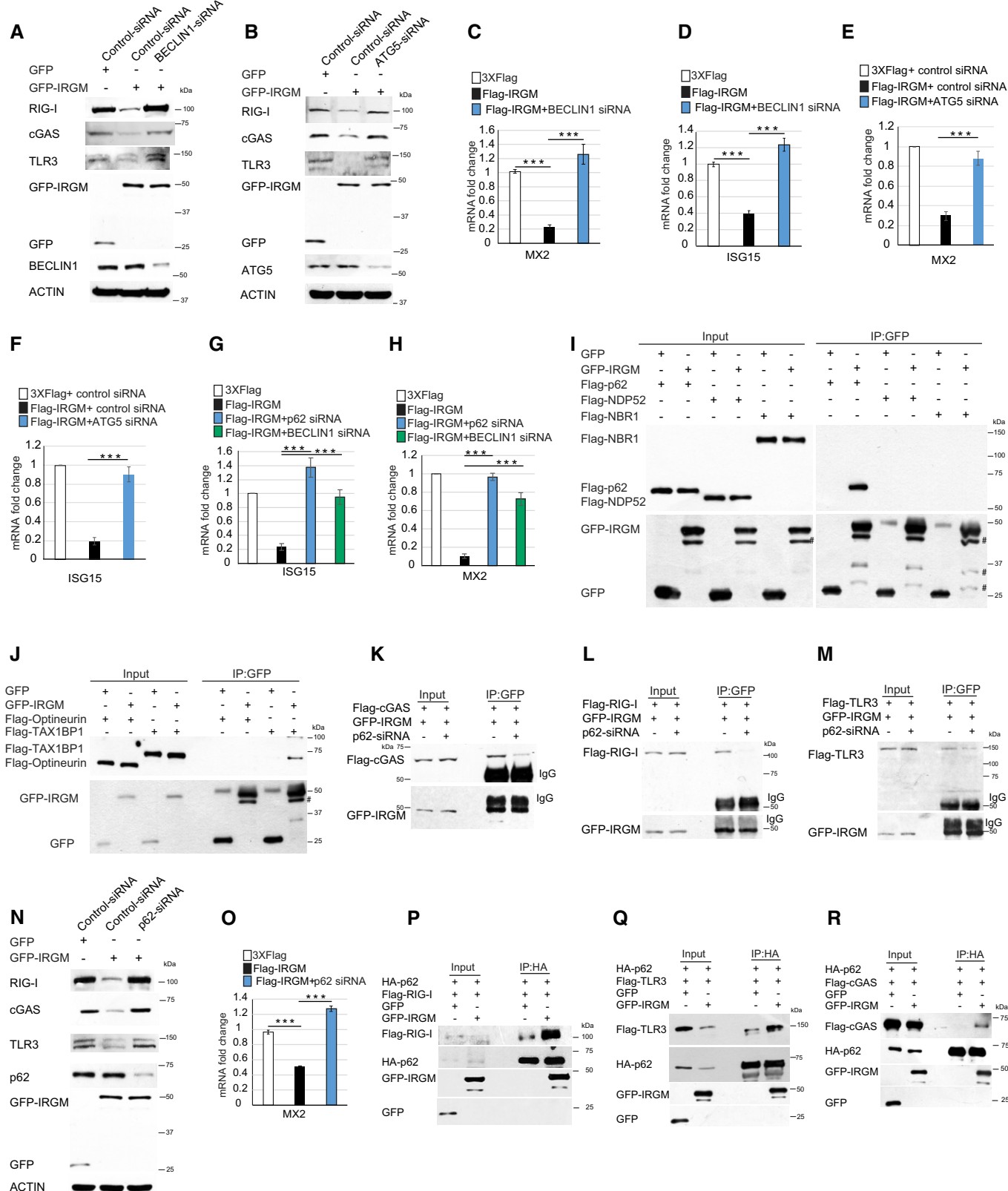

**Figure 4.**

**Figure 4.  IRGM utilizes p62-dependent autophagy to control the levels of nucleic acid sensors and ISGs.**

A       The cell lysates of control or BECLIN1 siRNA transfected THP-1 cells expressing GFP or GFP-IRGM were subjected to immunoblotting with indicated antibodies.
B       The cell lysates of control or ATG5 siRNA transfected THP-1 cells expressing GFP or GFP-IRGM were subjected to immunoblotting with indicated antibodies.
C, D    qRT–PCR analysis with RNA isolated from control or BECLIN1 siRNA transfected THP-1 cells expressing 3X-Flag epitope or Flag-IRGM as indicated. Mean ± SD, $n$ = 3 (biological replicates), ***$P$ < 0.0005, Student's unpaired $t$-test.
E, F    qRT–PCR analysis with RNA isolated from control or ATG5 siRNA transfected THP-1 cells expressing 3X-Flag epitope or Flag-IRGM as indicated. Mean ± SD, $n$ = 3 (biological replicates), ***$P$ < 0.0005, Student's unpaired $t$-test.
G, H    qRT–PCR analysis with RNA isolated from control or p62 siRNA or BECLIN1 siRNA transfected HT29 cells expressing 3X-Flag epitope or Flag-IRGM as indicated. Mean ± SD, $n$ = 3 (biological replicates), ***$P$ < 0.0005, Student's unpaired $t$-test.
I, J    Co-IP analysis of the interaction between GFP-IRGM with (I) Flag-p62 or Flag-NDP52 or Flag-NBR1 or (J) Flag-optineurin or Flag-TAX1BP1 in HEK293T cell lysates.
K–M     Co-IP analysis of the interaction between GFP-IRGM with (K) Flag-cGAS, (L) Flag-RIG-I, (M) Flag-TLR3 in control siRNA or p62 siRNA transfected HEK293T cell lysates.
N       The cell lysates of control or p62 siRNA transfected THP-1 cells expressing GFP or GFP-IRGM (as indicate) were subjected to immunoblotting with indicated antibodies.
O       The qRT–PCR analysis with RNA isolated from control or p62 siRNA transfected THP-1 cells expressing 3X-Flag epitope or Flag-IRGM as indicated. Mean ± SD, $n$ = 3 (biological replicates), ***$P$ < 0.0005, Student's unpaired $t$-test.
P–R     Co-IP analysis of the interaction between HA-p62 and (P) Flag-RIG-I, (Q) Flag-TLR3, (R) Flag-cGAS in the absence and presence of GFP or GFP-IRGM in HEK293T cell lysates.

Data information: #degradation products of full-length GFP-IRGM.
Source data are available online for this figure.

inhibitor bafilomycin A1, the accumulation of PARKIN, PINK1, and cytochrome c is increased in control cells but not in IRGM-depleted cells (Figs 5H and I, and EV4E) suggesting a block of mitophagy flux in IRGM/Irgm1 knockout and knockdown cells. The data indicate that mitophagy flux is impaired in IRGM-depleted cells leading to an accumulation of abnormal clumped rounded mitochondria in these cells.

We found that p62 does not have a role in basal mitophagy in THP1 cells. No differential accumulation of p62 was observed over the mitochondria in IRGM knockdown cells (Appendix Fig S3C). Further, p62 knockdown in THP1 cells did not change mitophagy flux (Appendix Fig S3D).

Next, we performed the Seahorse XF Cell Mito stress test (Fig 5J, Appendix Fig S3E), which is the gold standard assay for measuring mitochondrial function in cells. The basal respiration (energetic demand under baseline conditions), maximal respiration (maximum rate of respiration that the cell can achieve), spare respiration capacity (capability of the cell to respond to an energetic demand under stress conditions), proton leakage (mechanism of ATP production), and ATP production was significantly lower in the Irgm1 knockout BMDM cells compared to control cells (Fig 5J, Appendix Fig S3F) suggesting that the mitochondria of IRGM-depleted cells are defective and are incapable to meet energy demand of the cells.

Next, we used MitoTracker CMXRos™ dye to measure mitochondrial membrane potential. IRGM knockdown THP-1 cells and PBMCs from healthy human donors showed increased CMXRos staining compared to the control cells in flow cytometry analysis and immunofluorescence (Figs 5K–M and EV4F) suggesting an increased polarization of mitochondria in IRGM-deficient cells compared to the control cells. To further verify this finding, we performed JC-1 mitochondrial staining, whose accumulation is dependent only on membrane potential but not on size, shape, and density. At low mitochondrial potential, JC-1 is predominantly a monomer and yields green fluorescence, whereas at the higher mitochondrial potential, it forms aggregates and yield red fluorescence. The IRGM deficiency leads to increased red fluorescence (red to green ratio), suggesting a hyperpolarized state of the mitochondria (Fig 5N and O). These data are consistent with previous

observations where overexpression of IRGM leads to mitochondrial fission and depolarization (Singh et al, 2010), and here we observed fused and hyperpolarized mitochondria in IRGM-deficient cells. The hyperpolarized state of mitochondria is often associated with the production of mitochondrial reactive oxygen species (mtROS) leading to apoptosis or necrosis and also autoimmune diseases including SLE (Gergely et al, 2002a,b; Perl et al, 2004; Galloway & Yoon, 2012; Chen et al, 2017; Angajala et al, 2018). Indeed, the mitochondrial superoxide and cellular total ROS levels, as measured by Mito-SOX™ and CellRox™ dyes, were elevated in IRGM-depleted THP-1 cells, mouse BMDMs and human PBMCs (Figs 5P and Q, and EV4G). Taken together, the data suggest that cellular depletion of IRGM results in defective mitophagy flux leading to the accumulation of hyperpolarized dysfunctional mitochondria, and increased production of MitoROS.

## The cytosol of IRGM-deficient cells is soiled with DAMPs that fuel the IFN response

The endogenous dsDNA needs to be strictly compartmentalized within the nucleus or mitochondria, and any leakage of dsDNA into cytosol results into strong immune stimulation leading to inflammation and autoimmunity (Crow & Rehwinkel, 2009; Roers et al, 2016). The increase in cellular oxidative stress results in mitochondrial and genomic instability leading to the release of mitochondrial DNA (mtDNA) and nuclear DNA (micronuclei) into the cytosol (Gehrke et al, 2013). The mtDNA or micronuclei induces strong interferon response via cGAS-STING-IRF3 axis leading to autoimmune and auto-inflammatory conditions (Pisetsky, 2012; Ablasser et al, 2014; White et al, 2014; West et al, 2015; Mackenzie et al, 2017). Thus, next, we examined the status of dsDNA species in the cytosol. Immunofluorescence assays showed increased mtDNA nucleoids in the cytosol in both human and mouse cells (Figs 6A and EV4H–J). The micronuclei (both lamin B1 bound or unbound) were also substantially increased in IRGM-depleted cells (Figs 6B–E). Extracellular DNA is another potent DAMP, which is released from necrotic cells and can gain access to the cytoplasm of surviving cells to induce inflammation (Pisetsky, 2012). We observed

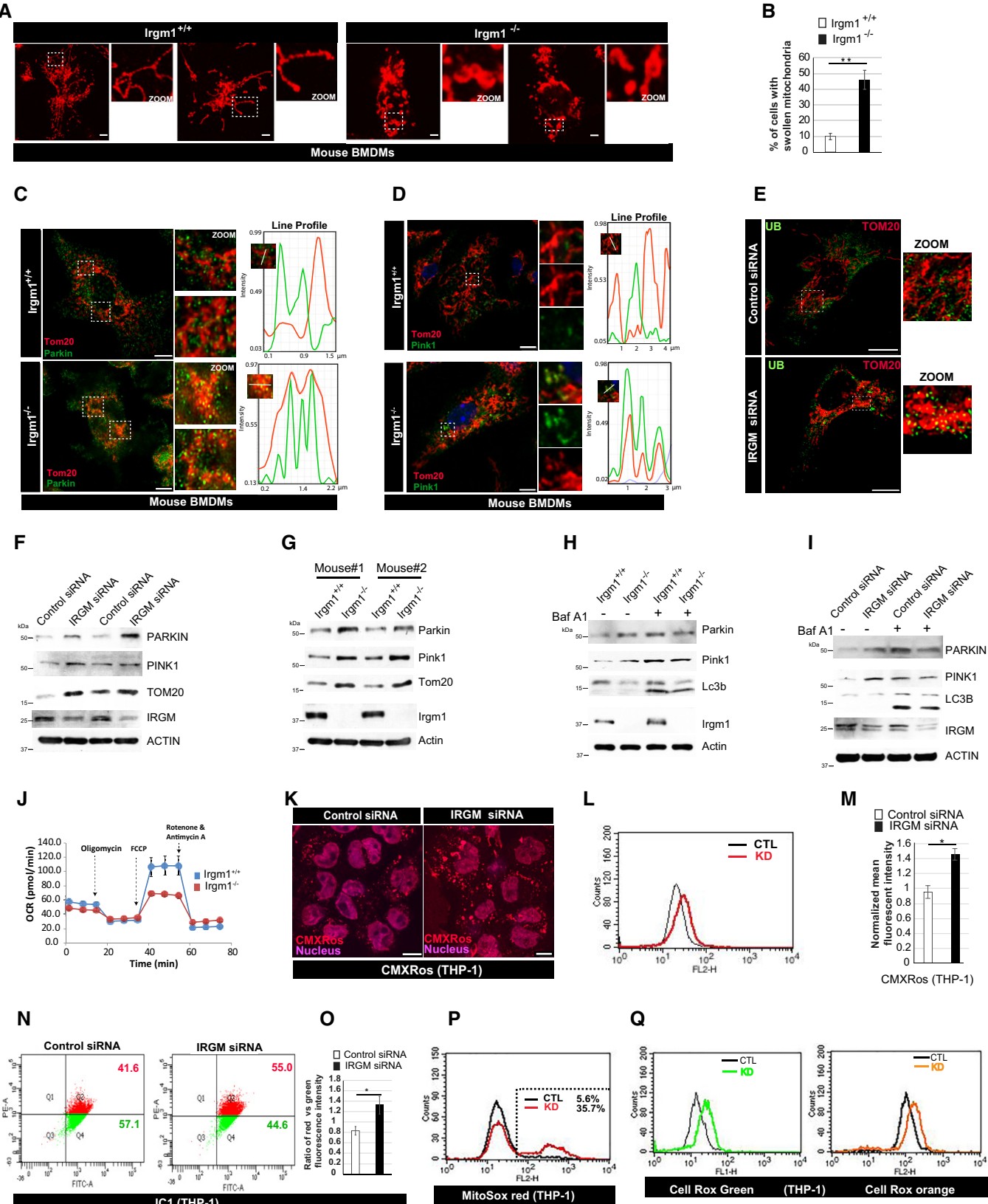

**Figure 5.**

Figure 5. IRGM depletion results in impaired mitophagy and accumulation of dysfunctional/deleterious mitochondria.

A   Representative confocal images of $Irgm1^{+/+}$ and $Irgm1^{-/-}$ mouse BMDMs processed for IF analysis with Tom20 antibody. Scale bar, 5 μm. Zoom panels are digital magnifications.
B   The graph depicts the percentage of cells with swollen/rounded mitochondria in $Irgm1^{+/+}$ and $Irgm1^{-/-}$ mouse BMDMs. Mean ± SD, n = 3 (biological repeats), **$P < 0.005$, Student's unpaired t-test.
C   Representative confocal (STED) images of $Irgm1^{+/+}$ and $Irgm1^{-/-}$ mouse BMDMs processed for IF analysis with Tom20 and Parkin antibodies. Line profile: Co-localization analysis using line intensity profiles. Scale bar, 5 μm. Zoom panels are digital magnifications.
D   Representative confocal (STED) images of $Irgm1^{+/+}$ and $Irgm1^{-/-}$ mouse BMDMs processed for IF analysis with Tom20 and Pink1 antibodies. Line profile: Co-localization analysis using line intensity profiles. Scale bar, 5 μm. Zoom panels are digital magnifications.
E   Representative confocal images of control siRNA and IRGM siRNA transfected THP-1 cells processed for IF analysis with ubiquitin (green) and TOM20 (red) antibodies. Scale bar, 10 μm. Zoom panels are digital magnifications.
F   Western blot analysis of mitochondrial fraction of control siRNA and IRGM siRNA transfected THP-1 cells, probed with indicated antibodies. Two biological replicates are shown.
G   Western blot analysis of mitochondrial fraction of $Irgm1^{+/+}$ and $Irgm1^{-/-}$ mouse BMDM cells, probed with indicated antibodies (n = 2 mice).
H   Western blot analysis of mitochondrial fraction of $Irgm1^{+/+}$ and $Irgm1^{-/-}$ mouse BMDM cells untreated or treated with bafilomycin A1 (300 nM, 3 h), probed with indicated antibodies.
I   Western blot analysis of mitochondrial fraction of control siRNA and IRGM siRNA transfected THP-1 cells untreated or treated with bafilomycin A1 (300 nM, 3 h), probed with indicated antibodies.
J   Seahorse XF Cell Mito stress test for analysis of mitochondrial function in $Irgm1^{+/+}$ and $Irgm1^{-/-}$ mouse BMDM cells. The experiments were performed using the XF24 extracellular flux analyzer, and the flow chart showed the measurement of oxygen consumption rate (OCR) as described in materials and methods. Results shown represent mean ± standard error (n = 3 mice). *$P < 0.05$.
K   Representative confocal images of control and IRGM siRNA transfected THP-1 cells stained with CMXRos red dye. Scale bar, 7.5 μm.
L   Flow cytometry analysis of control siRNA and IRGM siRNA transfected THP-1 cells stained with CMXRos red dye (10 nM, 30 min).
M   Graph depicts the normalized mean fluorescent intensity of control and IRGM knockdown THP-1 cells stained with CMXRos. Mean ± SD, n = 3 (biological repeats), *$P < 0.05$, Student's unpaired t-test.
N   Representative dot plot showing flow cytometry analysis of control and IRGM siRNA knockdown THP-1 cells stained with JC-1 dye (2 μM, 30 min). At low mitochondrial membrane potential, JC-1 is predominantly a monomer that yields green fluorescence, whereas at mitochondrial membrane potential the dye aggregates producing a red to orange colored emission.
O   The graph depicts the ratio of red vs. green fluorescent intensity in control and IRGM siRNA knockdown THP-1 cells stained with JC-1 dye. Mean ± SD, n = 3 (biological replicates), *$P < 0.05$, Student's unpaired t-test.
P   Representative flow cytometry analysis of control and IRGM siRNA transfected THP-1 cells stained with MitoSox red dye (1 μM, 20 min). The percentage of control and IRGM knockdown cells with increased red fluorescence (mitochondrial ROS generation) is depicted.
Q   Representative flow cytometry analysis of control and IRGM siRNA transfected THP-1 cells stained with CellRox green and CellRox orange dye (1 μM, 30 min).

Source data are available online for this figure.

increased extracellular DNA in the immunofluorescence slides of IRGM-depleted cells (Appendix Fig S3G and H). Next, we performed quantitative real-time PCR with primers of cytochrome c oxidase subunit, which is one of three mtDNA encoded subunits of respiratory complex IV, to assess the status of mtDNA in cytosol (devoid of mitochondria). The data show several fold increase of mtDNA in cytosol of IRGM-depleted mice and human cells compared to the control cells (Fig 6F and G).

To determine whether cytosolic mtDNA acts as a DAMP to induce type 1 IFN response in IRGM-depleted cells, we depleted the mtDNA using standard ethidium bromide method and generated $Rho^0$ cells in HT29 IRGM stable knockdown cells. The cytosolic mtDNA is significantly reduced in IRGM knockdown $Rho^0$ cells compared to parent cells as measured by qRT–PCR with cytochrome c oxidase subunit 1 primers (Fig 6H). Additionally, the increased ISGs (MX2 and ISG15) levels were considerably rescued in IRGM knockdown $Rho^0$ cells compared to parent cells (Fig 6I). The data suggest that indeed mtDNA plays a significant role in the induction of interferon response in IRGM-depleted cells. We employed another method to reconfirm whether cytosolic DNA is indeed responsible for augmented type I IFN response in IRGM knockdown cell, and we electroporated DNase I enzyme in these cells to deplete cytosolic DNA and then performed qRT–PCR to measure ISG expression. No cell toxicity was observed in cells due to DNase 1 electroporation. The increased mtDNA in cytosol is reduced upon DNase 1 electroporation (Fig 6J). The DNase 1 treatment significantly reduced the elevated expression of ISGs in IRGM knockdown cells (Fig 6K).

Taken together, the results suggest that cytosolic mtDNA provokes type I IFN signaling in IRGM-depleted cells.

If inhibition of mitophagy is one of the important reasons for heightened ISGs in IRGM knockdown cells; then, the depletion of PARKIN might also increase ISGs. Indeed, there was a significant increase of cytosolic mtDNA, and ISGs in PARKIN-depleted THP-1 cells (Fig EV4K, Appendix Fig S4A). PRRs expression was also increased (Fig EV4L). However, the increase of expression of ISGs and PRRs was considerably less as compared to IRGM knockdown cells (Figs EV4K and L, and 2), suggesting that PARKIN-dependent mitophagy may contribute but may not the sole reason for the heightened IFN response in IRGM knockdown cells. We attempted another experiment to understand the role of mitophagy in the IRGM-mediated IFN response. We used Mdivi1, known mitochondrial fission, and mitophagy inhibitor (Luo et al, 2019; Vo et al, 2019; Yao et al, 2019). The results show that (i) mitophagy inhibition indeed induces interferon response, but the increase is lesser than the induction of interferon response in IRGM knockdown cells (Fig EV4M). Again, this indicates that other factors, in addition to mitophagy, contribute to IFN induction in IRGM-depleted cells. (ii) There is no further increase of IFN response in IRGM-depleted cells upon Mdivi1 treatment, suggesting that mitophagy is already maximally inhibited in IRGM-depleted cells (Fig EV4M).

The viral and mitochondrial dsRNA can trigger the RIG-I/MDA5-MAVS-IRF3 signaling pathway leading to enhanced interferon production (Reikine et al, 2014; Dhir et al, 2018; Linder & Hornung, 2018). We performed immunofluorescence assays with extensively

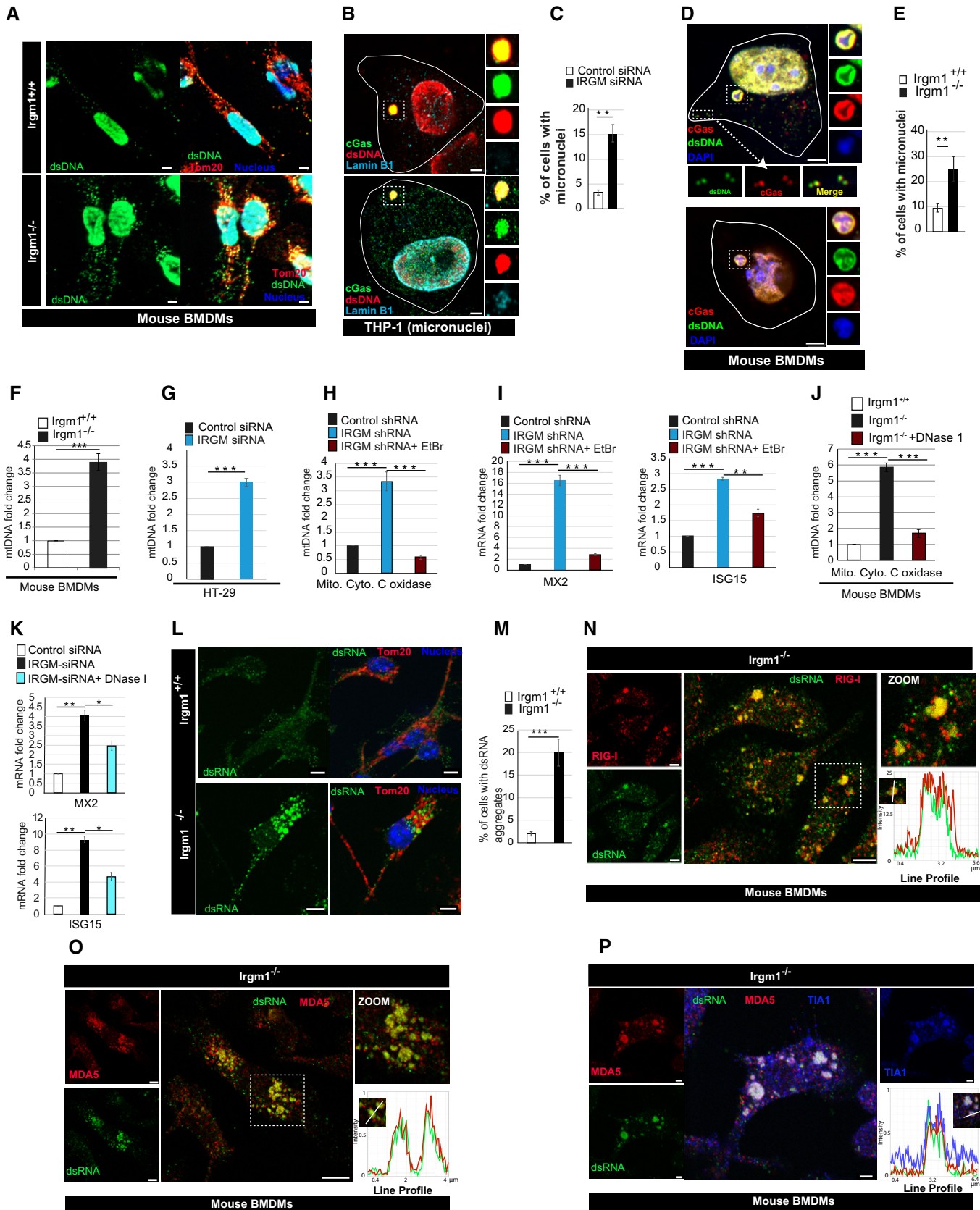

**Figure 6.**

**Figure 6. IRGM-depleted cells are soiled with DAMPs that induces IFN response.**

A   Representative confocal images of *Irgm1*$^{+/+}$ and *Irgm1*$^{-/-}$ mouse BMDMs processed for IF analysis with Tom20 (red) and dsDNA (green) antibody. Scale bar, 5 μm.

B   Representative STED microscopy images of control THP-1 cells processed for IF analysis with cGAS (green), dsDNA (red), and Lamin B1 (cyan) antibodies. Scale bar, 3 μm. Zoom panels are digital magnifications.

C   The graph depicts the percentage of cells with micronuclei in control siRNA and IRGM siRNA knockdown THP-1 cells. Mean ± SD, *n* = 3 (biological replicates), **$P$ < 0.005, Student's unpaired *t*-test.

D, E   Representative STED microscopy images of *Irgm1*$^{-/-}$ mouse BMDMs (D) transfected with mCherry-cGAS and immunostained with dsDNA (green) antibody. The white line depicts the periphery of the cells. Scale bar, 3 μm. (E) The graph depicts the percentage of cells with micronuclei in *Irgm1*$^{+/+}$ and *Irgm1*$^{-/-}$ mouse BMDMs. Mean ± SD, *n* = 3 (biological replicates), **$P$ < 0.005, Student's unpaired *t*-test. Zoom panels are digital magnifications.

F   The qRT–PCR analysis with cytosolic DNA (minus mitochondria) isolated from *Irgm1*$^{+/+}$ and *Irgm1*$^{-/-}$ mouse BMDMs. Mean ± SE, *n* = 3 mice, ***$P$ < 0.0005, Student's unpaired *t*-test.

G   The qRT–PCR analysis with cytosolic DNA (minus mitochondria) isolated from control and IRGM siRNA transfected HT-29 cells. Mean ± SE, *n* = 3 (biological replicates), ***$P$ < 0.0005, Student's unpaired *t*-test.

H   The qRT–PCR analysis with cytosolic DNA (minus mitochondria) isolated from control or IRGM shRNA or EtBr treated IRGM shRNA HT-29 (Rho$^0$) cells. Mean ± SE, *n* = 3 (biological replicates), ***$P$ < 0.0005, Student's unpaired *t*-test.

I   The qRT–PCR analysis with total RNA isolated from control or IRGM shRNA or EtBr treated IRGM shRNA HT-29 (Rho$^0$) cells. Mean ± SD, *n* = 3 (biological replicates), **$P$ < 0.005, ***$P$ < 0.0005, Student's unpaired *t*-test.

J   The qRT–PCR analysis with cytosolic DNA (minus mitochondria) isolated from *Irgm1*$^{+/+}$ and *Irgm1*$^{-/-}$ mouse BMDMs transfected with DNase I (15 μg, 6 h). Mean ± SE, *n* = 3 mice, ***$P$ < 0.0005, Student's unpaired *t*-test.

K   The qRT–PCR analysis with RNA isolated from control and IRGM siRNA knockdown THP-1 cells electroporated with DNase I (5 μg, 1 h) as indicated. Mean ± SD, *n* = 3 (biological replicates), *$P$ < 0.05, **$P$ < 0.005, Student's unpaired *t*-test.

L, M   Representative confocal images of *Irgm1*$^{+/+}$ and *Irgm1*$^{-/-}$ mouse BMDMs processed for IF analysis with (L) dsRNA (green) and Tom20 (red) antibodies. (M) The graph depicts the percentage of *Irgm1*$^{+/+}$ and *Irgm1*$^{-/-}$ mouse BMDMs with dsRNA aggregates. Mean ± SD, *n* = 3 (biological replicates), ***$P$ < 0.0005, Student's unpaired *t*-test. Scale bar, 5 μm.

N   Representative confocal images of *Irgm1*$^{-/-}$ mouse BMDMs processed for IF analysis with RIG-I (red) and dsRNA (green) antibodies. Line profile: Co-localization analysis using line intensity profiles. Scale bar, 5 μm. Zoom panels are digital magnifications.

O, P   Representative confocal images of *Irgm1*$^{-/-}$ mouse BMDMs processed for IF analysis with (O) Mda5 (red) and dsRNA (green) antibodies (Scale bar, 5 μm) or (P) Mda5 (red), dsRNA (green) and Tia1 (blue) antibodies (Scale bar, 3 μm). Line profile: Co-localization analysis using line intensity profiles. Zoom panels are digital magnifications.

used dsRNA-specific J2 antibody to explore the possibility of the presence of dsRNA in the cytosol of IRGM-depleted cells. A low level of endogenous dsRNA, which partly co-localizes with mitochondria, was observed in control cells (Fig 6L). In a very surprising observation, a significantly higher number of *Irgm1*$^{-/-}$ BMDMs showed an abundant number of dsRNA structures in the cytosol (Fig 6L and M, Appendix Fig S4B). Endogenous RIG-I and MDA5 showed complete co-localization with these dsRNA structures (Fig 6N and O, Appendix Fig S4C and D). Since we found that IRGM-deprived cells are under constitutive oxidative stress, we tested whether these dsRNA structures are the stress granules. Indeed, the stress granules marker, TIA1, completely co-localized with the dsRNA structures suggesting that they are the stress granules (Fig 6P). We electroporated dsRNA-specific RNase (RNase III) and RNA-DNA hybrid-specific RNase (RNase H) in the IRGM knockdown cells to deplete cytosolic dsRNA and then performed qRT–PCR to measure ISG expression. Contrary to DNase 1 exposure, the treatment of RNase resulted in increased cell toxicity, so the treatment was performed only for 1 h (no cell death at this time point). Nevertheless, the heightened ISG expression in IRGM-depleted human and mouse BMDMs was significantly reduced upon RNase treatment (Appendix Fig S4E and F). Our attempt to electroporate both DNAse I and RNases together is failed due to cytotoxicity. From the data presented here, we conclude that increased DNA and dsRNA in the cytosol of IRGM-deficient cells trigger enduring type 1 IFN signaling and response.

## Both cGAS-STING and RIG-I/MDA5-MAVS signaling contributes to enduring type 1 IFN responses in IRGM-depleted cells

The cytosolic DNA sensor cGAS was found to be perfectly co-localized with micronuclei and also with cytosolic nucleoid in the IRGM

knockout cells. This indicates that the cGAS-STING axis could account for heightened type I IFN response and augmented ISG production in IRGM-depleted cells. Indeed, the siRNA knockdown of cGAS and STING in IRGM-depleted HT29, THP-1, and mouse BMDMs significantly diminished the increased expression of IFN-β and ISGs (MX2 and ISG15) (Figs 7A–C and EV5A).

Upon viral infection, RIG-I localizes over the stress granules to enhance type 1 IFN response (Onomoto *et al*, 2012; Kuniyoshi *et al*, 2014). The presence of RIG-I/MDA5 in the atypical dsRNA stress granules in IRGM knockout BMDMs could be important for the activation of RIG-I/MDA5-mediated IFN signaling. Next, using siRNA, we depleted RIG-I, MDA5, MAVS, and IRF3 in IRGM knockdown cells (for knockdown efficiency, Appendix Fig S5A–F) to explore whether RIG-I/MDA5-MAVS-IRF3 axis contributes to elevated type I IFN levels. Indeed, the knockdown of RIG-I, MDA5, MAVS, and IRF3 in IRGM-depleted human and mouse cells abated the heightened type I interferon response as scored by expression of IFN-β and ISGs (Figs 7A–E and EV5A–D).

Since both DNA (cGAS-STING) and RNA (RIG-I-MAVS) sensing axis appear to contribute to augmented IFN response in IRGM-deficient cells, we tested whether depleting the sensors/adaptors of both pathways simultaneously may have a synergistic effect. Indeed, double knockdown of cGAS and RIG-I or STING and MAVS in IRGM-depleted HT29 and THP-1 cells resulted in a greater decrease of type 1 IFN response as compared to individual knockdowns (Figs 7A–C and EV5A, and C) suggesting that both DNA and RNA sensing pathways are activated in IRGM-deficient cells leading to augmented and sustained type I IFN response.

STAT1 and STAT2 are the transcription factors of the JAK-STAT pathway, which binds to ISRE (interferon-stimulated response element) to induce transcription of ISGs. In THP-1 cells, the depletion of STAT2 in IRGM knockdown cells completely abolished the

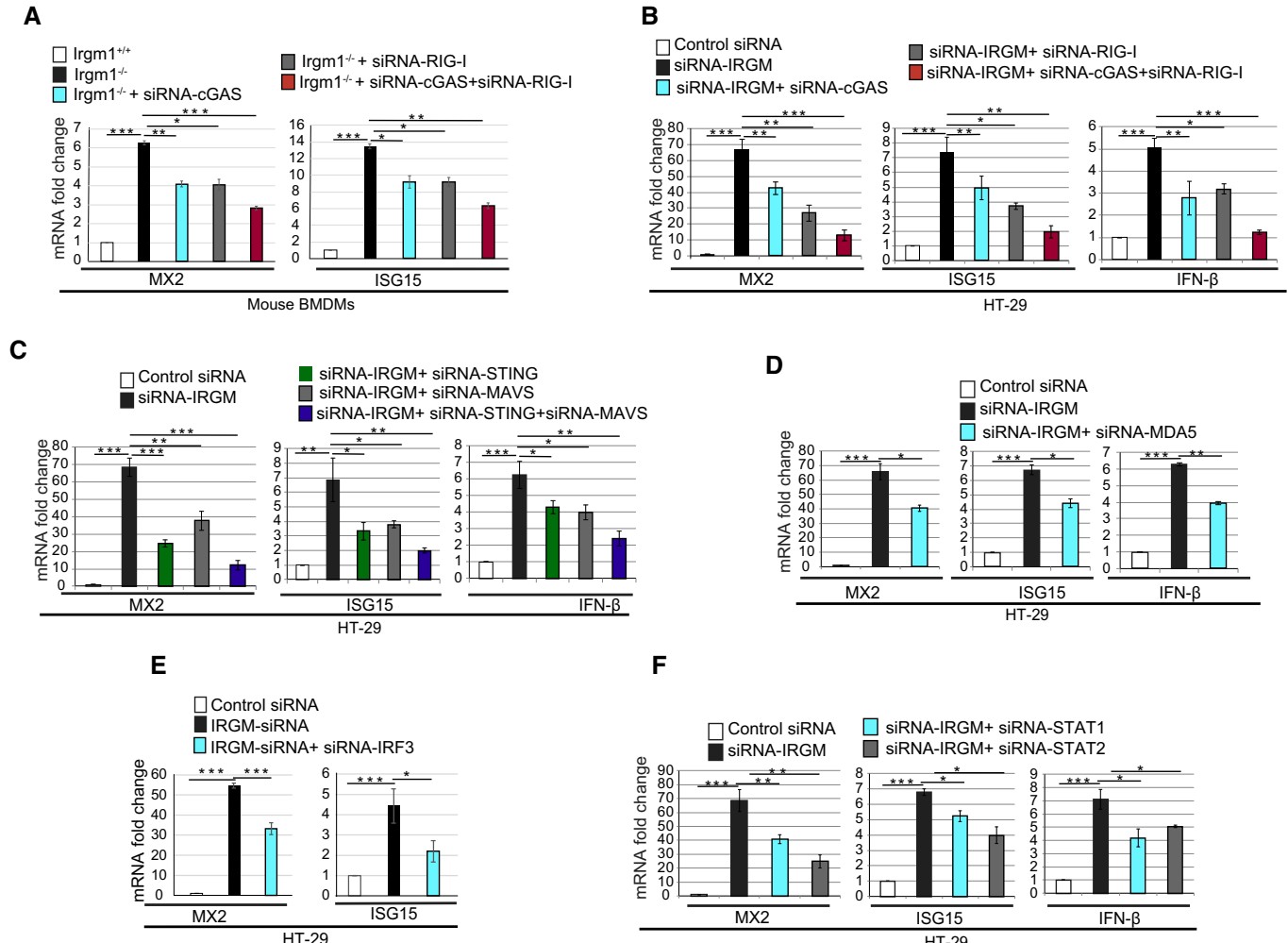

**Figure 7. Both cGAS-STING and RIG-I/MDA5-MAVS pathways contribute to enduring type 1 IFN response via JAK/STAT signaling in IRGM-depleted cells.**

A  qRT–PCR analysis with total RNA isolated from *Irgm1*[+/+] and *Irgm1*[−/−] mouse BMDMs transfected with control siRNA or cGAS siRNA or RIG-I siRNA or cGAS siRNA and RIG-I siRNA as indicated. Mean ± SD, *n* = 3 (biological replicates), *P* < 0.05, **P* < 0.005, ***P* < 0.0005, Student's unpaired *t*-test.

B  qRT–PCR analysis with total RNA isolated from HT-29 cells transfected with control siRNA or IRGM siRNA or doubly transfected with IRGM siRNA and cGAS siRNA or IRGM siRNA and RIG-I siRNA or transfected with three siRNAs as indicated. Mean ± SD, *n* = 3 (biological replicates), *P* < 0.05, **P* < 0.005, ***P* < 0.0005, Student's unpaired *t*-test.

C  qRT–PCR analysis with total RNA isolated from HT-29 cells transfected with control siRNA or IRGM siRNA or doubly transfected with IRGM siRNA and STING siRNA or IRGM siRNA and MAVS siRNA or transfected with all the three siRNAs as indicated. Mean ± SD, *n* = 3 (biological replicates), *P* < 0.05, **P* < 0.005, ***P* < 0.0005, Student's unpaired *t*-test.

D  qRT–PCR analysis with total RNA isolated from HT-29 cells transfected with control siRNA or IRGM siRNA or doubly transfected with IRGM siRNA and MDA5 siRNA as indicated. Mean ± SD, *n* = 3 (biological replicates), *P* < 0.05, **P* < 0.005, ***P* < 0.0005, Student's unpaired *t*-test.

E  qRT–PCR analysis with total RNA isolated from HT-29 cells transfected with control siRNA or IRGM siRNA or doubly transfected with IRGM siRNA and IRF3 siRNA as indicated. Mean ± SD, *n* = 3 (biological replicates), *P* < 0.05, ***P* < 0.0005, Student's unpaired *t*-test.

F  qRT–PCR analysis with total RNA isolated from HT-29 cells transfected with control siRNA or IRGM siRNA or doubly transfected with IRGM siRNA and STAT1 siRNA or IRGM siRNA and STAT2 siRNA as indicated. Mean ± SD, *n* = 3 (biological replicates), *P* < 0.05, **P* < 0.005, ***P* < 0.0005, Student's unpaired *t*-test.

increased expression of ISGs (Fig EV5E), whereas STAT1 depletion has a marginal effect (Fig EV5E) suggesting that STAT2 is more important for the increased expression of ISGs in IRGM-depleted THP-1 cells. However, in HT-29 cells, both STAT1 and STAT2 appear to be equally important for increased expression of ISGs (Fig 7F).

It appears that increased ROS in IRGM-depleted cells play a role in triggering the events, eventually inducing the ISGs. We tested

whether depletion of ROS by *N*-acetyl-ʟ-cysteine (NAC) affects the level of ISGs in IRGM knockdown cells. Indeed, exposure of IRGM-depleted cells with NAC for just 2 h significantly dampens the induced expression of ISGs (Fig EV5F), suggesting that certainly, ROS plays an important role in eliciting the events finally leading to heightened type I IFN response in IRGM knockdown cells.

Collectively, these data suggest that the events triggered by reduced mitophagy flux in IRGM-depleted cells culminating in the

production of DAMPs stimulate DNA/RNA sensing-signaling axes to produce IFNs, which then activate JAK-STAT pathways for ISG production.

## Discussion

This study reveals that IRGM is a master switch that controls the levels of type I IFN response by controlling the activation of DNA/RNA sensing inflammatory pathways. A prominent role of IRGM in suppressing inflammation presented here or previously (Chauhan et al, 2015; Mehto et al, 2019) justify why IRGM protein, which was dead for 20 million years of evolution might have revived back in ancestors of humans (Bekpen et al, 2009).

Our results suggest that IRGM controls the activation of IFN response by mediating autophagic degradation of nucleic acid sensors and promoting mitophagy. The depletion of IRGM results in increased expression of PRRs and accumulation of mtDAMPs producing defective mitochondria. A large number of experiments performed here suggest that both of these processes contribute to the elevated level of type IFN response in IRGM-depleted cells. However, it is difficult to determine the extent of contributions by these two events as both are highly interconnected processes.

### IRGM is a molecular link between autophagy and inflammation

Immunity Related GTPase M/Irgm1 is important for productive autophagic flux in both humans and mice (Singh et al, 2006, 2010; Chauhan et al, 2015; Dong et al, 2015; Hansen et al, 2017; Kumar et al, 2018; Mehto et al, 2019). IRGM interacts and stabilizes autophagy proteins, ULK1, ATG16L1, and BECLIN1 to mount antimicrobial autophagy in the immune cells (Chauhan et al, 2015). Utilizing similar autophagy machinery, IRGM mediates the degradation of NLRP3 and ASC (Mehto et al, 2019). A recent study shows the IRGM role in autophagosome–lysosome fusion (Kumar et al, 2018). Here, we found that IRGM selectively interacts with cGAS/STING, RIG-I, and TLR3 but not with TLR4, TLR7, AIM2, and NLRC4. Also, IRGM mediates BECLIN1 and p62-dependent autophagic degradation of cGAS, RIG-I, and TLR3 to control IFN and ISG production. The autophagy adaptor proteins such as p62, TAX1BP1, OPTN (optineurin), NDP52, and NBR1 recognizes the autophagy cargoes for selective degradation (Svenning & Johansen, 2013; Kim et al, 2016). Among these autophagy adaptor proteins, IRGM preferentially interacts with p62, an adaptor protein that is most commonly utilized for degradation of inflammation pathway proteins. In the agreement, we found that IRGM-mediated autophagic degradation of nucleic acid sensor proteins was dependent on p62. Taken together, these studies suggest that IRGM because of its capability to interact with both autophagy proteins and inflammatory signaling proteins can act as a molecular link between autophagy and inflammation. Specifically, it can utilize the autophagy machinery to degrade inflammatory proteins to maintain immune homeostasis.

### IRGM is vital in maintaining healthy mitochondria

The dysfunctional mitochondria are one of the major sources of DAMPs (Zhang et al, 2010; Li & Chen, 2018). The mtDNA is sensed by the cGAS-STING axis leading to sustained production of type I IFN response and autoimmunity (Ablasser et al, 2014; Rongvaux et al, 2014; White et al, 2014; Gao et al, 2015; West et al, 2015). IRGM is localized over mitochondria and mediates mitochondrial fission (Singh et al, 2010). The overexpression of IRGM leads to mitochondrial depolarization (Singh et al, 2010). We found that deletion or depletion of IRGM/Irgm1 resulted in inhibition of mitophagy flux leading to the accumulation of rounded and swollen hyperpolarized mitochondria. Hyperpolarized mitochondria are often associated with ROS production and autoimmune conditions (Gergely et al, 2002a,b; Perl et al, 2004). Indeed, the IRGM-depleted cells showed enhanced production of mtROS and cellular ROS. Increased ROS is known to cause DNA damage leading to the release of mtDNA nucleoid and micronuclei in the cytoplasm. The same was observed in IRGM-depleted cells. The presence of ROS also induces stress granules formation. IRGM-depleted cells showed an increased amount of cytoplasmic dsRNA containing stress granules like structure, where MDA5/RIG-I co-localized with dsRNA. We found N-acetylcysteine (a ROS quencher), cytosolic DNA depletion (ethidium bromide and DNase 1), and knockdown of cGAS, STING, RIG-I, MAVS, and IRF3 ablated the enhanced IFN response in IRGM knockdown cells. These data suggest that indeed increased mtROS and DAMPs induce ISG expressions via cGAS/STING and RIG-I-MAVS-IRF3 axis. A series of events that initiated from dysfunctional mitophagy flux leads to aberrant IFN/ISG production in IRGM-depleted cells. We found that the depletion of IRGM results in a more robust IFN response than in depletion of p62 (selective autophagy inhibition) or BECLIN1 (bulk and selective autophagy inhibition) or PARKIN (mitophagy inhibition). These results suggest that IRGM controls additional pathways to modulate IFN response.

### Loss of IRGM invokes a vicious cycle of events that endorse cellular Inflammation

The SNPs and deletion polymorphisms in promoter and exonic regions of IRGM have been shown to be strongly associated with susceptibility to Crohn's disease and other autoimmune diseases (McCarroll et al, 2008; Brest et al, 2011; Xia et al, 2017; Yao et al, 2018). Both of these types of genetic changes reduce IRGM expression (McCarroll et al, 2008; Brest et al, 2011), suggesting that decreased expression of IRGM enhances the susceptibility to autoimmune and auto-inflammatory diseases. The present and our previous study (Mehto et al, 2019) show that depletion of IRGM triggers a vicious cycle of inflammation leading to increased susceptibility to inflammatory diseases. We found that in the absence of IRGM, several inflammatory axis including the cGAS-STING, RIG-I-MAVS, and NLRP3-ASC get activated, leading to the production of strong inflammatory cytokines, including IL-1β and type 1 IFNs. The secretion of these cytokines induces inflammatory cell death (Mehto et al, 2019), resulting in the release of DAMPs, which in the feedback loop further fuels the inflammatory signaling pathways to produce more cytokines. These cytokines are chemoattractant for immune cells. The immune cells infiltrate and get activated, releasing more cytokines and also the production of autoantibodies. We envisage that this vicious feed-forward cycle of inflammation is triggered in autoimmune patients carrying mutations in the IRGM gene. Further studies are needed in human patients to test this hypothesis. Nevertheless, this study defines the mechanism by which IRGM is a

master regulator of IFN responses and highlights IRGM as a strong potential target for new therapeutic interventions against inflammatory/autoimmune diseases.

# Materials and Methods

### Cell culture

The cell lines, HT29 (ATCC #HTB-38), THP-1 (ATCC #TIB-202), HEK293T (ATCC #CRL-11268) were obtained from American Type Culture Collection (ATCC). HEK293T, HT29 cells were grown in DMEM (Gibco #10569044) supplemented with 10% fetal bovine serum (FBS) and penicillin/streptomycin (10,000 units/ml). Human monocytic cell line, THP-1 and BMDMs were grown in RPMI-1640 (Gibco #61870127) media supplemented with 10% FBS, 5 mM L-glutamine, glucose (5%), HEPES buffer, sodium pyruvate (1 mM), penicillin/streptomycin (10,000 units/ml). All the experiments were performed with cells before the 20[th] passage was reached.

### Isolation of human peripheral blood mononuclear cells

Isolation of human PBMCs was performed using the density gradient centrifugation method using Histopaque (Sigma #10771) as per the manufacturer's recommendation. Briefly, 5 ml of homogenized blood was mixed with an equal volume of DPBS and slowly layered on the 3 ml Histopaque. The sample was then centrifuged at 400 $g$ for 20 min at room temperature (RT) using slow acceleration and deceleration. Buffy coat generated in the middle of the gradient was collected, mixed with DPBS, and centrifuged at 100 $g$ for 10 min at RT. Cells were then washed twice with DPBS and resuspended in RPMI-1640 supplemented with 10% FBS and incubated at 37°C with 5% $CO_2$ overnight. Cells were counted in an Invitrogen countless II automated cell counter using an equal volume of trypan blue (Sigma #T8154). Total RNA was extracted using TRIzol (Invitrogen #15596018). The human blood samples were collected as per the standard protocols and procedures approved by the institutional human ethical committee at the Institute of Life Science, Bhubaneswar, India.

### CRISPR knockout cell line

For the generation of IRGM knockout in cell lines, HT-29 cells were transfected with an IRGM CRISPR/Cas9 KO plasmid (sc-407889) and IRGM HDR plasmid (sc-407889-HDR) using Lipofectamine 2000 DNA transfection reagents(Invitrogen #11668027). After 48 h of transfection, cells were treated with 2 μg/ml puromycin (InvivoGen #ant-pr-1) for 7 days. Single clones were selected, and IRGM knockout was confirmed by western blotting using anti-IRGM antibody.

### Lentivirus production and generation of stable IRGM knockdown cells

For the generation of lentivirus, HEK293T cells cultured in 15 cm plate and transfected with control shRNA or IRGM shRNA (Santa Cruz #TRCN0000197363; Sequence: CCGGCCGGTATGACTTCATCA TGGTCTCGAGACCATGATGAAGTCATACCGGTTTTTTG) plasmid (22.5 μg), the pMD2G plasmid (7.9 μg) and pCMVR 8.74 (14.6 μg)

using CalPhos Mammalian Transfection Kit (Clonetech #631312). After 36 h of transfection, the viruses in the culture medium were harvested and centrifuged at 500 $g$ for 5 min at 4°C. Culture medium was then filtered through 0.22 μm filter and used for the transduction and generation of stable cell lines.

For the generation of stable IRGM knockdown cells, HT29 cells were plated in six-well plates and transduced with virus particles. After 24 h, the medium was replaced and kept for another 48 h. IRGM stable knockdown cells were selected using 2 μg puromycin for 1 week. IRGM knockdown was confirmed by western blotting using anti-IRGM antibody and qRT–PCR using IRGM TaqMan gene expression assay.

### Transient transfection with siRNA

The THP-1 cells were electroporated (Neon, Invitrogen #MPK5000; setting: 1,400 V, 10 ms, three pulses using 100 μl tip (Neon, Invitrogen #MPK10096)) with non-targeting siRNA (30 nM) or specific siRNA (30 nM) and incubated for 24 h. Another round of siRNA transfection was performed after 24 h in similar conditions as described above and incubated for the next 48 h. The siRNA (10 nM) transfection in HT29 cells was performed using the Lipofectamine® RNAiMAX (Invitrogen #13778075) as per the manufacturer's instructions. Following siRNA were used in the present study: non-specific siRNA (SMARTpool: siGENOME ns siRNA; Dharmacon #D-001206-13-20),IRGM siRNA (SASI_HS02_00518571), p62 siRNA (SASI_Hs01_00118616), BECLIN1 siRNA (SASI_Hs02_ 00336256), Human RIG-I siRNA (SASI_Hs01_00047980), TLR3 siRNA (SASI_Hs01_00231802), Human cGAS siRNA (SASI_Hs01_ 00197466), MAVS siRNA (SASI_Hs01_00128708), MDA5 siRNA (SASI_Hs01_00171929), STING siRNA (SASI_Hs02_00371843), IRF3 siRNA (SASI_Hs02_00332144), STAT1 siRNA (SASI_Hs02_ 00343387), STAT2 siRNA (SASI_Hs01_00111824), ATG5 siRNA (SASI_Hs01_00173156), PARK2 (SASI_Hs01_00041567), TAX1BP1 siRNA (SASI_Hs02_00313970), NBR1 siRNA (SASI_Hs01_ 00169473), NDP52 siRNA (SASI_Hs01_00137944), Mouse RIG-I siRNA (SASI_Mm01_00145086), Mouse cGAS siRNA (SASI_Mm01_ 00129826).

### Transient transfection with plasmids

For transient expression, HEK293T cells were transfected with plasmids using the calcium phosphate method. THP-1 cells were transfected using the Neon electroporation system with the following parameters: 1,300 V, 30 ms, one pulse. Briefly, $2 \times 10^6$ THP-1 cells were transfected with p62 (30 nM) or BECLIN1 (30 nM) siRNA. After 72 h, the cells were transfected with EGFP (3 μg) or GFP-IRGM (3 μg) or 3X-Flag or Flag-IRGM (3 μg). After 4 h, cell lysates were prepared, and western blotting was performed.

### DNase I Transfection in THP-1 cells

Approximately $3 \times 10^6$ THP-1 cells were electroporated (Neon, Invitrogen; setting: 1,400 V, 10 ms, three pulses using the 100 μl tip) with non-targeting siRNA (30 nM) or IRGM siRNA (30 nM). After 24 h, the cells were again transfected with siRNAs. Next day, the cells were electroporated (1,400 V, 10 ms, two pulses using the 100 μl tip) with either 10 μg or 5 ug BSA (as control) or 10 μg or

5 µg DNAse I enzyme (NEB #M0303S). The cells were harvested after indicated time points and analyzed by qRT–PCR.

## RNase III and RNase H transfection in THP-1 cells

Approximately $3 \times 10^6$ THP-1 cells were electroporated (Neon, Invitrogen; setting: 1,400 V, 10 ms, three pulses using the 100 µl tip) with non-targeting siRNA (30 nM) or IRGM siRNA (30 nM). After 24 h, the cells were again transfected with siRNAs. The next day, the cells were electroporated (1,400 V, 10 ms, two pulses using the 100 µl tip) with either 10 µg BSA (as control) or 10 units of RNase III (NEB #M0245S) and RNase H (NEB #M0297S) enzymes each. The cells were harvested after 1 h and analyzed by qRT–PCR.

## Sample preparation for RNA sequencing

RNA was extracted from brain tissues and BMDMs of three different mice and HT29 cells (three biological replicates) using RNeasy mini kit (QIAGEN, #74104). The quality and quantity of total RNA were checked using agarose gel and Qubit 3.0. After assessing the quality of RNA, 800–900 ng of total RNA was subjected to NEBNext® Ultra™ directional RNA library prep kit for Illumina® (#E7420) using NEBNext Poly(A) mRNA magnetic isolation module (NEB #E7490). The QC of the prepared library was performed using High-Sensitivity Tape Station Kit (Agilent 2200, 5067-5585 [reagents] #5067-5584 [screen tapes]) for fragment length distribution and Qubit dsDNA HS assay kit (Invitrogen, Q32851) for quantifying the library. The library was sequenced using the HiSeq 4000 Illumina platform.

## RNA sequencing data processing and gene expression analysis

Paired-end (PE) reads quality checks were performed using the FastQC v.0.11.5 (http://www.bioinformatics.babraham.ac.uk/projects/fastqc/). The adaptor sequence was trimmed using the "bbduk" using BBDuk version 37.58, using "bbduk.sh" script. The files were further processed for alignment using STAR v.2.5.3a with default parameters (Dobin et al, 2013). The mouse genome build GRCm38.p6genome.fa was used from Gencode.vM18.annotation.gtf. Human GRCh38 genome build, gencode v21 gtf (GRCh38) from gencode was used. Duplicates were removed from Picard-2.9.4 (https://broadinstitute.github.io/picard/) from the aligned bam files. Count matrix for each comparison was generated for differential gene analysis, featureCounts v.1.5.3 from subread-1.5.3 package (http://bioinf.wehi.edu.au/featureCounts/) was used with $Q = 10$ for mapping quality, and these count files were used as input for downstream differential gene expression analysis with DESeq2 version 1.14.1 (Love et al, 2014). Genes with read counts of $\leq 10$ in any comparison were removed followed by using DESeq2 "R" library for count transformation and statistical analysis. "P" values were adjusted using the Benjamini and Hochberg multiple testing correction (Haynes, 2013). Significantly differentially expressed genes were identified based on a fold change of 1.5-fold or greater (up- or downregulated) and a P-value < 0.05.

## Clustering and heatmap generation

Unsupervised hierarchical clustering was performed, and the heatmap was plotted using "ComplexHeatmap" library using "R"

Bioconductor package (https://www.bioconductor.org/) where the gene expression matrix was transformed into z-score (Gu et al, 2016).

## Western blotting

Cell and tissue lysate preparation and western blot analysis were performed as described previously (Mehto et al, 2019). Briefly, cell lysates were prepared using the NP-40 lysis buffer (Invitrogen #FNN0021) containing protease inhibitor cocktail (Roche #11836170001), phosStop (Roche #4906845001) and 1 mM PMSF (Sigma #P7626). Mouse tissues (100 mg) were homogenized in 1 ml Radio-immunoprecipitation assay (RIPA) buffer (20 mM Tris pH 8.0; 1 mM EDTA; 0.5 mM EGTA; 0.1% sodium deoxycholate; 150 mM NaCl; 1% IGEPAL; 10% glycerol) with protease inhibitor cocktail, phosStop, and 1 mM PMSF using tissue tearor (BioSpec #985370). Lysates were separated using SDS–polyacrylamide gel, transferred onto a nitrocellulose membrane (Bio-Rad) and blocked for 1 h in 5% skimmed milk followed by incubation in primary antibody overnight at 4°C. Membranes were then washed thrice with 1× PBS/PBST and incubated for 1 h with HRP conjugated secondary antibody. After washing with PBS/PBST, the blots were developed using enhanced chemiluminescence reagent (Thermo Fisher #32132X3).

## Mitochondrial isolation

Mitochondria from control and IRGM knockdown THP-1 cells were isolated using the Qproteome mitochondria isolation kit (QIAGEN #37612) according to the manufacturer's instructions. Briefly, $10 \times 10^6$ cells were lysed in ice-cold lysis buffer (supplemented with protease inhibitor) by centrifuging at 1,000 $g$ at 4°C for 10 min. The cell pellet was resuspended in disruption buffer supplemented with a protease inhibitor. The cells were disrupted entirely, and the lysates were centrifuged at 1,000 $g$ for 10 min at 4°C. The supernatant was transferred and centrifuged at 6,000 $g$ for 10 min at 4°C to form a pellet containing mitochondria. The pellet was washed and resuspended in a mitochondrial storage buffer for further analysis.

## Semi-denaturing agarose gel electrophoresis

For the MAVS oligomerization assay, the crude mitochondrial pellet was resuspended in 4× sample buffer (0.5× TAE, 10% glycerol, 2% SDS, and 0.0025% bromophenol blue) followed by incubation at RT for 5 min and then loaded onto a vertical 1.5% agarose gel. Electrophoresis was performed in the running buffer (1× TAE and 0.1% SDS) for 40 min with a constant voltage of 100 V at 4°C. After electrophoresis, the proteins were transferred onto a PVDF membrane for immunoblotting.

## Antibodies and dilution

Primary antibodies used in western blotting with dilutions: Flag (Sigma #F1804; 1:1,000), c-Myc (Santa Cruz sc40; 1:750 and sc764; 1:1,000), p62 (BD #610832; 1:2,000), Actin (Abcam #ab6276; 1:5,000), HA (CST #3724; 1:1,000), GFP (Abcam #ab290; 1:5,000), IRGM antibody rodent specific (CST #14979; 1:1,000), IRGM

(Abcam #ab69494; 1:500), RIG-I (CST #3743; 1:1,000), TLR3 (CST #6961; 1:1,000), Cgas (CST #15102; 1:1,000, Santa Cruz #sc515777; 1:500), STING (CST #3337, #13647; 1:1,000), MDA5 (Santa Cruz #sc48031; 1:500), TRIF (Abcam #ab13810; 1:5,000), MAVS (CST #3993; 1:1,000, Santa Cruz #sc365334; 1:500), TBK1/NAK (CST #3504; 1:1,000), pTBK1 (CST #5483S; 1:1,000), pIRF7 (CST #12390S; 1:1,000), IRF7 (CST #4920; 1:1,000, Abcam #ab62505; 1:5,000), pIRF3 (CST #29047; 1:1,000), IRF3 (CST #11904; 1:1,000, Abcam #ab25950; 1:5,000) pSTAT1 (CST #7649; 1:1,000), STAT1 (CST #9172; 1:1,000), pSTAT2 (CST #88410; 1:1,000, Abcam #ab53132; 1:5,000) STAT2 (CST #72604; 1:1,000), TLR4 (Santa Cruz #sc293072; 1:500), IRF9 (Abcam #ab51639; 1:5,000), TLR7 (Santa Cruz #sc57463; 1:500), BECLIN1 (CST #3738; 1:1,000), MX1 (CST #37849; 1:1,000), OAS1 (CST #14498; 1:1,000), ISG15 (CST #2743; 1:1,000), AIM2 (CST #12948; 1:1,000), NLRC4 (CST #12421; 1:1,000); GAPDH (CST #2118; 1:1,000), Normal rabbit IgG (CST #2729), LC3B (Sigma #L7543; 1:1,000), TOM20 (Santa Cruz #sc11415; 1:1,000), PINK1 (CST #6946; 1:1,000), PINK1 (Abcam #ab186303; 1:1,000) Parkin (CST #4211; 1:1,000), Cytochrome-C (Santa Cruz # sc7159; 1:500).

HRP conjugated secondary antibodies were purchased from Santa Cruz (1:2,000) or Promega (1:5,000) or Abcam (1:10,000) or Novus (1:5,000).

Primary antibodies used in immunofluorescence assays with dilutions: IRGM (Abcam #69494; 1:100), dsRNA (Kerafast # ES2001; 1:60), TOM20 (Santa Cruz #sc11415; 1:150), RIG-I (CST #3743, 1:50), MDA5 (Santa Cruz #sc48031; 1:50), TIA1 (Abcam #140595; 1:100), dsDNA (Abcam #ab27156; 1:10,000), Lamin A/C (Santa cruz #sc6215; 1:50), MX1 (CST #37849; 1:200), cGAS (CST #15102; 1:50 or Santa Cruz # sc515777; 1:50), PINK1 (Abcam #ab186303; 1:100), Parkin (CST #4211; 1:100), p62 (Santa Cruz # sc28359; 1:100 or CST #5114; 1:100), ubiquitin (clone FK2 MBL #D058-3; 1:500), Cytochrome-C (Santa Cruz # sc7159; 1:100).

## Co-immunoprecipitation

Co-immunoprecipitation assays were performed as described previously (Jena *et al*, 2018). The cells were lysed in NP-40 lysis buffer (Thermo Fisher #FNN0021) supplemented with protease inhibitor/phosphatase inhibitor cocktails and 1 mM PMSF for 30 min at 4°C and subjected to centrifugation. The supernatant was incubated with the desired antibody at 4°C (2 h to overnight) on rotational cell mixer followed by incubation with protein G dynabeads (Invitrogen, #10004D) for 2 h at 4°C. The beads were washed with lysis buffer and ice-cold PBS (3–4 times), and the proteins were eluted by boiling for 5 min in 2× laemmli buffer and proceeded for western blot analysis.

## Immunofluorescence analysis

Approximately $10^5$ cells were seeded on a coverslip and allowed to adhere to the surface. For THP-1, cells were differentiated into the macrophage-like state by the addition of 50 ng/ml of phorbol 12-myristate 13-acetate (PMA; Sigma #P8139) for 16 h. Next, the culture medium was replaced and incubated for 48 h. The adhered cells were fixed in 4% paraformaldehyde for 10 min and permeabilized with 0.1% Triton X-100 for 10 min, followed by blocking with 1% BSA for 30 min at RT. The cells were then incubated with primary antibody for 1 h at RT, washed thrice with 1× PBS, followed by 1 h incubation with Alexa Fluor-conjugated secondary antibody. Cells were again washed thrice with 1× PBS, mounted (Prolong gold antifade, Invitrogen #P36931), and visualized using Leica TCS SP5 and TCS SP8STED confocal microscope. For Mito-Tracker CMXRos (Invitrogen #M7512) staining, cells were stained with the dye (10 nM, 30 min) as per the manufacturer's instructions, washed, fixed with 4% PFA for 10 min, and visualized using Leica TCS SP5 confocal microscope.

## Luciferase assay

Luciferase assay was performed using a dual luciferase assay kit as per the manufacturer's protocol (Promega #E1910). Briefly, HEK293T cells were cultured in 24-well plates and transfected with pISRE-luciferase reporter plasmid (100 ng) and renilla luciferase plasmid (50 ng) along with required plasmids (300 ng). After 36 h, growth medium was removed entirely, and after washing thrice with 1× PBS, cells were lysed using 100 μl 1× passive lysis buffer by incubating the plate on an orbital shaker for 15 min at RT. Cell lysates were cleared by centrifugation for 30 s at 12,000 *g* in a refrigerated centrifuge. In 96-well plate, 20 μl cell lysate mixed with 100 μl LARII reagents and luminescence was measured using Perkin-nElmer VICTOR Nivo multimode plate reader. Stop & Glo reagent was used to measure renilla luciferase activity.

## RNA isolation and quantitative real-time PCR

RNA was extracted using TRIzol by following the manufacturer's protocol. The cDNA was synthesized using the high capacity DNA reverse transcription kit (Applied Biosystems, #4368813) and qRT–PCR was performed using TaqMan master mix (Applied Biosystems, #4369016) or Power SYBR green PCR master mix (Applied Biosystems, #4367659) according to manufacturer's protocol. For normalization of the assay, the housekeeping gene GAPDH or β-Actin was used. The fold change in expression was calculated by the $2^{-\Delta\Delta C_t}$ method.

## Flow cytometry

Mitochondrial membrane potential/polarization experiment was performed in live cells. The cells were incubated with CMXRos dye (Invitrogen #M7512; 10 nM, 30 min) or JC-1 (eBiosciences #65085138; 2 μM, 30 min) to measure the change in mitochondrial potential. Mitochondrial superoxide generation was measured using MitoSox (Invitrogen #M36008) staining (1 μM, 20 min). The cellular ROS levels were measured using CellRox (Invitrogen #C10448) staining (1 μM, 30 min). Briefly, $10^5$ cells were stained with the mentioned dyes and incubated for indicated time points at 37°C and 5% $CO_2$. Samples were washed twice with 1× PBS and resuspended in 1× PBS, followed by acquisition on FACS Calibur (Beckton and Dickinson). The results were analyzed using the Cell Quest Pro software.

## Mouse bone marrow cell isolation and differentiation into macrophages

*Irgm1* knockout (C57BL/6) mice were described previously (Liu *et al*, 2013).The mouse experiments were performed with

procedures approved by the institutional animal ethical committee at the Institute of Life Science, Bhubaneswar, India. For each experiment, littermates were used and age of mice were matched. The bone marrow cells from wild-type ($Irgm1^{+/+}$) and knockout ($Irgm1^{-/-}$) mice were isolated and differentiated into macrophages by standard procedure. Briefly, 6- to 8-week old male C57BL/6 $Irgm1^{+/+}$ and $Irgm1^{-/-}$ mice were sacrificed by cervical dislocation, bone marrow cells from the tibia and femurs were flushed out in RPMI medium. Red blood cells were removed by cell lysis buffer containing (155 mM $NH_4Cl$, 12 mM $NaHCO_3$, and 0.1 mM EDTA). Bone marrow cells were differentiated in RPMI medium (10% FBS, 1 mM sodium pyruvate and 0.05 M 2-mercaptoethanol) containing 20 ng/ml mouse M-CSF (Gibco #PMC2044) for 5 days. On every alternate day, media were replaced with fresh media containing M-CSF.

### Seahorse Mito stress assay

Bone marrow-derived macrophages from $Irgm1^{+/+}$ and $Irgm1^{-/-}$ mice were seeded in XFp mini cell culture plate ($5 \times 10^5$ cells/well) in RPMI supplemented with 10% FBS and incubated at 37°C in $CO_2$ incubator for overnight. The sensor cartridge was hydrated with Agilent Seahorse XF Calibrant in non-$CO_2$ incubator at 37°C for overnight. Before starting the assay, cells were gently washed twice with RPMI assay medium and incubated in 180 µl/well RPMI assay medium supplemented with 10 mM glucose, 1 mM pyruvate, and 2 mM glutamine having pH = 7.4 into non-$CO_2$ incubator for 1 h at 37°C. The hydrated sensor cartridge was loaded with oligomycin (4.5 µM), FCCP (2 µM), and rotenone (2 µM) solution in the respective port. The real-time OCR (pMoles $O_2$/min) was measured using Seahorse XFp Analyzer.

### Isolation of mitochondrial DNA from cytosolic fraction

Mitochondrial DNA was isolated from cytosolic fraction using the Qproteome mitochondria isolation kit (Qiagen, Cat No # 37612) with minor modifications. Briefly, $5 \times 10^6$ BMDMs from $Irgm1^{+/+}$ and $Irgm1^{-/-}$ mice were seeded in 10 cm dish. Next day, cells were harvested and washed twice with ice-cold 1× PBS and once with 0.9% sodium chloride solution at 500 $g$ for 5 min at 4°C. The cell pellet was resuspended in 500 µl lysis buffer and homogenized using KIMBLE Dounce tissue grinder (Sigma, Cat. No # D8938) with 8–10 strokes on ice. The cell suspension was centrifuged at 700 $g$ for 10 min at 4°C and again at 7,000 $g$ for 10 min at 4°C, and the supernatant was collected. The total DNA was isolated from the supernatant (cytosolic fraction) by using the QIAquick nucleotide removal kit (Qiagen, Cat No. 28304) and eluted in 25 µl elution buffer. qRT–PCR was performed using cytochrome $c$ oxidase subunit 1 primer. For normalization of the assay, 18S RNA primer was used.

### Generation of HT-29 Rho⁰ cells

For generation of Rho⁰ cells, HT-29 cells were chronically exposed 50 ng/ml ethidium bromide in DMEM supplemented with 10% FBS, 2 mM L-glutamine, 100 µg/ml pyruvate, and 50 mg/ml uridine for one month. The medium was changed every 2 days, and subculture was performed after the cells reached 80–90% confluency. Cells were subcloned in 96-well plates by limiting dilution. Cloned cells were maintained in the culture medium supplemented with pyruvate and uridine and ethidium bromide and confirmed through qRT–PCR with mitochondrial cytochrome-c oxygenase gene. The loss of mtDNA was confirmed by qRT–PCR.

### GST pulldown assay

GST pulldown assay was done as described previously (Mehto et al, 2019). GST-IRGM recombinant protein was expressed in SoluBL21 (Amsbio) and purified on Glutathione Sepharose 4 Fast-Flow beads (GE Healthcare). [$^{35}$S]-labeled Myc-cGAS, Myc-RIG-I, and Myc-TLR3 were cotranscribed/translated using the TnT T7-coupled reticulocyte lysate system (Promega). The in vitro-translated [$^{35}$S]-labeled proteins were then incubated with GST or GST-IRGM in 250 µl of NETN-E buffer (50 mM Tris pH 8.0, 100 mM NaCl, 6 mM EDTA, 6 mM EGTA, 0.5% NP-40, and 1 mM dithiothreitol supplemented with Complete mini EDTA-free protease inhibitor cocktail [Roche]) for 2 h at 4°C and then washed five times with 1 ml of NETN-E buffer, boiled with loading buffer, and subjected to SDS–PAGE. The gel was stained with Coomassie Blue and vacuum-dried. The GST-IRGM was detected by staining with Coomassie Blue, whereas the [$^{35}$S]-labeled PRRs were detected in PharosFX imager (Bio-Rad Laboratories).

## Data availability

The RNA-seq datasets produced in this study have been deposited in the ArrayExpress database at EMBL-EBI (www.ebi.ac.uk/arrayexpress) under accession number E-MTAB-9164 (http://www.ebi.vac.uk/arrayexpress/experiments/E-MTAB-9164) and E-MTAB-9142 (http://www.ebi.ac.uk/arrayexpress/experiments/E-MTAB-9142).

Expanded View for this article is available online.

### Acknowledgements

This work is funded by the DBT/Wellcome Trust India Alliance (IA/I/15/2/502071) fellowship to Santosh Chauhan. Punit Prasad is supported by Rama-lingaswami Re-entry fellowship of the Department of Biotechnology (BT/HRD/35/02/2006). Tor Erik Rusten and Ashish Jain are supported by grants #262652 and #276070 from the Norwegian Research Council. Subhash Mehto is financially supported by DBT-RA Program in Biotechnology & Life Sciences. We acknowledge the technical assistance of Kshitish Rout and Paritosh Nath (FACS facility). We gratefully acknowledge the support of the Institute of Life Sciences central facilities funded by Department of Biotechnology (India).

### Author contributions

SC secured funding, conceived the project, designed the experiments, and wrote the manuscript. KKJ, SM, PN, and NRC performed the majority of experiments. RS KD, SKD, SPK, KCM, AJ, SK, BSS, performed the experiments. SwC, PP, TER and SoC provided critical inputs, and guidance for experiments and also edited the manuscript.

### Conflict of interest

The authors declare that they have no conflict of interest.

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
