## [Review Process File · EMBO Reports]

Autoimmunity Gene IRGM Suppresses cGAS-STING and RIG-I-MAVS Signaling to control Interferon Response

Kautilya Kumar Jena, Subhash Mehto, Parej Nath, Nishant Ranjan Chauhan, Rinku Sahu, Kollori Dhar, Saroj Kumar Das, Srinivasa Prasad Kolapalli, Krushna C Murmu, Ashish Jain, Sivaram Krishna, Bhabani Sankar Sahoo, Soma Chattopadhyay, Tor Erik Rusten, Punit Prasad, Swati Chauhan, and Santosh Chauhan

DOI: [10.15252/embr.202050051](https://doi.org/10.15252/embr.202050051)

Corresponding author(s): [Santosh Chauhan \(schauhan@ils.res.in\)](mailto:schauhan@ils.res.in)

Review Timeline:

Submission Date:	17th Jan 20
Editorial Decision:	11th Feb 20
Revision Received:	20th May 20
Editorial Decision:	16th Jun 20
Revision Received:	22nd Jun 20
Editorial Decision:	26th Jun 20
Revision Received:	27th Jun 20
Accepted:	2nd Jul 20

Editor: *Achim Breiling*

Transaction Report:

Dear Dr. Chauhan,

Thank you for the submission of your research manuscript to EMBO reports. We have now received reports from the three referees that were asked to evaluate your study, which can be found at the end of this email.

As you will see, all referees think that the findings are of interest, but they also have several comments, concerns and suggestions, indicating that a major revision of the manuscript is necessary to allow publication in EMBO reports. As the reports are below, and I think all points need to be addressed, I will not detail them here.

Given the constructive referee comments, we would like to invite you to revise your manuscript with the understanding that all referee concerns must be addressed in the revised manuscript and/or in a detailed point-by-point response. Acceptance of your manuscript will depend on a positive outcome of a second round of review. It is EMBO reports policy to allow a single round of revision only and acceptance of the manuscript will therefore depend on the completeness of your responses included in the next, final version of the manuscript.

Revised manuscripts should be submitted within three months of a request for revision; they will otherwise be treated as new submissions. Please contact me if a 3-months time frame is not sufficient so that we can discuss the revisions further.

PLEASE NOTE that upon resubmission revised manuscripts are subjected to an initial quality control prior to exposition to re-review. Upon failure in the initial quality control, the manuscripts are sent back to the authors, which may lead to delays. A frequent reasons for such a failure is the presence of statistics based on $n=2$ (the authors are then asked to present scatter plots or provide more data points).

- 1) a .docx formatted version of the final manuscript text (including legends for main figures, EV figures and tables), but without the figures included. Please make sure that the changes are highlighted to be clearly visible. Figure legends should be compiled at the end of the manuscript text.
- 2) individual production quality figure files as .eps, .tif, .jpg (one file per figure), of main figures and EV figures. Please upload these as separate, individual files upon re-submission. Please provide their legends in a specific section called 'Figure Legends' at the end of the main manuscript text file.

The Expanded View format, which will be displayed in the main HTML of the paper in a collapsible format, has replaced the Supplementary information. You can submit up to 5 images as Expanded View. Please follow the nomenclature Figure EV1, Figure EV2 etc. The figure legend for these should be included in the main manuscript document file in a section called Expanded View Figure Legends after the main Figure Legends section. Additional Supplementary material should be supplied as a single pdf labeled Appendix. The Appendix should have page numbers and needs to

include a table of content on the first page (with page numbers) and legends for all content. Please follow the nomenclature Appendix Figure Sx, Appendix Table Sx etc. throughout the text, and also label the figures and tables according to this nomenclature.

For more details please refer to our guide to authors:

See also our guide for figure preparation:

http://wol-prod-cdn.literatemonline.com/pb-assets/embosite/EMBOPress_Figure_Guidelines_061115-1561436025777.pdf

4) a complete author checklist, which you can download from our author guidelines (<https://www.embopress.org/page/journal/14693178/authorguide>). Please insert page numbers in the checklist to indicate where the requested information can be found in the manuscript. The completed author checklist will also be part of the RPF.

Please also follow our guidelines for the use of living organisms, and the respective reporting guidelines: <http://www.embopress.org/page/journal/14693178/authorguide#livingorganisms>

5) We strongly encourage the publication of original source data with the aim of making primary data more accessible and transparent to the reader (see also comment 2 of referee #3). The source data will be published in a separate source data file online along with the accepted manuscript and will be linked to the relevant figure. If you would like to use this opportunity, please submit the source data (for example scans of entire gels or blots, data points of graphs in an excel sheet, additional images, etc.) of your key experiments together with the revised manuscript. If you want to provide source data, please include size markers for scans of entire gels, label the scans with figure and panel number, and send one PDF file per figure.

6) Our journal encourages inclusion of *data citations in the reference list* to directly cite datasets that were re-used and obtained from public databases. Data citations in the article text are distinct from normal bibliographical citations and should directly link to the database records from which the data can be accessed. In the main text, data citations are formatted as follows: "Data ref: Smith et al, 2001" or "Data ref: NCBI Sequence Read Archive PRJNA342805, 2017". In the Reference list, data citations must be labeled with "[DATASET]". A data reference must provide the database name, accession number/identifiers and a resolvable link to the landing page from which the data can be accessed at the end of the reference. Further instructions are available at: <http://www.embopress.org/page/journal/14693178/authorguide#referencesformat>

7) Regarding data quantification and statistics, can you please specify, where applicable, the number "n" for how many independent experiments (biological replicates) were performed, the bars and error bars (e.g. SEM, SD) and the test used to calculate p-values in the respective figure legends. Please provide statistical testing where applicable, and also add a paragraph detailing this to the methods section. See:

<http://www.embopress.org/page/journal/14693178/authorguide#statisticalanalysis>

8) Please format the references according to our journal style. See:

9) Please add up to 5 key words to the title page of the manuscript.

I look forward to seeing a revised version of your manuscript when it is ready. Please let me know if you have questions or comments regarding the revision.

Yours sincerely,

Achim Breiling
Editor
EMBO Reports

Referee #1:

IRGM is a GTPase genetically and functionally connected with numerous autoimmune diseases. The mechanism of action of IRGM remains unknown. The authors of this study reveal that IRGM is a key negative regulator of the interferon response. In particular, they show that IRGM interacts with proteins like cGAS and RIG-I, and mediates their p62-dependent autophagic turnover to restrain interferon signaling, and this leads to the high expression of interferon-stimulated genes. Moreover, IRGM deficiency results in defective mitophagy leading to accumulation of damaged mitochondria that releases DNA, dsRNA and ROS. In turn, these factors activate signaling axis leading that enhance even further the interferon response. Altogether, those data assign an important function to IRGM and may explain some severe pathologies caused by defects in IRGM.

This is a very well executed study, and although a very large number of experiments has been performed, the manuscript is easy to follow because well written.

I have just a series of comments aimed at strengthening the conclusions of the authors.

Most relevant points:

- Page 9, lines 17-30. In addition to macroautophagy, bafilomycin also inhibits the last steps of endocytosis, microautophagy and endosomal microautophagy. This latter pathway also delivers autophagy receptors into the lysosomal lumen (Mejlvang et al, 2018, J Cell Biol, 217, 3640-3655). As a result, the terms autophagy/autophagosomal/autophagic degradation in this paragraph have to be changed into lysosome/lysosomal/lysosomal degradation. The authors, however, can state that their results could suggest that the examined factors are turned over by autophagy.

- The experiments with BECLIN1 are aimed at demonstrating that the lysosomal turnover of the examined factors is due to autophagy. In addition to be part of the ATG14L-containing VPS34 complex, which is involved in autophagy, BECLIN1 is also part of the endolysosomal VPS34 complexes containing UVRAG and UVRAG plus RUBICON. As a result, depletion of BECLIN1 is also affecting endosomal functions, possibly also the endosomal microautophagy and consequently another pathway delivering autophagy receptors into the lysosomes. Consequently, the authors cannot exclude that IRGM restrains interferon signaling in a p62-dependent manner through

endosomal microautophagy or other endolysosomal pathways. The authors have to repeat some of the key experiments performed with BECLIN1 by depleting a different ATG protein.

- The model of the authors is that IRGM restrains interferon signaling in a p62-dependent manner. Knockdown/knockout of p62 must therefore phenocopy at least in part IRGM depletion. This has to be tested. As IRGM also binds TAXBP1 and it cannot be excluded that it also interacts with the other autophagy receptors (the authors did not detect an association between IRGM and them, but this could be due to the pull-down conditions), the authors may consider to knockdown the 5 autophagy receptors simultaneously.

Minor:

- As indicated above, there are different types of autophagy and the authors use the term autophagy throughout the text, without defining precisely. Therefore, I would suggest to use the term macroautophagy in state of autophagy in the entire text.

- The authors looked at AIM2. Why did they not also follow NLRP3, another subunit of the inflammasome, which they previously shown that is interacting with p62 via IRGM (Mehto et al, 2018, Mol Cell, 73, 429-445)? It would have served as a positive control.

- The authors examined the turnover of TRIF, which appears to be mediated by TAXBP1 and NDP52, but not that of TRIF6, which has been shown to depend on NDP52 and p62 (Chan et al, 2016, J Virol, 90, 10928-10935; Inomata et al, Cell Mol Life Sci, 69, 963-79). They could check the mRNA and protein levels of this protein. Similarly, the NF κ B pathway components IKK β and IKK γ are also degraded by autophagy in a p62-dependent manner (Liu et al, 2018, J Mol Cell Biol, 10, 205-215; Zhang et al, 2017, Nat Commun 8, 2164). The authors could look at the role of IRGM in the lysosomal turnover of these two components as well, by also looking at the mRNA and protein levels. An accumulation of TRIF6, IKK β and IKK γ in cells depleted of IRGM will further strengthen their main conclusion and model.

- In the absence of IRGM, PARKIN is recruited to mitochondria. This implies a higher ubiquitination of this organelle and possibly a recruitment (or not) of p62 and the other autophagy receptors. It would be interesting to check which kind of influence IRGM has on this event.

- It has been shown that NDP52 is involved in the turnover of NLRP3, TRIF, TRAF6 and MAVS. In the discussion, the authors must comment whether that are other regulator than IRGM regulating this autophagy receptor, or it may be that it is also regulated by IRGM but they failed to detect an interaction in the used conditions.

Referee #2:

In this study by Jena et al. the authors have analyzed the role IRGM plays in regulating the innate immune system for recognition of nucleic acids, in particular. They propose that through promoting autophagy IRGM acts to suppress the interferon system. This is proposed to be through both autophagic clearance of PRRs such as cGAS, TLR3 or MDA5 as well as through promoting mitophagy. Thus loss of IRGM leads to enhanced production of DAMPs due to defective mitophagy, and subsequent hyperactivation of the PRRs leading to overproduction of interferon and upregulation of ISGs.

This study provides a very nice model for how loss of IRGM could lead to spontaneous inflammation, there are however a few points that could be further clarified.

1) Role of mtDNA.

The authors propose that due to defective mitochondrial homeostasis in IRGM deficient cells, mtDNA is released into the cytosol and activates the cGAS/STING system. They also identify micronuclei as a source of cGAS/STING activation. The relative contribution to the effect has however not really been shown. While the microscopy images using dsDNA antibodies certainly show more DNA in the IRGM deficient cells, it is not completely convincing that it is cytosolic in regards to the proposed mtDNA. Indeed in Figure S6 it almost seems as if the dsDNA is outside of the cell. In Figure 8a, using only Tom20 as a marker for mitochondria under these circumstances could be somewhat limiting and some marker on the inside of the mitochondria would be advisable to use to avoid possible complications due to parkin mediated degradation of Tom20 which has been shown previously.

The fact that there are more and seemingly larger structures for mtDNA does not mean they are cytosolic or that this is what is driving the effect. The authors could make use of RhoO cells (cells lacking mtDNA) to confirm this. This can be achieved rapidly with mitochondrial targeted nucleases or the traditional method of ethidium bromide in the culture. These cells could then be examined for IFN responses as in the manuscript to confirm if mtDNA or micronucleoids are playing a significant role. Additionally, immunofluorescence could be performed on the cells with DNASE1 transfection to determine if indeed the dsDNA signal is reduced.

It is also unclear from the DNASE treatment how the samples were prepared. It seems that total cells were taken, but this of course would take mitochondrial content too, and thus the increased signal could simply be due to increased mitochondrial mass, not increased cytosolic DNA. This would fit with the images, which seem to suggest more mitochondrial mass in the IRGM deficient cells. If some other technique was used to only isolate cytosol, then this needs to be described in more detail as it is not clear and verified by also performing PCR for mitochondrial DNA as in supp figure 6 but also +/- DNase transfection to confirm that the signal for mitochondrial DNA is reduced.

2. Role for IRGM interaction with receptors?

While the authors provide strong support for the interaction of IRGM with the various PRRs, but it is not clear to what extent this mechanism is at play in relation to the mitophagy deficiency. It is not clear from the discussion or the results to what extent the authors believe that the degradation of the receptors themselves as shown from figures 2-5 is important or the production of the DAMPS from mitochondria. I realise this is not easy to distinguish, but this could be better covered in the discussion at the least and could be experimentally addressed to some extent as follows:

The contribution of mitophagy failure to the inflammation could be further addressed and its role further differentiated from that of degradation of the receptors by looking in parkin deficient cells to show that loss of mitophagy specifically does or does not lead to the increased interferon responses. This could be coupled with dsDNA/RNA staining as already done to show the loss of mitophagy still results in these effects, but the degradation of the receptors should be more or less unchallenged.

Do the authors have a suggestion as to why interaction of IRGM with the receptors is needed? They have previously reported that IRGM regulates autophagy induction through binding to beclin1 complexes to promote their activation? It is not clear then why interaction directly with the receptors would be needed, or how this is happening? Is it through ubiquitylation of the receptors

after their activation? It would be informative to determine if the receptors are also ubiquitylated. Additionally, is there a role for p62 in the regulation of mitophagy in your system or is it simply the degradation of the receptors? It would be informative to also look at mitophagy in p62^{-/-} cells in your systems to answer this. In addition it would be interesting to see if, in the absence of p62, does IRGM still interact with the PRRs? P62 is known to mediate degradation of STING and TRIF and its phosphorylation via TBK1 regulates recruitment LC3 to the complexes. Perhaps IRGM is also recruited through p62 mediated binding to ubiquitylated PRRs in a similar fashion to help recruit the phagophore machinery?

3. Minor points:

Figure 5, have the authors also looked in beclin1 deficient cells without IRGM overexpression? Presumably if the phenotype is only due to autophagic degradation of receptors and mitophagy, then the beclin1 deficient cells should look the same as IRGM^{-/-} cells in terms of ISG upregulation.

Figure 6I-J. It is curious given the role of IRGM in regulating autophagy as well as targeted autophagy/mitophagy, that no increase in LC3-II is seen in the IRGM^{-/-} samples. Can the authors explain this discrepancy? Additionally I am not so sure that simply by showing no increase in parkin or pink1 in the presence of bafilomycin A that one can argue that mitophagic flux is blocked given their tight regulation through PTMs and stability/localisation etc.. Other mitochondrial proteins, preferentially ones inside the mitochondria would be useful to further support this claim.

Figure 6K, there should be labels about drug additions to the Seahorse data.

Overall this is a very interesting study and will be of great interest to EMBO Reports readers.

Referee #3:

IRGM has been shown to connect with several autoimmune diseases. This manuscript tried to provide mechanistic understanding for IRGM in mediating IFN signaling as well as autoimmunity. However, the phenomena in the manuscript were inconsistent with previous studies and should be further confirmed, and several results in this manuscript were not consistent. In addition, the figures are not well-organized and made, and do not fit the style of EMBO Reports to my knowledge.

1. As the author indicated, IRGM was identified as an autoimmune diseases-related gene and plenty of studies from different groups have tried to elucidate the relationship between IRGM and autoimmunity. The author should first fully discuss former works about IRGM and provide basic understanding about IRGM. For example, in the recent article entitled "The Crohn's Disease Risk Factor IRGM Limits NLRP3 Inflammasome Activation by Impeding Its Assembly and by Mediating Its Selective Autophagy" from Molecular Cell indicated that knockout or knockdown of IRGM/Irgm1 could enhance the pro-inflammatory responses and IRGM/Irgm1 protects from pyroptosis and gut inflammation in a Crohn's disease mouse model by targeting NLRP3. However, in Figure 1 of this study, the author showed that IRGM/Irgm1 deficiency led to strong induction of IFN signaling in the steady-state conditions and suggested the correlation between IFN induction and autoimmunity. Compared with the induction of IFN signaling in IRGM/Irgm1 deficiency cells, what happens to other signaling such as pro-inflammatory response and NLRP3 inflammation. And the author might better analyze different pathways that may induce autoimmune responses in IRGM/Irgm1 deficiency cells and figure out which signals contribute dominantly to the autoimmunity.

2. One serious concern is that plenty of the western blots (including the loading controls) appear to be overexposed/saturated. It made it hard to tell the differences between different lanes. Since this data in Figure 2 to Figure 6 largely relies on quantification of western bands, such mistake would have a profound impact on the validity of the conclusions. The authors should use the low exposed images or reduce the loading quantity of the samples. In addition, the authors should show how many times they repeat such experiments, and show these repeats in the supplements or the response letter.

3. In Figure 2, the author showed the enhanced protein levels of different sensors in IFN signaling pathway in IRGM deficiency cells. The authors should point out the reasons for showing multiple β -Actin in one figure such as Figure 2C, 2D, 2G, 2N. I supposed the authors showed different samples they collected different times. And the loading controls from the samples collected from different times showed obviously inconsistent. For example, in Figure 2D, the Actin in the sixth line together with Irgm1 showed a sustained decline while the two Actin in the fourth and ninth lines remained unchanged. In Figure 2N the last two lanes of Actin in line three were much weaker while the Actin in the line 6 showed similar. The authors should use one group of samples collected one time to run a set of western blots instead of using different samples collected from different times to piece together one figure. And the author casually put the repeated results in one figure such as Figure 2E, 2H, 2M, 3G, 3H and 3I, and some of these repeated results showed inconsistent. For example, in Figure 2M, the levels of p-STAT1 in line 3 and line 4 were much higher than the p-STAT1 levels in line 1 and line 2, and more obvious differences were also showed in p-STAT2 and IRF9 in the same figure. These inconsistencies above largely reduce the credibility and reliable of this manuscript.

4. The authors claimed that IRGM/Irgm1 deficiency led to enhancement of transcription levels of multiple sensors in IFN signaling pathway, which suggested overexpressing IRGM/Irgm1 might reduce the transcription levels of these sensors, such RIG-I, cGAS and TLR3 since overexpression of IRGM led to reduction of mRNA levels of MX2 and ISG15 in Figure 3J and 3K. In Figure 3E, the authors should first show whether overexpression of IRGM/Irgm1 affect the mRNA levels of RIG-I, cGAS and TLR3. Secondly, the authors should rule out the possibility that the reduction of the endogenous protein levels of these sensors were due to their reduced transcription levels. Moreover, in Figure 3E, overexpression of IRGM led to significant reduction of the protein level of RIG-I, cGAS and TLR3 even though the protein level of Flag IRGM was not very strong. However, in Figure 3F, overexpression of IRGM failed to significantly reduce the level of cGAS and TLR3 even though the expression level of Flag-IRGM showed much stronger than that in Figure 3E. The author should explain this.

5. IRGM is an autophagy-related gene as the author indicated and lots of previous showed IRGM could affect the autophagy flux. However, in Figure 4A, 4B and 4C, the author showed that overexpression of IRGM failed to enhance the switch of LC3/II without autophagic inhibitors. These results were inconsistent with the previous study entitled "Human IRGM Induces Autophagy to Eliminate Intracellular Mycobacteria" of Science in 2006. In Figure 2A and 2B in that paper, they showed overexpression of Irgm1 could significantly promote the switch of LC3/II. The author should explain these inconsistencies. Moreover, as IRGM could enhance strongly the autophagy flux, IRGM may indirectly promote the autophagic degradation of RIG-I and TLR3 by manipulating autophagy flux. In that case, using autophagy inhibitors could also restrict the IRGM-mediated reduced protein levels of RIG-I and TLR3 as well as the reduced induction of IFN signaling. The authors should provide more evidence to prove IRGM could directly mediate the autophagic degradation of RIG-I and TLR-3.

6. In Figure 6, the author showed overexpression of knockdown of Beclin-1 or p62 could rescue the reduction of IFN signaling mediated by IRGM. However, these assays lacked an important control in which the authors should show only knockdown of Beclin-1 or p62 deficiency might enhance IFN signaling without overexpression of IRGM. Previous studies indicated that Atg5 deficiency could also enhance the induction of IFN signaling and reduce VSV infection. The enhancement of IFN signaling might only due to knockdown of Beclin-1 or p62, but not the association between Beclin-1 or p62 with IRGM. That might be the reason why mRNA level of MX2 in p62 knockdown group was higher than the control group in Figure 6I. In Figure 6F and 6G, the authors only showed IRGM could associate with p62 and TAX1BP1. It was insufficient to prove IRGM could mediate p62-dependent selective autophagy. The authors should provide more evidence. In Figure 5H, as IRGM could promote autophagy flux and mediate p62-dependent degradation of RIG-I, the author should explain why the protein levels of p62 remain unchanged after overexpression of IRGM.

7. A previous studies entitled "Human IRGM regulates autophagy and cell-autonomous immunity functions through mitochondria" of Nature Cell Biology in 2010 showed that human IRGM was a mitochondrial-located protein that induces mitochondrial depolarization and promotes mitochondrial fission, both of which were triggers for mitophagy. The author should tell the differences between that work and the results in Figure6 and Figure7, and explain the necessity to repeat these results in two large figures.

8. The authors showed that IRGM deficiency led to severe release of mtDNA in cytosol in Figure8A and 8B, which raises the concern whether the release of mtDNA and constitutive oxidative stress might induce cell death. However, in Figure 8C and 8E, the release of mtDNA did not show significant enhancement in the cytosol, and the decrease of mtDNA and cGAS in Figure 8D and 8F was due to the reduction of protein level of cGAS in the cytosol. The author should explain these inconsistencies. In Figure 8G, the authors electroporated DNase I enzyme in the cells to determine the role of cytosolic DNA in enhancing IFN signaling. That may not be a very assay as the transfection levels as well as the enzyme activity of DNase I were hard to be detected. The authors might have better use mitophagy inhibitors, such as Mdivi 1. In Figure 8J, 8K and 8L, why did the authors only use the IRGM deficiency cells but not show the control cells?

9. Figure 9 only showed that IRGM deficiency could promote IFN signaling at RIG-I and cGAS levels and failed to broaden the understanding of IRGM. The authors may consider putting these results in the supplementary section.

(EMBOR-2020-50051V1)

Manuscript title: "Autoimmunity Gene IRGM Suppresses cGAS-STING and RIG-I-MAVS Signaling to control Interferon Response"

Referee #1:

IRGM is a GTPase genetically and functionally connected with numerous autoimmune diseases. The mechanism of action of IRGM remains unknown. The authors of this study reveal that IRGM is a key negative regulator of the interferon response. In particular, they show that IRGM interacts with proteins like cGAS and RIG-I, and mediates their p62-dependent autophagic turnover to restrain interferon signaling, and this leads to the high expression of interferon-stimulated genes. Moreover, IRGM deficiency results in defective mitophagy leading to accumulation of damaged mitochondria that releases DNA, dsRNA and ROS. In turn, these factors activate signaling axis leading that enhance even further the interferon response. Altogether, those data assign an important function to IRGM and may explain some severe pathologies caused by defects in IRGM.

This is a very well executed study, and although a very large number of experiments has been performed, the manuscript is easy to follow because well written.

Response: We are very thankful to the reviewer for reading our manuscript thoroughly and for the appreciation.

We have now further improved our manuscript by carefully addressing the constructive comments of the reviewer.

I have just a series of comments aimed at strengthening the conclusions of the authors.

More relevant points:

- Page 9, lines 17-30. In addition to macroautophagy, bafilomycin also inhibits the last steps of endocytosis, microautophagy and endosomal microautophagy. This latter pathway also delivers autophagy receptors into the lysosomal lumen (Mejlvang et al, 2018, J Cell Biol, 217, 3640-3655). As a result, the terms autophagy/autophagosomal/autophagic degradation in this paragraph have to be changed into lysosome/lysosomal/lysosomal degradation. The authors, however, can state that their results could suggest that the examined factors are turned over by autophagy.

Response: We agree with the reviewer, and we have corrected the terms (highlighted yellow in the text) page number 9, Paragraph 3 Thanks for this insightful comment.

- The experiments with BECLIN1 are aimed at demonstrating that the lysosomal turnover of the examined factors is due to autophagy. In addition to be part of the ATG14L-containing VPS34 complex, which is involved in autophagy, BECLIN1 is also part of the endoly-

sosomal VPS34 complexes containing UVRAG and UVRAG plus RUBICON. As a result, depletion of BECLIN1 is also affecting endosomal functions, possibly also the endosomal microautophagy and consequently another pathway delivering autophagy receptors into the lysosomes. Consequently, the authors cannot exclude that IRGM restrains interferon signaling in a p62-dependent manner through endosomal microautophagy or other endolysosomal pathways. The authors have to repeat some of the key experiments performed with BECLIN1 by depleting a different ATG protein.

Response: We agree with the reviewer. To rule out the possibility of involvement of microautophagy and other endolysosomal pathways, as suggested by reviewer, we have repeated the key experiments by depleting ATG5. The depletion of ATG5 clearly rescued IRGM mediated degradation of RIG-I, cGAS, and TLR3 (Figure 1, Manuscript Figure 4B). Moreover, like BECLIN1, the depletion of ATG5 clearly rescued IRGM mediated suppression of ISG's (MX2 and ISG15) (Figure 2, Manuscript Figure 4E and F). Taken together, the data suggest that IRGM utilizes BECLIN-1 and ATG5 dependent, p62-mediated macroautophagy for degradation of PRR's and restraining the interferon response. Thanks to reviewer for this wonderful suggestion!

The model of the authors is that IRGM restrains interferon signaling in a p62-dependent manner. Knockdown/knockout of p62 must therefore phenocopy at least in part IRGM depletion. This has to be tested. As IRGM also binds TAXBP1 and it cannot be excluded that it also interacts with the other autophagy receptors (the authors did not detect an association between IRGM and them, but this could be due to the pull-down conditions), the authors may consider to knockdown the 5-autophagy receptors simultaneously.

Response: We appreciate the interesting viewpoint, and we have put all the possible efforts to answer the question, which resulted in a large set of experiments below.

1. As suggested by the reviewer, we knocked down the p62 and determined whether it's knocked down phenocopies the IRGM knockdown in terms of IFN response. We found depletion of p62, increased expression of several ISG's significantly (MX2, ISG15, IFN-β, OAS1), as was seen in the case of IRGM (Figure 3, Manuscript Figure EV3J). However, as predicted by

the reviewer, the response was not as strong as in case of IRGM (Figure 3, Manuscript Figure EV3J) suggesting that it is more than p62-mediated autophagy of PRR's (mitophagy defect, as shown in the second part of the manuscript) that contributes to total IFN response in IRGM depleted cells.

2. In the older version of the manuscript, we found that IRGM interacts strongly with p62 and weakly with TAX1BP1, and no interaction was detected with NBR2, NDP52, and Optineurin. In very much line with this data, we found that depleting p62 almost completely rescued the IRGM-mediated suppression of interferon response. Here, now we tested whether depleting NBR1, NDP52, or TAX1BP1 rescues the IRGM-mediated suppression of IFN response. None of the adaptor protein was able to rescue the IRGM-mediated suppression of ISG's (Figure 4, Manuscript Appendix Supplementary Figure 2I and J). Taken together, the data suggest that p62, which is a major interactor of IRGM, is the main receptor utilized by IRGM for the degradation of nucleic acid-sensing PRR's.

3. To further understand the mechanism of IRGM-p62 mediated PRR autophagy, we first tested whether p62 is required for IRGM and PRR interaction by depleting p62 and performing co-immunoprecipitation assays. We found that p62 knockdown dramatically reduced the interaction

of IRGM with cGAS, RIG-I, and TLR3 suggesting that p62 is required for IRGM and PRR's interactions (Figure 5, Manuscript Figure 4K-M).

4. We previously found that IRGM can act as a scaffold protein for increasing interaction between NOD2/NOD1 and autophagy proteins (Chauhan et al, 2015). We tested here whether

IRGM can potentiate interaction between cGAS/RIG-I/TLR3 and p62. Indeed, we found that the interaction between p62 and PRRs are increased in the presence of IRGM in co-immunoprecipitation assays (Figure 6, Manuscript Figure 4P-R). Taken together, the data suggest that p62 and IRGM, along with PRR's form a ternary complex where both p62 and IRGM cooperatively increases each other interactions with PRR's for autophagic degradation of PRR's.

5. Several publications suggest that p62 interacts with RIG-I and cGAS and is the adaptor of the protein for autophagic degradation of RIG-I and cGAS to control IFN response (Chen et al, 2016; Du et al, 2018; Prabakaran et al, 2018; Xian et al, 2020). In these publications, NBR1, NDP52, and Optineurin and other adaptor proteins were inefficient in interacting and degrading RIG-I or cGAS. This literature, along with our old and new data, strongly suggests that p62 is the major adaptor for IRGM-mediated autophagic degradation of RIG-I, TLR3, and cGAS. We are thankful to the reviewer for this suggestion and helping us to make our conclusion stronger!!

Minor:

- As indicated above, there are different types of autophagy and the authors use the term autophagy throughout the text, without defining precisely. Therefore, I would suggest to use the term macroautophagy in state of autophagy in the entire text.

Response: We agree with the reviewer. We have now described the micro, macro, and chaperone autophagy in the introduction and also mentioned that we are using tern autophagy in this paper for macroautophagy (paragraph 4).

- The authors looked at AIM2. Why did they not also follow NLRP3, another subunit of the inflammasome, which they previously shown that is interacting with p62 via IRGM (Mehto et al, 2018, Mol Cell, 73, 429-445)? It would have served as a positive control.

Response: In reality, the work published in Molecular Cell and submitted here were all started together. So we were always performing experiments in parallel for this work and NLRP3. Later, we divided the work into two components. Here, we are showing the role of IRGM in controlling the IFN response via modulating nucleic acid-sensing under basal conditions. AIM2 is a nu-

cleic acid sensor and also inflammasome inducer. So it was used as a control in the previous study as well as in this study.

- The authors examined the turnover of TRIF, which appears to be mediated by TAXBP1 and NDP52, but not that of TRIF6, which has been shown to depend on NDP52 and p62 (Chan et al, 2016, *J Virol*, 90, 10928-10935; Inomata et al, *Cell Mol Life Sci*, 69, 963-79). They could check the mRNA and protein levels of this protein. Similarly, the NF κ B pathway components IKK β and IKK γ are also degraded by autophagy in a p62-dependent manner (Liu et al, 2018, *J Mol Cell Biol*, 10, 205-215; Zhang et al, 2017, *Nat Commun* 8, 2164). The authors could look at the role of IRGM in the lysosomal turnover of these two components as well, by also looking at the mRNA and protein levels. An accumulation of TRF6, IKK β and IKK γ in cells depleted of IRGM will further strengthen their main conclusion and model.

*Response: We completely understand the scientific curiosity and thankful for the excellent suggestion of the reviewer. This manuscript covers the aspects of the regulation of nucleic acid sensor by IRGM-mediated autophagy and mitophagy in relation to autoimmunity. In this context, it is important to understand how IRGM control nucleic acid signaling at **basal levels**. We have performed 100's of experiments (Nine main figure, six supplementary figures, having >200 panels) to show that in steady-state conditions, IRGM suppresses cGAS-STING and RIG-I-MAVS signaling to control interferon response, which plays a significant role in autoimmunity. Most humbly, we feel that the above suggestion is a completely new study and is out-of-scope of the study area of the current manuscript. Further, we think it may dilute the main theme of the paper.*

*However, we would like to inform reviewer that the thought process is **very valid**, and this work, where we are looking at how IRGM controls TRAF6-IKK signaling in infection conditions (**not basal**) is under progress and is a standalone study.*

- In the absence of IRGM, PARKIN is recruited to mitochondria. This implies a higher ubiquitination of this organelle and possibly a recruitment (or not) of p62 and the other autophagy receptors. It would be interesting to check which kind of influence IRGM has on this event.

Response: As suggested by the reviewer, we performed the experiment and the data indicate that Ubiquitin recruitment is higher on mitochondria in IRGM KD cells (Figure 7, Manuscript Figure 5E). However, there is no difference in the recruitment of p62 (Figure 8, Manuscript Appendix Supplementary Figure 3C). Although, clearly, the p62 aggregates were bigger in IRGM depleted cells.

Figure 7

Figure 8

- It has been shown that NDP52 is involved in the turnover of NLRP3, TRIF, TRAF6 and MAVS. In the discussion, the authors must comment whether that are other regulator than IRGM regulating this autophagy receptor, or it may be that it is also regulated by IRGM but they failed to detect an interaction in the used conditions.

Response: This is correct that NDP52 play a role in the turnover of NLRP3, TRIF, and TRAF6 but not cGAS and RIG-I (Chen et al, 2016; Du et al, 2018; Prabakaran et al, 2018; Xian et al, 2020). As suggested by reviewer, we have discussed the role of other adaptor proteins in controlling PRR's. NDP52 does not interact with IRGM, and also NDP52 does not rescue the IRGM-mediated suppression of IFN response (so does the NBR1) and hence indicates that NDP52 is not a part of IRGM orchestrated autophagy complex.

(We would like to bring in kind notice of the reviewer that in Figure 9 of the old manuscript (New Figure) in THP1 data sets, we have inadvertently made a mistake in fold changes calculations in qRT-PCR's results (both in controls as well as knockdowns). We have now corrected the calculations, and these changes do not alter the results and conclusions anyways. No change is made in the text.)

References:

Chauhan S, Mandell MA, Deretic V (2015) IRGM governs the core autophagy machinery to conduct antimicrobial defense. *Mol Cell* **58**: 507-521

Chen M, Meng Q, Qin Y, Liang P, Tan P, He L, Zhou Y, Chen Y, Huang J, Wang RF, Cui J (2016) TRIM14 Inhibits cGAS Degradation Mediated by Selective Autophagy Receptor p62 to Promote Innate Immune Responses. *Mol Cell* **64**: 105-119

Du Y, Duan T, Feng Y, Liu Q, Lin M, Cui J, Wang RF (2018) LRRC25 inhibits type I IFN signaling by targeting ISG15-associated RIG-I for autophagic degradation. *EMBO J* **37**: 351-366

Prabakaran T, Bodda C, Krapp C, Zhang BC, Christensen MH, Sun C, Reinert L, Cai Y, Jensen SB, Skouboe MK, Nyengaard JR, Thompson CB, Lebbink RJ, Sen GC, van Loo G, Nielsen R, Komatsu M, Nejsum LN, Jakobsen MR, Gyrd-Hansen M, Paludan SR (2018) Attenuation of cGAS-STING signaling is mediated by a p62/SQSTM1-dependent autophagy pathway activated by TBK1. *EMBO J* **37**

Xian H, Yang S, Jin S, Zhang Y, Cui J (2020) LRRC59 modulates type I interferon signaling by restraining the SQSTM1/p62-mediated autophagic degradation of pattern recognition receptor DDX58/RIG-I. *Autophagy* **16**: 408-418

Referee #2:

In this study by Jena et.al. the authors have analyzed the role IRGM plays in regulating the innate immune system for recognition of nucleic acids, in particular. They propose that through promoting autophagy IRGM acts to suppress the interferon system. This is proposed to be through both autophagic clearance of PRRs such as cGAS, TLR3 or MDA5 as well as through promoting mitophagy. Thus loss of IRGM leads to enhanced production of DAMPs due to defective mitophagy, and subsequent hyperactivation of the PRRs leading to overproduction of interferon and upregulation of ISGs.

This study provides a very nice model for how loss of IRGM could lead to spontaneous inflammation, there are however a few points that could be further clarified.

We are very thankful to the reviewer for reading our manuscript thoroughly and admiring the work. We have put all the efforts to clarify each and every constructive comments below. In this process, we have performed several new experiments.

1) Role of mtDNA.

This point of reviewer has 6 queries. We are dividing the reviewer's comments/concerns into parts (red font superscript) so that we can answer each of them properly. Six new experiments were performed to answer the queries.

#1The authors propose that due to defective mitochondrial homeostasis in IRGM deficient cells, mtDNA is released into the cytosol and activates the cGAS/STING system. They also identify micronuclei as a source of cGAS/STING activation. The relative contribution to the effect has however not really been shown.

#2While the microscopy images using dsDNA antibodies certainly show more DNA in the IRGM deficient cells, it is not completely convincing that it is cytosolic in regards to the proposed mtDNA. Indeed in Figure S6 it almost seems as if the dDNA is outside of the cell.

#3In Figure 8a, using only Tom20 as a marker for mitochondria under these circumstances could be somewhat limiting and some marker on the inside of the mitochondria would be advisable to use to avoid possible complications due to parkin mediated degradation of Tom20 which has been shown previously.

Response:

#1*We understand the reviewer's concern. We found both mtDNA and micronuclei increased in the cytosol of IRGM knockdown and knockout cells. I am sure that the reviewer may acknowledge that it is not easy to define the relative contribution of these two entities in the induction of cGAS-STING pathway. This will be a highly intricate process to be bifurcated as ROS (as found in IRGM KD cells) will increase both. Specifically, it will lead to a work, which will not have significance pertaining to the main theme of the current manuscript.*

#2*We observed not only the increased amount of dsDNA **inside the cytosol** of IRGM KD/KO cells (Figure 6A-J and Figure EV4H-J, page 12 paragraph 3) but also **in the extracellular milieu**, which we have shown in Figure S5 by increasing the magnification to show the intercellular regions and probably the same is pointed by the reviewer. So supplementary Figure*

S5 (which reviewer might mistakenly written as Figure S6) in old manuscript and Appendix Supplementary Figure 3G and H in new manuscript, was used to illustrate the point that indeed there is an increased extracellular dsDNA outside the cells in case of IRGM KO cells in addition to the increased intracellular dsDNA. By several ways and in different cell lines, we repeatedly observed increased dsDNA **in the cytosol** of IRGM knockdown or knockout cells (BMDM and THP-1).

#2 To further strengthen the conclusions, we provide more new experiments and evidence, including the one asked by the reviewer:

1. In this revised manuscript, we performed immunofluorescence with IRGM knockdown HT29 colon epithelial cells (previous data was with THP-1 and BMDM's), the IF data again clearly show a considerable increase of cytosolic dsDNA in IRGM knock down HT29 cells compared to control cells (**Figure 1**) (Manuscript Figure EV4J). Please note that the normal single-cell morphology of HT-29 is rounded.

2. Our conclusion of increased cytosolic DNA was not just based on immunofluorescence assays (THP-1, BMDM, and now HT29 cells) but also was based on the cytochrome c oxidase subunit 1 (mitochondrial gene) qRT-PCR from cytosolic fractions (devoid of mitochondria) of BMDM's (Manuscript Figure 6F). The data showed about 5 fold induction of mtDNA in cytosolic fractions of IRGM knockout BMDM's compared to control cells. **Please note that the mitochondrial fraction is removed in this procedure, and the only cytosolic fraction is used for the analysis of mtDNA.**

3. Since this is an important concern pertaining to the theme, we repeated the mtDNA (Cytochrome c oxidase subunit 1 primers) qRT-PCR with a cytosolic fraction (minus mitochondria) extracted from HT-29 colon epithelial control and stable IRGM knockdown cells. We again observed a significant increase of mtDNA in the cytosolic fraction of knockdown cells compared to the control cells (**Figure 2**) (Manuscript Figure 6G).

4. **#3** As suggested by the reviewer, we repeated the experiments with cytochrome C antibody, an inside protein marker of mitochondria (instead of TOM20). The results again showed an increased amount of cytosolic DNA (outside mitochondria) in IRGM knockout BMDMs compared to the control BMDMs (**Figure 3**) (Manuscript Figure EV4H). In these assays also a clear increase in extracellular DNA was observed in IRGM knockout BMDMs (**Figure 3, Figure 4**). Since these assays are confirmatory, only one figure is inserted in the main manuscript; rest are for the scrutiny of the reviewer.

(We would like to bring to the kind notice of the reviewer that similar work from Dr. Michael Fessler group from NIH is under review in another journal. Here is the link to their abstract,

Figure 3

Figure 4

which they presented in keystone meeting <https://virtual.keystonesymposia.org/ks/articles/4294/view>.

They also have similar findings (mtDNA-cGAS-STING-IFN), but their work is mainly in mice, and we have shown in both mice and humans.)

#4 The fact that there are more and seemingly larger structures for mtDNA does not mean they are cytosolic or that this is what is driving the effect. The authors could make use of RhoO cells (cells lacking mtDNA) to confirm this. This can be achieved rapidly with mitochondrial targeted nucleases or the traditional method of ethidium bromide in the culture. These cells could then be examined for IFN responses as in the manuscript to confirm if mtDNA or micronucleoids are playing a significant role.

Response:

#4 We appreciate the wonderful suggestion. We used the ethidium bromide method to deplete mitochondrial DNA and generate Rho-0 cells in HT29 IRGM stable knockdown cells and scored the IFN response. The data clearly show that the cytosolic mtDNA is considerably reduced in IRGM knockdown Rho-0 cells compared to parent cells as measured by qRT-PCR with cytochrome c oxidase subunit 1 primers (Figure 5)

(manuscript Fig 6H). Additionally, the increased ISGs (MX2 and ISG15) levels were significantly rescued in IRGM knockdown Rho0 cells compared to parent cells (Figure 6) (Fig 6I). The data suggest that indeed mtDNA play a significant role in the induction of interferon response in IRGM-depleted cells.

#5 It is also unclear from the DNASE treatment how the samples were prepared. It seems that total cells were taken, but this of course would take mitochondrial content too, and thus the increased signal could simply be due to increased mitochondrial mass, not increased cytosolic DNA. This would fit with the images, which seem to suggest more mitochondrial mass in the IRGM deficient cells. If some other technique was used to only isolate cytosol, then this needs to be described in more detail as it is not clear and #6 verified by also performing PCR for mitochondrial DNA as in supp figure 6 but also +/- DNASE transfection to confirm that the signal for mitochondrial DNA is reduced.

Response:

#5 We are very sorry that there is a confusion that we have used total cells for this analysis. We used only the cytosol (subtracted of mitochondria) as described in the material and methods (in the previous version also). We have now stated this in manuscript main text also at page number 13 paragraph 1.

#6 Further, as suggested by the reviewer, we electroporated DNase 1

and performed qRT-PCR with cytochrome c oxidase subunit 1 to confirm whether DNase 1 is able to effectively reduce the mtDNA soiling of the cytosol of IRGM knock out BMDM's. We found almost 6 folds induction of mtDNA in IRGM knockout cells compared to the control cells. The DNase 1 treatment of IRGM KO cells considerably reduced the mtDNA content in the cytosol (Figure 7) (Manuscript Figure 6J), suggesting that indeed, DNase 1 electroporation reduces the cytosolic mtDNA.

We thank the reviewer for brilliant suggestions and helping us in strengthening our results and conclusions.

2. #1 Role for IRGM interaction with receptors? While the authors provide strong support for the interaction of IRGM with the various PRRs, but it is not clear to what extent this mechanisms is at play in relation to the mitophagy deficiency. It is not clear from the discussion or the results to what extent the authors believe that the degradation of the receptors themselves as shown from figures 2-5 is important or the production of the DAMPS from mitochondria. I realise this is not easy to distinguish, but this could be better covered in the discussion.

#2 at the least and could be experimentally addressed to some extent as follows: The contribution of mitophagy failure to the inflammation could be further addressed and its role further differentiated from that of degradation of the receptors by looking in parkin deficient cells to show that loss of mitophagy specifically does or does not lead to the increased interferon responses. This could be coupled with dsDNA/RNA staining as already done to show the loss of mitophagy still results in these effects, but the degradation of the receptors should be more or less unchallenged.

Response:

#1 *The reviewer view is absolutely correct that it is not straightforward to discern the extent of contribution by PRR accumulation vs mtDAMPs accumulation to the heightened interferon response in IRGM knockdown cells. As both of these pathways are intricately interconnected. The presence of DAMPs in cytosol results in interferon response via nucleic acid-sensing PRRs, and these PRRs themselves are Interferon stimulated genes. So any trigger, which will modulate ISG's will affect nucleic acid PRR's expression and vice versa. Our data suggest that both defective autophagy of PRRs and defective mitophagy contributes to such a robust interferon response in IRGM knockdown cells. Where the increased PRRs primes the response, and the DAMPs fuels the response. As suggested by the reviewer we have now discussed this*

Figure 8

Figure 9

point in the discussion (page number 16, Paragraph number 2)

#2 *We performed the experiments, as suggested by the reviewer. Interestingly, we found that the knockdown of PARKIN in human THP1 cells resulted in a significant increase of ISG's in basal conditions its (Figure 8) (Manuscript Figure EV4K). However, the extent of induction of ISG's was found to be significantly lower as compared to IRGM knockdown cells indicating that PARKIN-mediated mitophagy may contribute but may not be the sole reason for heightened IFN response in IRGM knockdown cells. Further, we observed that*

depletion of PARKIN did increased cytosolic mtDNA (Figure 9) (Manuscript Appendix Supplementary Figure 4A) and also a marginal increase in protein expressions of PRRs

(expected as they are also ISG's) (Figure 10) (Manuscript Figure EV4L). However, the increase was marginal as compared to IRGM knockdown cells (Figure 2B), which indicates that there is a role of IRGM mediated autophagic degradation of PRR's, in addition to mitophagy defect.

In a nutshell, it appears (as realized by the reviewer also) that it is not straight forward to delineate the EXACT extent of contributions by the two events (mitophagy defect vs PRR expression levels) as both are highly interconnected events. Nevertheless, several experiments in the old manuscript and now in the new manuscript strongly suggest that both events contributes to perturbed IFN response in IRGM depleted cells and above experiment further strengthen this conclusion.

We also attempted one more experiment to understand the role of mitophagy in IRGM-mediated IFN response (asked by reviewer#3). We used Mdivi1, known mitochondrial fission and mitophagy inhibitor (Luo et al, 2019; Vo et al, 2019; Yao et al, 2019). The results (Figure 11) (Manuscript Figure EV4M) show that:

1) Mitophagy inhibition indeed induces interferon response, but that increase is lesser than induction of interferon response in IRGM knockdown cells. Again, this indicates that other factors in addition to mitophagy contribute to IFN induction in IRGM depleted cells.

2) There is no further increase of IFN response in IRGM depleted cells upon Mdivi1 treatment, suggesting that mitophagy is already maximally inhibited in IRGM depleted cells.

#1 Do the authors have a suggestion as to why interaction of IRGM with the receptors is needed? They have previously reported that IRGM regulates autophagy induction through binding to beclin1 complexes to promote their activation? It is not clear then why interaction directly with the receptors would be needed, or

#2 how this is happening? Is it through ubiquitylation of the receptors after their activation? It would be informative to determine if the receptors are also ubiquitylated.

#3 Additionally, Is there a role for p62 in the regulation of mitophagy in your system or is it simply the degradation of the receptors? It would be informative to also look at mitophagy in p62/- cells in your systems to answer this. #4 In addition it would be interesting to see if, in the absence of p62, does IRGM still interact with the PRRs? P62 is known to mediate degradation of STING and TRIF and its phosphorylation via TBK1 regulates recruitment LC3 to the complexes. Perhaps IRGM is also recruited through p62 mediated binding to ubiquitylated PRRs in a similar fashion to help recruit the phagophore machinery?

Response:

#1 During this study, first, we found that IRGM degrades nucleic acid sensors (PRRs), and this degradation is mediated by autophagy. Several lines of evidence in old manuscript and now revised manuscript that indeed we found p62 is a critical factor for degradation of these sensors. We think, and an overwhelming amount of literature show that the PRRs degradation needs receptors (mostly p62-dependent), suggesting that degradation of PRRs is a selective rather

than a bulk process. Thus, autophagy may require a special set of receptors and interacting proteins to accomplish this job and we feel, and our data strongly argue the IRGM along with p62 is one of such systems.

#2 #4 To answer concern #2 and #4, we attempted a few more experiments to understand how IRGM-mediated p62-dependent selective autophagy of PRRs takes place. In our old

manuscript, we showed that IRGM interacts strongly with p62 (but not with other receptors), and the p62 depletion rescues the IRGM-mediated degradation of PRRs and also the heightened IFN response.

Here, as suggested by the reviewer, first, we asked whether p62 is required for the interaction between IRGM and PRRs. For this, we performed co-IP experiments between IRGM and RIG-I,

cGAS, and *TLR3* in the presence and absence of *p62*. The results show that the depletion of *p62* considerably reduces the interaction between *IRGM* and *PRR*'s (Figure 12)

(Manuscript Figure 4K-M), suggesting that *p62* is absolutely required for these interactions. Next, we asked what is the role of *IRGM* in the interaction between *p62* and *PRR*'s? We previously found that *IRGM* can oligomerize and can act as a scaffold protein for increasing the interaction of *PRR*'s

and autophagy proteins (Chauhan et al., 2015, *Molecular Cell*). We tested here whether *IRGM* can potentiate interaction between *cGAS*/*RIG-I*/*TLR3* and *p62*. Indeed, we found that the interaction between *p62* and *PRR*'s are increased in the presence of *IRGM* in co-immunoprecipitation assays (Figure 13) (Manuscript Figure 4P-R). Taken together, the data suggest that *p62* and *IRGM*, along with *PRR*'s form a ternary complex (at least) where both *p62* and *IRGM* cooperatively increase each other interaction with *PRR*'s.

Further, we know that the *p62* interacts with ubiquitinated (especially K63-linked) cargoes to deliver them to autophagosomes. In a quest, how *IRGM* might increase the interaction between *p62* and *PRR*'s, we tested whether the presence of *IRGM* increases the K63-linked ubiquitination of *PRR*'s. Indeed, the presence of *IRGM* increased the K63-linked ubiquitination of *PRR*'s (Figure 14) (Manuscript Figure EV3K), suggesting that *IRGM* increases interaction between *p62* and *PRR*'s by increasing K63-linked ubiquitination of *PRR*'s. To conceptualize the whole scenario, our data indicate that *IRGM* supports the ubiquitination of *PRR*'s leading to their *p62*-mediated selective autophagic degradation.

Figure 15

Figure 16

We would like to mention that we understand that IRGM is not an E3-Ligase, and it cannot itself increase the ubiquitination of these PRRs. At this moment, we don't know what all E3-ligases can engage with IRGM to control the ubiquitination of PRRs. This is an expedition itself (another manuscript), and moreover, it is completely out of the scope of the theme of the current manuscript

Response to concern #3:

#3 In the basal conditions, in IRGM knockdown cells, we observed increased accumulation of Ub over the mitochondria (Figure 15) (Manuscript Figure 5E), but there is no difference in the recruitment of p62 (Figure 16) (Manuscript Appendix Supplementary Figure 3C). Although, clearly, the p62 aggregates were bigger in IRGM depleted cells (Figure 16). It indicates that p62 is not important for mitophagy, at least in our conditions. Indeed, knocking down p62 does not change the extent of mitophagy flux in THP-1 cells (Figure 17) (Manuscript Appendix Supplementary Figure 3D). The data suggest that p62 is not important for mitophagy, at least in our cell system and in basal conditions.

Figure 17

3. Minor points:

Figure 5, have the authors also looked in beclin1 deficient cells without IRGM overexpression? Presumably if the phenotype is only due to autophagic degradation of receptors and mitophagy, then the beclin1 deficient cells should look the same as IRGM^{-/-} cells in terms of ISG upregulation.

Response: The depletion of Beclin1 resulted in significantly increased expression of ISG's in basal conditions itself p62 (Figure 18) (Manuscript Figure EV3I). The induction, however, is less than what is observed in the case of IRGM depletion. It appears that IRGM depletion is more detrimental than Beclin1

depletion and maybe additional pathways to supplement the IFN response. The point is discussed now in the discussion (page number 16 paragraph number 2).

Figure 6I-J. #1 It is curious given the role of IRGM in regulating autophagy as well as targeted autophagy/mitophagy, that no increase in LC3-II is seen in the IRGM^{-/-} samples. Can the authors explain this discrepancy?

#2 Additionally I am not so sure that simply by showing no increase in parkin or pink1 in the presence of bafilomycin A that one can argue that mitophagic flux is blocked given their tight

regulation through PTMs and stability/localisation etc.. Other mitochondrial proteins, preferentially ones inside the mitochondria would be useful to further support this claim.

Response:

#1 In basal conditions, we never observed increased LC3-II upon knockdown of IRGM here (Figure 6I and 6J) or before (Mehto et al., Molecular Cell; Figure 5A and 5B). Several other publications also show similar results (either decrease or no change) in basal conditions (Hansen et al, 2017; Kumar et al, 2018; Lin et al, 2016; Singh et al, 2006). We always found that both LC3-I and LC3-II is reduced upon knockdown of IRGM. Upon inhibition of autophagy flux by bafilomycin, we always found that LC3-II is lower in IRGM-depleted cells compared to the control cells suggesting that autophagy flux in the basal condition is lower in IRGM-depleted cells. It may be possible that the depletion of IRGM not only reduces autophagy flux but also reduces transcription of MAP1LC3 (LC3-1 in western blots), and that could be the reason that both LC3-I and LC3-II remains less (however this is just hypothesis).

#2 We have shown results of TOM20 in the same figure along with PARKIN/PINK. Additionally, now we have performed western blot with cytochrome C antibody, an inner membrane protein (Figure 19) (Manuscript Figure EV4E). The results show that the mitophagy flux (Bafilomycin panels) is clearly inhibited in IRGM knockdown cells.

Figure 6K, there should be labels about drug additions to the seahorse data.

Response: We have corrected it. Thanks!!

Overall this is a very interesting study and will be of great interest to EMBO Reports readers.

Response: We are very thankful to the reviewer for appreciating our work and recommendation. We have tried our best to answer each of the concerns of the reviewer. The experiment recommended by the reviewer has made our manuscript stronger on each point. Thanks a lot!

Although, not asked by the reviewers, we would like to bring to the kind notice of reviewer that now we have checked whether IRGM directly interact with cGAS, RIG-I, and TLR3 using purified proteins and in-vitro translated PRR's in GST-pull down assays. A direct and specific interaction was observed between IRGM and all the three PRR's (Figure 20) (Manuscript Figure 3E). The GST-IRGM strongly interacted with cGAS but relatively weaker interaction was observed between IRGM and RIG-I or TLR3. Negligible interaction was observed with GST controls.

Chauhan S, Mandell MA, Deretic V (2015) IRGM governs the core autophagy machinery to conduct antimicrobial defense. *Mol Cell* 58: 507-521

Chen M, Meng Q, Qin Y, Liang P, Tan P, He L, Zhou Y, Chen Y, Huang J, Wang RF, Cui J (2016) TRIM14 Inhibits cGAS Degradation Mediated by Selective Autophagy Receptor p62 to Promote Innate Immune Responses. *Mol Cell* 64: 105-119

Du Y, Duan T, Feng Y, Liu Q, Lin M, Cui J, Wang RF (2018) LRRC25 inhibits type I IFN signaling by targeting ISG15-associated RIG-I for autophagic degradation. *EMBO J* 37: 351-366

Hansen MD, Johnsen IB, Stiberg KA, Sherstova T, Wakita T, Richard GM, Kandasamy RK, Meurs EF, Anthonsen MW (2017) Hepatitis C virus triggers Golgi fragmentation and autophagy through the immunity-related GTPase M. *Proc Natl Acad Sci U S A* 114: E3462-E3471

Kumar S, Jain A, Farzam F, Jia J, Gu Y, Choi SW, Mudd MH, Claude-Taupin A, Wester MJ, Lidke KA, Rusten TE, Deretic V (2018) Mechanism of Stx17 recruitment to autophagosomes via IRGM and mammalian Atg8 proteins. *J Cell Biol* 217: 997-1013

Lin YC, Chang PF, Lin HF, Liu K, Chang MH, Ni YH (2016) Variants in the autophagy-related gene IRGM confer susceptibility to non-alcoholic fatty liver disease by modulating lipophagy. *J Hepatol* 65: 1209-1216

Luo X, Liu R, Zhang Z, Chen Z, He J, Liu Y (2019) Mitochondrial Division Inhibitor 1 Attenuates Mitophagy in a Rat Model of Acute Lung Injury. *Biomed Res Int* 2019: 2193706

Prabakaran T, Bodda C, Krapp C, Zhang BC, Christensen MH, Sun C, Reinert L, Cai Y, Jensen SB, Skouboe MK, Nyengaard JR, Thompson CB, Lebbink RJ, Sen GC, van Loo G, Nielsen R, Komatsu M, Nejsum LN, Jakobsen MR, Gyrd-Hansen M, Paludan SR (2018) Attenuation of cGAS-STING signaling is mediated by a p62/SQSTM1-dependent autophagy pathway activated by TBK1. *EMBO J* 37

Singh SB, Davis AS, Taylor GA, Deretic V (2006) Human IRGM induces autophagy to eliminate intracellular mycobacteria. *Science* 313: 1438-1441

Vo MT, Smith BJ, Nicholas J, Choi YB (2019) Activation of NIX-mediated mitophagy by an interferon regulatory factor homologue of human herpesvirus. *Nat Commun* 10: 3203

Xian H, Yang S, Jin S, Zhang Y, Cui J (2020) LRRC59 modulates type I interferon signaling by restraining the SQSTM1/p62-mediated autophagic degradation of pattern recognition receptor DDX58/RIG-I. *Autophagy* 16: 408-418

Yao N, Wang C, Hu N, Li Y, Liu M, Lei Y, Chen M, Chen L, Chen C, Lan P, Chen W, Chen Z, Fu D, Ye W, Zhang D (2019) Inhibition of PINK1/Parkin-dependent mitophagy sensitizes multidrug-resistant cancer cells to B5G1, a new betulinic acid analog. *Cell Death Dis* 10: 232

Referee #3:

IRGM has been shown to connect with several autoimmune diseases. This manuscript tried to provide mechanistic understanding for IRGM in mediating IFN signaling as well as autoimmunity. However, the phenomena in the manuscript were inconsistent with previous studies and should be further confirmed, and several results in this manuscript were not consistent. In addition, the figures are not well-organized and made, and do not fit the style of EMBO Reports to my knowledge.

Response:

We are very thankful to the reviewer for reviewing our manuscript. Below, we have attempted to explain each point raised. In addition, we have performed all the experiments as suggested.

The EMBO reports guidelines says that we don't need to reformat our manuscript at the first submission. Therefore, we did not changed it. Now, we have formatted the new manuscript according to EMBO reports guidelines. We have also attempted to improve the organization of manuscript. Thanks a lot for the suggestions!

We very respectfully differ on the overall assessment of the manuscript by the reviewer and also from the view that our data is inconsistent with previous studies. We have attempted to explain each of the queries raised.

First, we would like to mention the main features of the manuscript about novelty, breadth/amount of work, and biological/physiological relevance, which is very relevant to overall assessment and below point-to-point response. We sincerely request to please consider these points before reading the point-to-point response.

1. **Figure 1.** *This is the first report that attempted transcriptomics (in three different cell types colon, brain, macrophages) in human and mice and showed for the first time that IRGM is a **master** regulator of interferon response. It regulates almost all genes of interferon response. Please note that before this study there is no other literature that show so evidently that human IRGM (a 21 Kda protein) and mice Irgm1 (a 42 Kda protein) although having different in molecular weight are so similar in inflammatory responses. This is an important finding since to date there was always a dilemma whether mice Irgm1 studies make any sense for humans.*

2. **Figure 2.** *We have showed that indeed the complete IFN signaling pathway starting from nucleic acid sensors, adaptors proteins, TBK1, IRF3/IRF7, JAK/STAT, and ISG's are activated and induced in IRGM knockdown cells at protein levels in both human and mice. Enormous amount of work (70 western blots using 32 antibodies) was performed to validate the pathways. NO study have reported this.*

3. **Figure 3.** *In order to understand mechanisms, we performed screen of IRGM-PRR's interaction and found that three nucleic acid sensors, cGAS, TLR3, and RIG-I interact with IRGM but several other not. Using different types of several endogenous and exogenous assays, we showed that IRGM degrades the three PRRs and controls IFN response. This is for the **first time** shown in the literature.*

4. **Figure 4 and 5.** *We further observed that the IRGM-mediated degradation of these PRRs and the control of the IFN response is autophagy dependent. To elaborate more this mechanism, we showed that IRGM invokes ATG5 and BECLIN1 dependent autophagy for degradation these PRRs. Furthermore, we found that IRGM utilizes p62 as adaptor molecule for selective*

autophagic degradation of these PRRs. This complete mechanism of IRGM-mediated control of PRRs is novel and no other work has reported any part of this work yet.

*5. **Figure 6, 7 8, 9.** Although, the above data could be a manuscript itself, we didn't stopped here. Since we want to understand what is the mechanism that fuels such a robust IFN response. We found that mitophagy is inhibited in IRGM depleted cells (knockdown and knock-out), mitochondria potential is increased, mitochondrial respiration is reduced, mtROS/ROS is increased, mtDNA/RNA has soiled the cytoplasm. We further showed that the mitochondrial DAMPs are the fuels, which maintains strong IFN response in IRGM knock out cells and knock out mice. Furthermore, we showed that these mtDAMPS induces cGAS-STING and RIG-I-MAVS signaling leading to a persistent IFN response in IRGM-depleted cells and mice. Each and every part of this point are novel and to date no one in the literature showed that IRGM controls cGAS-STING and RIG-MAVs pathways.*

6. In nutshell, we have performed enormous amount of novel work to understand the detailed mechanism by which IRGM controls cGAS-STING and RIG- MAVS signaling to control the IFN response.

During the revision process, we have now performed ~25 new experiments to further strengthen the conclusion.

1. #1 As the author indicated, IRGM was identified as an autoimmune diseases-related gene and plenty of studies from different groups have tried to elucidate the relationship between IRGM and autoimmunity. The author should first fully discuss former works about IRGM and provide basic understanding about IRGM.

#2 For example, in the recent article entitled "The Crohn's Disease Risk Factor IRGM Limits NLRP3 Inflammasome Activation by Impeding Its Assembly and by Mediating Its Selective Autophagy" from Molecular Cell indicated that knockout or knockdown of IRGM/Irgm1 could enhance the pro-inflammatory responses and IRGM/Irgm1 protects from pyroptosis and gut inflammation in a Crohn's disease mouse model by targeting NLRP3. However, in Figure 1 of this study, the author showed that IRGM/Irgm1 deficiency led to strong induction of IFN signaling in the steady-state conditions and suggested the correlation between IFN induction and autoimmunity. Compared with the induction of IFN signaling in IRGM/Irgm1 deficiency cells, what happens to other signaling such as pro-inflammatory response and NLRP3 inflammation.

#3 And the author might better analyze different pathways that may induce autoimmune responses in IRGM/Irgm1 deficiency cells and figure out which signals contribute dominantly to the autoimmunity.

Response:

#1 *As per the suggestion of the reviewer, we have now discussed other "IRGM-related studies for basic understanding" in the results, introduction and discussion also. Please see introduction-paragraph 3 and 4; Results-page 8, paragraph 3; Discussion- paragraph 3, 4 and 5. Only the relevant literature is cited. Thanks for the suggestions.*

#2 *The publication indicated by the reviewer in the comment, published in Molecular Cell is from our lab. Most of the published literature on "IRGM and autoimmunity" is genetic and functional*

“correlations” (as indicated in introduction paragraph 3), and no mechanistic studies were attempted to date.

We would like to indicate that there is a significant difference between mechanisms leading to different kinds of autoimmune diseases. Also, whether Crohn's is an autoimmune disease is highly debated as external/environmental triggers/factors plays a deceive role in Crohn's, including the gut infection. On the other hand, interferonopathies are the set of autoimmune disorders with the manifestation of quite a high amount of interferons and interferon response genes. In the introduction of this manuscript, we mentioned that we want to understand how IRGM control the innate immune system in basal conditions (**not stressed or triggered conditions**).

Please note that IRGM controls NLRP3 and Crohn's diseases (mice model) when the external stress is provided. I am sure reviewer know that NLRP3 inflammasome activation requires two signals “first signal (eg. LPS) primes the cells and increases the expression of NLRP3, ASC, and pro-IL-1 β , and the second signal (Nigericin, MSU, Silica, etc) leads to the assembly and activation of the complex. The previous study of our lab, which is mentioned by the reviewer (Mehto et al., 2019; Molecular Cell), attempted to understand how IRGM controls Crohn's disease by controlling innate immunity (NLRP3 inflammasome) under external stress conditions whereas this study tried to understand that how IRGM controls innate immunity **under steady conditions** to prevent autoimmune diseases including interferonopathies.

Please note that in **basal conditions**, IRGM knockdown does not induce/effect NLRP3 inflammasome or NLRP3 transcription until we trigger it with LPS, LPS/Nigericin, or other stresses (Mehto et al., 2019; Molecular Cell).

In the basal conditions itself, in IRGM knockdown cells and in IRGM knockout mice, the interferons are robustly induced. The transcriptomic data shows that **this is the predominant** pathway induced in basal conditions in three different cell lines and in humans and mice.

#3 The theme of the manuscript is “IRGM controls cGAS-STING and RIG-I-MAVS signaling to modulate IFN response”. Although in the above paragraphs we have explicitly described the difference in “IRGM mediated regulations” in different conditions, and also we firmly believe that IFN response is the predominant pathway leading to autoimmunity, but in no part of this manuscript, we have claimed that we have described all the autoimmunity pathways (impossible to do so in single manuscript) or we intend to do so. Within the limits of a single manuscript and the scope and theme of this manuscript; we have attempted and discovered all the possible mechanisms leading to **“higher IFN response in IRGM depleted cells”**. I am sure that the reviewer may appreciate the enormous amount of data shown in this manuscript (~200 panels, 17 figures) pertaining to this theme and conditions.

2. **#1** One serious concern is that plenty of the western blots (including the loading controls) appear to be overexposed/saturated. It makes it hard to tell the differences between different lanes. Since this data in Figure 2 to Figure 6 largely relies on quantification of western bands, such mistake would have a profound impact on the validity of the conclusions. The authors should use the low exposed images or reduce the loading quantity of the samples. **#2** In addition, the authors should show how many times they repeat such experiments, and show these repeats in the supplements or the response letter.

#1 We are very sorry if the reviewer feels so. We would have happily changed the exposures if the reviewer had pointed out the blots. The two other reviewers did not find any problems in ex-

posures. From our side, we have very carefully selected the optimum exposures in our old manuscript itself to show the results. Nevertheless, we have again gone through each of the blot and, replaced with better exposures if appears to have any problem.

#2 For western blotting from mice organ/BMDM lysates, we have used two or three mice, and we have either included all the three replicates or two replicates in figures itself. For the human cell line, we have similar number of biological replicates, and most of the figures contains already two replicates in old manuscript. We have now included raw western blots of the figures (Source data) with repeats including for those that are not presented in the manuscript.

3. #1 In Figure 2, the author showed the enhanced protein levels of different sensors in IFN signaling pathway in IRGM deficiency cells. The authors should point out the reasons for showing multiple β -Actin in one figure such as Figure 2C, 2D, 2G, 2N.

#2 I supposed the authors showed different samples they collected different times. And the loading controls from the samples collected from different times showed obviously inconsistent.

#3 For example, in Figure 2D, the Actin in the sixth line together with Irgm1 showed a sustained decline while the two Actin in the fourth and ninth lines remained unchanged. In Figure 2N the last two lanes of Actin in line three were much weaker while the Actin in the line 6 showed similar.

#4 The authors should use one group of samples collected one time to run a set of western blots instead of using different samples collected from different times to piece together one figure.

#5 And the author casually put the repeated results in one figure such as Figure 2E, 2H, 2M, 3G, 3H and 3I, and some of these repeated results showed inconsistent.

#6 For example, in Figure 2M, the levels of p-STAT1 in line 3 and line 4 were much higher than the p-STAT1 levels in line 1 and line 2, and more obvious differences were also showed in p-STAT2 and IRF9 in the same figure.

#7 These inconsistencies above largely reduce the credibility and reliable of this manuscript.

#1 *It is generally believed (please see published reference below) that to reduce artifacts each western blot for any specific protein should be accompanied with its own loading control. We invariably do it to be sure that each of the blot results is controlled by its own actin blot. it is considered as best practice. Here are some publication, which discuss this in details: (1) please read points 3.3 to 3.6 in (Butler et al, 2019)) (2) This is another publication (Pillai-Kastoori et al, 2020), which talks about "systematic approach to perform Western blotting" gives several references for this statement:-"Normalization of a target protein is most accurate when the target protein and ILC (internal loading control) are detected in the same lane on a single blot".*

Nevertheless, based on reviewers' concern, we have now removed extra actin blots.

#2 *The reviewer assumption is not correct. Three mice lysates were made together in one time. However, western blot with different antibodies are not performed in a same day (not possible with ~30 antibodies used in Figure 2). I am sure that reviewer might be knowing that each antibody need standardization in western blots and especially when using different cell line and ani-*

mal organs. In cell culture, 2 biological repeats or 3 biological repeats are performed, for that we collected the lysates and run together for single antibody.

#3 This concern raised by the reviewer again justify that each blot should have its own Actin blot. The problem pointed out by the reviewer is just artifact of blot development problem as our other two actin blots show no differences in the lanes (from the same lysates of three different mice). And even if the concern is correct that 6th lane has less actin, it does not affect the conclusion of our results anyways (or I will say it make it our data more robust). The reviewer should consider why we are have given many actin blots when we can get away with one (according to him also). It is just to show accuracy of each western blot.

Nevertheless, based on reviewers' concern, we have removed extra actin blots.

#4 We are sorry, but we respectfully disagree entirely. The reviewer's advice appears not to come under best practices (biological repeat vs technical repeats) (Bell, 2016; Vaux et al, 2012) (<https://www.licor.com/bio/guide/westerns/replicates>). The biological replicates are those where the experiments are repeated in a different time (maybe with different passage numbers and other variables), but when the experiments are performed simultaneously in triplicates, it is technical replicates (Bell, 2016; Vaux et al, 2012). We perform experiments in biological replicates, which is considered more reliable and reproducible (Bell, 2016; Vaux et al, 2012) (<https://www.licor.com/bio/guide/westerns/replicates>). We have used two or three biological replicates throughout the manuscript and there is NO piecing together anywhere. More than 60 antibodies are used in this study. Not all western give clear result in one shot.

#5 These are the biological replicates in the indicated figures. I am very sorry, but I am very much confused now, as in point number 2 above, the reviewer wants us to show repeats (even which are not present in this manuscript), and here when we are showing the biological repeats, the reviewer does not like it.

#6 This is a highly puzzling remark, in figure 2M, the lanes 1 and 2 are biological replicates of lane 3 and 4. Both of the biological replicates show the same results, but they are not replica since they are not technical repeats, and such minor difference is expected in western blotting's. Besides, both replicates convey the same results and conclusions. We could have removed the one replicates (even before submission), but we thought it would increase the confidence of the reviewer. Now, as per the reviewer suggestion, we have kept only one replicate at several places.

#7 We have now provided a response of all the concerns raised by reviewer in this point, citing the publications on Western blot methodologies and other best practices procedures. We feel that whatever issue raised by the reviewer increases (rather than decreasing) the credibility and reliability of our work. It took 4 years of hard work (several students and 6 labs) to complete this study. Most respectfully, I humbly request such comments of reliability and credibility hurts the sentiments of authors and should be sent directly to the editor.

We would also like to bring to the kind notice of the reviewer that similar work from Dr. Michael Fessler group from NIH is under review in another journal. Here is the link to their abstract, which they presented in keystone meeting <https://virtual.keystonesymposia.org/ks/articles/4294/view>. They also have similar findings (mtDNA-cGAS-STING-IFN), but their work is mainly in mice, and we have shown in both mice

and humans. This indicates the reliability and credibility of the work, as two labs independently reaching to same mechanisms and conclusions.

4. #1 The authors claimed that IRGM/Irgm1 deficiency led to enhancement of transcription levels of multiple sensors in IFN signaling pathway, which suggested overexpressing IRGM/Irgm1 might reduce the transcription levels of these sensors, such RIG-I, cGAS and TLR3 since overexpression of IRGM led to reduction of mRNA levels of MX2 and ISG15 in Figure 3J and 3K. In Figure 3E, the authors should first show whether overexpression of IRGM/Irgm1 affect the mRNA levels of RIG-I, cGAS and TLR3.

#2 Secondly, the authors should rule out the possibility that the reduction of the endogenous protein levels of these sensors were due to their reduced transcription levels.

#3 Moreover, in Figure 3E, overexpression of IRGM led to significant reduction of the protein level of RIG-I, cGAS and TLR3 even though the protein level of Flag IRGM was not very strong. However, in Figure 3F, overexpression of IRGM failed to significantly reduce the level of cGAS and TLR3 even though the expression level of Flag-IRGM showed much stronger than that in Figure 3E. The author should explain this.

#1 We performed the experiment as suggested by the reviewer, and as expected, we found that overexpression of IRGM reduced the transcription levels of RIG-I and TLR3 but not of cGAS (Figure 1) (Manuscript Appendix supplementary Figure 2E) since RIG-I and TLR3 are interferon-stimulated genes beyond doubt anything which affects interferon response will affect RIG-I and TLR3 endogenous transcription.

#2 In the case of the cGAS, the possibility that the reduction of the endogenous protein levels is due to their reduced transcription levels, is ruled out. As there is no change in cGAS at transcription levels upon overexpression of IRGM (Figure 1), but protein is degraded (Figure 3).

#2 We think the reviewer has somehow missed a complete set of experiments. Actually, the question posed by the reviewer is exactly the same as we asked ourselves in the old manuscript and provided several experiments to rule out the possibility. Please see paragraph 1 and 2 on page number 9 in the old manuscript. We have written there, "Since cGAS, RIG-I, and TLR3 expression is controlled by IFN response, the reduction of endogenous levels of these proteins could be an indirect effect of IRGM-mediated suppression of

IFN response. To rule out this possibility,.....Overall, the data suggest that IRGM interacts and degrades RIG-I, cGAS, and TLR3 to keep type I IFN response under-check”.

Since in the above set of experiments (Western blots and luciferase assays), we are just scoring the stability of CMV promoter-driven ORFs (RIG-I, TLR3, and cGAS) expressions in the absence and presence of IRGM, **there is no question of endogenous promoter regulations.**

#2 To further rule out the possibility that the reduction of the endogenous protein levels of these sensors was due to their reduced transcription levels, we blocked the transcription in cells using actinomycin D and chase the Flag-RIG-I protein degradation in absence and presence of GFP-IRGM. The results show faster protein degradation in the presence of GFP-IRGM in comparison to GFP controls (Figure 2) (Manuscript Appendix Fig S2D), suggesting that indeed IRGM mediates degradation of sensor proteins.

#3 The figure 3E is **stably** expressing IRGM in **HT-29 colon epithelial cells** whereas 3F is **transient overexpression** (only 4 hours) of IRGM in **THP1 monocytic cells** (This difference is mentioned in figure, figure legend and manuscript text). The first thing is that they are entirely different cell lines; second, they are entirely different conditions (stable vs very transient). Also, in stable HT29 cell line, its single copy in the genome and in THP1 it is transfected in multiple copies. I am sure that the reviewer understands that with such a sizeable biological difference, we cannot expect the exact same results. Nevertheless, the results clearly show that in two different cell lines and in two different conditions, the IRGM degrades the PRR's. We would like to point out that the expression of Flag-IRGM in HT-29 colon cells is not that low (Blot on right side for three stable clones). It is appearing low because actin bands are developed along with IRGM expression, so it is lesser exposure for IRGM protein. We are adding one more blot of the same stable cell line in the figure.

5. #1 IRGM is an autophagy-related gene as the author indicated and lots of previous showed IRGM could affect the autophagy flux. However, in Figure 4A, 4B and 4C, the author showed that overexpression of IRGM failed to enhance the switch of LC3I/II without autophagic inhibitors.

#2 These results were inconsistent with the previous study entitled "Human IRGM Induces Autophagy to Eliminate Intracellular Mycobacteria" of Science in 2006. In Figure 2A and 2B in that paper, they showed overexpression of Irgm1 could significantly promote the switch of LC3I/II. The author should explain these inconsistencies.

#3 Moreover, as IRGM could enhance strongly the autophagy flux, IRGM may indirectly promote the autophagic degradation of RIG-I and TLR3 by manipulating autophagy flux. In that case, using autophagy inhibitors could also restrict the IRGM-mediated reduced protein levels of RIG-I and TLR3 as well as the reduced induction of IFN signaling. The authors should provide more evidence to prove IRGM could directly mediate the autophagic degradation of RIG-I and TLR-3.

#1 The reviewer is correct that in basal conditions, overexpression of human IRGM does not show an increase in LC3-II. An increase in autophagy flux (by LC3-II westerns) by IRGM overexpression can only be appreciated when bafilomycin or chloroquine is added in these conditions. So autophagy is increased, but it is not apparent (by LC3-II westerns) until flux inhibitor is added, which is a standard method to score autophagy modulation.

#2 In the concern, the reviewer has mentioned about the Figure 2A (lane 1 vs lane 2) of Singh et al., Science, 2006. Unfortunately, the mentioned blot does not have its ACTIN control, and the increase is very marginal, if it is. Therefore, we cannot comment on it. We would like to point out that there are several other papers where IRGM overexpression does not increase the LC3-II. Please see figure 5B (input panels) in (Kumar et al, 2018). Please see figure 3a (lane 1 vs lane 2) in (Brest et al, 2011).

Besides that, we think that IRGM is mainly required for selective autophagy than bulk autophagy. This is very much true for several other selective autophagy protein that their overexpression does not induce extensive bulk autophagy and conversion of LC3-I to LC3-II. I hope the above points explain that there is no inconsistency but its biological property of this protein.

#3 IRGM plays a significant role in selective autophagy than bulk autophagy (Chauhan et al, 2015; Mehto et al, 2018). It does not degrade all the autophagy targets but degrades certain PRRs selectively. If reviewer prediction might be correct than IRGM overexpression should have degraded AIM2 and MAVS, both are established autophagy targets (He et al, 2019; Jin et al, 2017; Liu et al, 2016) but IRGM does not degrade them (Please see Supplementary Figure 4C of this manuscript and Supplementary Figure 5C of Mehto et al, 2018, Molecular Cell).

To show the direct involvement of IRGM in selective autophagy of RIG-I, TLR3, and cGAS, we revealed that IRGM interacts with these PRRs both in endogenous and exogenous conditions (Figure 3A, 3B, 3C, and 3D). Further, IRGM overexpression degrades endogenous and exogenously expressed RIG-I, TLR3, and cGAS (Figure 3E, 3F, 3G, 3H, and 3I). Furthermore, in luciferase assays, IRGM overexpression reduces PRRs and reduces the Interferon response (Figure 3L, 3M, and 3N). Moreover, we showed that degradation of PRR's is selective as depleting selective autophagy adaptor protein, p62, rescued the IRGM-mediated PRR degradation and IFN response. As explained above (in point 4), we have performed several experiments to show that IRGM mediated reduction of PRR's is not just due to transcriptional change but also due to the direct degradation of PRR's.

In answer to the query 6 below, we have provided more experiments, which further strengthen our claims. We observed that IRGM-P62-PRR's forms a ternary complex where we found that both p62 and IRGM potentiate each other's interactions with PRRs leading to the degradation (Figure 5, 6, 7). A large number of evidence from old and new data show that IRGM is directly involved in p62-mediated selective autophagy RIG-I, TLR3, and cGAS.

Moreover, now, we have checked whether IRGM directly inter-

Moreover, now, we have checked whether IRGM directly inter-

act with cGAS, RIG-I, and TLR3 using purified proteins and invitro GST-pull down assays. A direct interaction was observed between IRGM and all the three PRR's (Figure 3) (Manuscript Figure 3E). The GST-IRGM strongly interacted with cGAS but relatively weaker interaction was observed between IRGM and RIG-I or TLR3.

Taken together, this manuscript now show several experiments to conclude that IRGM directly interact with PRR's to mediate their p62-dependent autophagic degradation to constrain IFN response.

6. #1 In Figure 6, the author showed overexpression or knockdown of Beclin-1 or p62 could rescue the reduction of IFN signaling mediated by IRGM. However, these assays lacked an important control in which the authors should show only knockdown of Beclin-1 or p62 deficiency might enhance IFN signaling without overexpression of IRGM. Previous studies indicated that Atg5 deficiency could also enhance the induction of IFN signaling and reduce VSV infection. The enhancement of IFN signaling might only due to knockdown of Beclin-1 or p62, but not the association between Beclin-1 or p62 with IRGM. That might be the reason why mRNA level of MX2 in p62 knockdown group was higher than the control group in Figure 6I.

#2 In Figure 6F and 6G, the authors only showed IRGM could associate with p62 and TAX1BP1. It was insufficient to prove IRGM could mediate p62-dependent selective autophagy. The authors should provide more evidence.

#1 We performed the experiment as suggested by the reviewer. We found that knocking down of BECLIN-1 increases MX2 response by only ~1.5 folds. Whereas the IRGM reduced expression of MX2 by ~5 folds, which is completely rescued by Beclin-1 knockdown in these cells (Figure 4), suggesting that it is not just basal induction (as predicted by reviewer) but indeed a rescue of IRGM-mediated selective autophagic degradation as seen in western blots also (Manuscript Figure 4A).

#2 To answer the concern, we attempted a few more experiments to understand how IRGM-mediated p62-dependent selective autophagy of PRRs takes place. In our old manuscript, we showed that IRGM interacts strongly with p62 (but not with other receptors), and the p62 depletion rescues the IRGM-mediated degradation of PRRs and also the heightened IFN response.

Figure 4

Here, first, we asked whether p62 is required for the interaction between IRGM and PRRs. For this, we performed co-IP experiments between IRGM and RIG-I, cGAS, and TLR3 in the presence and absence of p62. The results show that the depletion of p62 considerably reduces the interaction between IRGM and PRR's (Figure 5) (Manuscript Figure 4K-M), suggesting that p62

is required for these interactions. Next, we asked what is the role of IRGM in the interaction between p62 and PRRs? We previously found that IRGM can oligomerize and can act as a scaffold protein for increasing interaction of PRRs and autophagy proteins. We tested here whether IRGM can potentiate interaction between cGAS/RIG-I/TLR3 and p62. Indeed, we found that the

interaction between p62 and PRRs is increased in the presence of IRGM in co-immunoprecipitation assays (Figure 6) (Manuscript Figure 4P-R). Taken together, the data suggest that p62 and IRGM, along with PRR's form a ternary complex (at least) where both p62 and IRGM cooperatively increase each other interaction with PRR's.

7. A previous studies entitled "Human IRGM regulates autophagy and cell-autonomous immuni-

ty functions through mitochondria" of Nature Cell Biology in 2010 showed that human IRGM was a mitochondrial-located protein that induces mitochondrial depolarization and promotes mitochondrial fission, both of which were triggers for mitophagy. The author should tell the differences between that work and the results in Figure 6 and Figure 7, and explain the necessity to repeat these results in two large figures.

We think that there is no comparison between the indicated publication and information in figure 6 and 7.

*1. We in this work are attempting to identify the trigger that induces IFN response in IRGM depleted cells in **immune cells** of humans and mice (THP-1, BMDMs). In contrast, this 2010 paper tried to understand how autophagy is induced by human IRGM (mainly overexpression in HeLa, and HEK293T). In an attempt to find the trigger, we started looking at the mitochondria, which recent finding suggests is an excellent source of DAMPs (IFN inducers).*

*2. Although the cited publication show overexpression of IRGM increases mitochondrial depolarization and mitochondrial fission, but that does not imply that the knocking down of IRGM would have reverse effect in all conditions and all cell lines especially in immune cells. Indeed, they found that IRGM knockdown in **HeLa cells** results in abnormally elongated mitochondria. In contrast, we found IRGM **knock out/down** in mouse, and human **immune cells** result in increased fused and short mitochondria. This suggests that until we perform the experiments, it could be very inaccurate to extrapolate data for our conditions. Further, there is no data of Irgm knock out mice in indicated publication.*

3. We showed that IRGM depletion results in mitophagy defect using PARKIN/PINK as a marker. NONE of their data shows that IRGM (overexpression or deletion) modulates mitophagy. We cannot extrapolate the previously published data to new results without doing experiments.

4. Furthermore, we showed in Figure 6 itself that mitochondrial respiration is defective in IRGM knock out cells. I do not see any such data in this publication.

5. In figure 7 (old manuscript), we found that mitochondrial ROS and total cellular ROS are increased in IRGM depleted immune cells (basal conditions. In contrast, in HeLa cells, they found that starvation-induced ROS is reduced in IRGM knockdown cells. Therefore, again it suggests that until we perform the experiments, we should not extrapolate data for our conditions.

In a nutshell, the information in our Figure 6 and 7 is new and is required for understanding the mitochondrial status in immune cells in our conditions. Following the reviewer's suggestions, we have now combined data from Figures 6 and 7 into a single main figure.

8. #1 The authors showed that IRGM deficiency led to severe release of mtDNA in cytosol in Figure 8A and 8B, which raises the concern whether the release of mtDNA and constitutive oxidative stress might induce cell death.

#2 However, in Figure 8C and 8E, the release of mtDNA did not show significant enhancement in the cytosol, and the decrease of mtDNA and cGAS in Figure 8D and 8F was due to the reduction of protein level of cGAS in the cytosol. The author should explain these inconsistencies.

#3 In Figure 8G, the authors electroporated DNase I enzyme in the cells to determine the role of cytosolic DNA in enhancing IFN signaling. That may not be a very assay as the transfection lev-

els as well as the enzyme activity of DNase I were hard to be detected. The authors might have better use mitophagy inhibitors, such as Mdivi 1.

#4 In Figure 8J, 8K and 8L, why did the authors only use the IRGM deficiency cells but not show the control cells?

#1 We understand the reviewer's concern. Please see Figure 6 C (top panel), 6D, 6E, 6F, and 6G of our lab publication, Mehto et al., 2019 (complete reference below). The conditions and the THP1 cells used in this manuscript and the previous study are the same. We observed that IRGM knockdown in basal conditions marginally but not significantly increases cell death.

Figure 8

#2 We are not able to understand the reviewer concern and assumptions. I think the reviewer wants to know why he is not able to see mtDNA in these figures. The reviewer is trying to visualize the mtDNA in the images, which are captured to show micronuclei. In immunofluorescence studies, for looking mtDNA (small and less intense) in cells, we need to increase exposure time or laser power while taking the images on confocal. Whereas for micronuclei, because of the big size and higher intensity, they can be captured without increasing the exposure time. Capturing both together will oversaturate the micronuclei's. In any case, the Fig. 8C (red channel, second image) and 8E (green channel, first image) do show ample amount of dsDNA. We have revisited the same images and shown them here by increasing the brightness in photoshop. The images clearly show an ample amount of mtDNA (Figure 8). Thus, there are no inconsistencies.

#3 The Mdivi1 is a mitochondrial fission and mitophagy inhibitor that may increase mtDNA in the cytosol. Whereas DNase 1 will decrease mtDNA in the cytosol. The two methods are not comparable and are **exactly opposite**.

Figure 9

Figure 10

By doing

DNase 1 experiments, we are showing the rescue of "IRGM depletion induced IFN response". Whereas Mdivi1 being mitophagy inhibitor itself (like IRGM knockdown conditions) theoretically may not rescue the responses of IRGM-depletions. Indeed, the Mdivi1 induced the IFN response in basal conditions (Figure 9) but failed to rescue the "IRGM knockdown induced IFN response" (Figure 9). Although this experiment did not serve the purpose of DNase 1, but have made our point stronger that IRGM knockdown results in mitophagy inhibition. Mdivi1 is not able to further increase IFN response in IRGM knockdown cells, suggesting that mitophagy is already inhibited in these

cells. We thank the reviewer for this query and helping us to make our conclusions stronger. Nevertheless, we understand the concern of reviewer and performed several complementary assays (recommended by other reviewers), which suggest that indeed dsDNA is increased and is essential for IFN response in IRGM knockdown cells.

1. We used the ethidium bromide method to deplete mitochondrial DNA and generate Rho0 cells in HT29 IRGM stable knockdown cells and scored the IFN response. The data clearly show that upon ethidium bromide treatment, cytosolic mtDNA is considerably reduced in IRGM knockdown cells as measured by qRT-PCR of cytochrome c oxidase subunit 1 (Figure 10) (Manuscript Fig 6H). The ISGs (MX2 and ISG15) levels were significantly rescued in IRGM knockdown cells treated with ethidium bromide. The data suggest that indeed mtDNA plays a significant role in the induction of interferon response in IRGM-depleted cells (Figure 10) (manuscript Fig 6I).

2. Further, we electroporated DNase 1 and performed qRT-PCR with cytochrome c oxidase subunit 1 to confirm whether DNase 1 is able to effectively reduce the mtDNA soiling of the cytosol of IRGM knock out BMDM's. We found almost 6 folds induction of mtDNA in IRGM knockout cells compared to the control cells. The DNase 1 treatment of IRGM KO cells considerably reduced the mtDNA content in the cytosol (Figure 12) (Manuscript Figure 6J), suggesting that indeed DNase 1 electroporation reduces the cytosolic mtDNA.

#4 As we have space constraints, and also there is no additional information in the controls, so we have not shown them previously. Now, we have included it in Appendix supplementary figures 4C and D.

9. Figure 9 only showed that IRGM deficiency could promote IFN signaling at RIG-I and cGAS levels and failed to broaden the understanding of IRGM. The authors may consider putting these results in the supplementary section.

This is the figure, which tells us that RIG-I/MDA5-MAVS, cGAS-STING-IRF3, and JAK/STAT signaling pathways are essential for IRGM mediated IFN response in three different cell lines, including primary cells. We have used 9 different siRNA's (including double and triple knockdowns) to delineate the signaling involved in heightened IFN response in IRGM depleted cells. We feel this is

one of the most crucial figures pertaining to the theme of the manuscript, and its moving to supplementary is not justified. However, as per the suggestion of the reviewer, now, half of the results are moved to Expanded view figure 5.

References

Bell G (2016) Replicates and repeats. *BMC Biol* **14**: 28

Brest P, Lapaquette P, Souidi M, Lebrigand K, Cesaro A, Vouret-Craviari V, Mari B, Barbry P, Mosnier JF, Hebuterne X, Harel-Bellan A, Mograbi B, Darfeuille-Michaud A, Hofman P (2011) A synonymous variant in IRGM alters a binding site for miR-196 and causes deregulation of IRGM-dependent xenophagy in Crohn's disease. *Nat Genet* **43**: 242-245

Butler TAJ, Paul JW, Chan EC, Smith R, Tolosa JM (2019) Misleading Westerns: Common Quantification Mistakes in Western Blot Densitometry and Proposed Corrective Measures. *Biomed Res Int* **2019**: 5214821

Chauhan S, Mandell MA, Deretic V (2015) IRGM governs the core autophagy machinery to conduct antimicrobial defense. *Mol Cell* **58**: 507-521

Chen M, Meng Q, Qin Y, Liang P, Tan P, He L, Zhou Y, Chen Y, Huang J, Wang RF, Cui J (2016) TRIM14 Inhibits cGAS Degradation Mediated by Selective Autophagy Receptor p62 to Promote Innate Immune Responses. *Mol Cell* **64**: 105-119

Du Y, Duan T, Feng Y, Liu Q, Lin M, Cui J, Wang RF (2018) LRRC25 inhibits type I IFN signaling by targeting ISG15-associated RIG-I for autophagic degradation. *EMBO J* **37**: 351-366

Hansen MD, Johnsen IB, Stiberg KA, Sherstova T, Wakita T, Richard GM, Kandasamy RK, Meurs EF, Anthonen MW (2017) Hepatitis C virus triggers Golgi fragmentation and autophagy through the immunity-related GTPase M. *Proc Natl Acad Sci U S A* **114**: E3462-E3471

He X, Zhu Y, Zhang Y, Geng Y, Gong J, Geng J, Zhang P, Zhang X, Liu N, Peng Y, Wang C, Wang Y, Liu X, Wan L, Gong F, Wei C, Zhong H (2019) RNF34 functions in immunity and selective mitophagy by targeting MAVS for autophagic degradation. *EMBO J* **38**: e100978

Jin S, Tian S, Luo M, Xie W, Liu T, Duan T, Wu Y, Cui J (2017) Tetherin Suppresses Type I Interferon Signaling by Targeting MAVS for NDP52-Mediated Selective Autophagic Degradation in Human Cells. *Mol Cell* **68**: 308-322 e304

Kumar S, Jain A, Farzam F, Jia J, Gu Y, Choi SW, Mudd MH, Claude-Taupin A, Wester MJ, Lidke KA, Rusten TE, Deretic V (2018) Mechanism of Stx17 recruitment to autophagosomes via IRGM and mammalian Atg8 proteins. *J Cell Biol* **217**: 997-1013

Lin YC, Chang PF, Lin HF, Liu K, Chang MH, Ni YH (2016) Variants in the autophagy-related gene IRGM confer susceptibility to non-alcoholic fatty liver disease by modulating lipophagy. *J Hepatol* **65**: 1209-1216

Liu T, Tang Q, Liu K, Xie W, Liu X, Wang H, Wang RF, Cui J (2016) TRIM11 Suppresses AIM2 Inflammasome by Degrading AIM2 via p62-Dependent Selective Autophagy. *Cell Rep* **16**: 1988-2002

Luo X, Liu R, Zhang Z, Chen Z, He J, Liu Y (2019) Mitochondrial Division Inhibitor 1 Attenuates Mitophagy in a Rat Model of Acute Lung Injury. *Biomed Res Int* **2019**: 2193706

Mehto S, Jena KK, Nath P, Chauhan S, Kolapalli SP, Das SK, Sahoo PK, Jain A, Taylor GA, Chauhan S (2018) The Crohn's Disease Risk Factor IRGM Limits NLRP3 Inflammasome Activation by Impeding Its Assembly and by Mediating Its Selective Autophagy. *Mol Cell*

Pillai-Kastoori L, Schutz-Geschwender AR, Harford JA (2020) A systematic approach to quantitative Western blot analysis. *Anal Biochem* **593**: 113608

Prabakaran T, Bodda C, Krapp C, Zhang BC, Christensen MH, Sun C, Reinert L, Cai Y, Jensen SB, Skouboe MK, Nyengaard JR, Thompson CB, Lebbink RJ, Sen GC, van Loo G, Nielsen R, Komatsu M, Nejsum LN, Jakobsen MR, Gyrd-Hansen M, Paludan SR (2018) Attenuation of cGAS-STING signaling is mediated by a p62/SQSTM1-dependent autophagy pathway activated by TBK1. *EMBO J* **37**

Singh SB, Davis AS, Taylor GA, Deretic V (2006) Human IRGM induces autophagy to eliminate intracellular mycobacteria. *Science* **313**: 1438-1441

Vaux DL, Fidler F, Cumming G (2012) Replicates and repeats--what is the difference and is it significant? A brief discussion of statistics and experimental design. *EMBO Rep* **13**: 291-296

Vo MT, Smith BJ, Nicholas J, Choi YB (2019) Activation of NIX-mediated mitophagy by an interferon regulatory factor homologue of human herpesvirus. *Nat Commun* **10**: 3203

Xian H, Yang S, Jin S, Zhang Y, Cui J (2020) LRRC59 modulates type I interferon signaling by restraining the SQSTM1/p62-mediated autophagic degradation of pattern recognition receptor DDX58/RIG-I. *Autophagy* **16**: 408-418

Yao N, Wang C, Hu N, Li Y, Liu M, Lei Y, Chen M, Chen L, Chen C, Lan P, Chen W, Chen Z, Fu D, Ye W, Zhang D (2019) Inhibition of PINK1/Parkin-dependent mitophagy sensitizes multidrug-resistant cancer cells to B5G1, a new betulinic acid analog. *Cell Death Dis* **10**: 232

Dear Santosh,

Thank you for the submission of your revised manuscript to our editorial offices. We have now received the reports from the three referees that were asked to re-evaluate your study, you will find below. As you will see, the referees now fully support the publication of your study in EMBO reports.

Before we can proceed with formal acceptance, I have these editorial requests we ask you to address in a final revised manuscript:

- Please order the manuscript sections like this:

Title page - Abstract - Introduction - Results - Discussion - Materials and Methods - DAS - Acknowledgements - Author contributions - Conflict of interest - References - Figure legends - Expanded View Figure legends

- Please add up to 5 key words to the title page.

- Please adjust the final manuscript text to our new reference format:

- Please fill in properly the author checklist (sections B, D and E). There are animal experiments in the paper, and also experiments with samples from human donors.

- Regarding data quantification and statistics, can you please check again that where applicable the number "n" for how many independent experiments (biological or technical replicates - please clearly indicate the nature of the replicate) were performed is, the bars and error bars (e.g. SEM, SD) and the test used to calculate p-values is indicated in the respective figure legends. Please see also the attached file from our publisher (see below).

- Presently, many scale bars in the microscopic images are too small/thin. Please add similar looking scale bars to all these images, using clearly visible black or white bars (depending on the background). Please do not write on or near the bars in the image, but define the size in the respective figure legend.

- Thank you for providing the source data, in particular of the Western blots. However, we need the source data uploaded as one PDF file per figure (for main and EV figures). Please also provide the WB source data for the Appendix figures, but here in one PDF file.

- Many Western blots in the figures are overcontrasted. Please provide these as unmodified as possible and with similar contrast/intensities, matching the source data.

- In figure 5G, please label the diagram 'mouse#1' and 'mouse#2'.

- There are tables uploaded as Tables S1 and S2. These should be renamed Dataset EV1 and Dataset EV2. Please check that the callouts are adjusted accordingly. Both dataset files need to have a legend added to the first TAB in the respective excel files.

- Please add page numbers to the Appendix file, and also add these to the ToC.

- Please correct the nomenclature of the figures in the Appendix. This should be Appendix Figure S1, Appendix Figure S2 and so on. Please check that these figures are correctly called out using this nomenclature. Please also add each legend below the respective figure in the Appendix. This is more comprehensible for the readers.

- The V1 version of the manuscript had a primer table in the methods section (qRT-PCR primers). I can't see that in the V2. As this is important information, please add this to the Appendix as Appendix Table S1. Please use this nomenclature and use this call-out in the manuscript text.

- Please make sure that the funding information added in the online submission system is complete and similar to the one mentioned in the manuscript.

- Finally, please find attached a word file of the manuscript text (provided by our publisher) with changes we ask you to include in your final manuscript text, and some queries, we ask you to address. Please provide your final manuscript file with track changes, in order that we can see any modifications done.

In addition I would need from you:

- a short, two-sentence summary of the manuscript
- two to three bullet points highlighting the key findings of your study

Kind regards,

Achim

Achim Breiling
Editor
EMBO Reports

Referee #1:

The authors addressed most of my concerns and overall, the manuscript is more complete in the current version.

Referee #2:

The authors have addressed my concerns adequately and the manuscript is suitable for publication.

Referee #3:

The authors provided more data to address all the questions.

June 22, 2020

To

Editor

EMBO Reports

Dear Dr. Achim Breiling

We are submitting a revised version of our manuscript (EMBOR-2020-50051V2) entitled **“Autoimmunity Gene IRGM Suppresses cGAS-STING and RIG-I-MAVS Signaling to control Interferon Response”** for consideration in publication in *EMBO Reports*.

We have corrected and made changes in manuscript and figures according to editorial requests:

Q1. Please order the manuscript sections like this:

Title page - Abstract - Introduction - Results - Discussion - Materials and Methods - DAS - Acknowledgements - Author contributions - Conflict of interest - References - Figure legends - Expanded View Figure legends

Corrected

Q2. - Please add up to 5 key words to the title page.

Added

Q3- Please adjust the final manuscript text to our new reference format:

Updated the format

Q4- Please fill in properly the author checklist (sections B, D and E). There are animal experiments in the paper, and also experiments with samples from human donors.

Updated the Author Checklist

Q5- Regarding data quantification and statistics, can you please check again that where applicable the number "n" for how many independent experiments (biological or technical replicates - please clearly indicate the nature of the replicate) were performed is, the bars and error bars (e.g. SEM, SD) and the test used to calculate p-values is indicated in the respective figure legends. Please see also the attached file from our publisher (see below).

Corrected

Q6- Presently, many scale bars in the microscopic images are too small/thin. Please add similar looking scale bars to all these images, using clearly visible black or white bars (depending on

the background). Please do not write on or near the bars in the image, but define the size in the respective figure legend.

Corrected

Q7- Thank you for providing the source data, in particular of the Western blots. However, we need the source data uploaded as one PDF file per figure (for main and EV figures). Please also provide the WB source data for the Appendix figures, but here in one PDF file.

Corrected

Q8- Many Western blots in the figures are overcontrasted. Please provide these as unmodified as possible and with similar contrast/intensities, matching the source data.

Corrected, These blots which are corrected for contrast 2B, 2D, 2I , 2J, 2M, 2N, 3A, 3O, 4A 4B, 4M, 4N, 5F, 5G, 5H, 5I, EV3G, E4E, and EV4L

Q9- In figure 5G, please label the diagram 'mouse#1' and 'mouse#2'.

Corrected

Q10- There are tables uploaded as Tables S1 and S2. These should be renamed Dataset EV1 and Dataset EV2. Please check that the callouts are adjusted accordingly. Both dataset files need to have a legend added to the first TAB in the respective excel files.

Corrected

Q11- Please add page numbers to the Appendix file, and also add these to the ToC.

Corrected

Q12- Please correct the nomenclature of the figures in the Appendix. This should be Appendix Figure S1, Appendix Figure S2 and so on. Please check that these figures are correctly called out using this nomenclature. Please also add each legend below the respective figure in the Appendix. This is more comprehensible for the readers.

Corrected

Q13- The V1 version of the manuscript had a primer table in the methods section (qRT-PCR primers). I can't see that in the V2. As this is important information, please add this to the Appendix as Appendix Table S1. Please use this nomenclature and use this call-out in the manuscript text.

Corrected

Q14- Please make sure that the funding information added in the online submission system is complete and similar to the one mentioned in the manuscript.

Corrected

Q15- Finally, please find attached a word file of the manuscript text (provided by our publisher)

with changes we ask you to include in your final manuscript text, and some queries, we ask you to address. Please provide your final manuscript file with track changes, in order that we can see any modifications done.

Addressed

Q16. In addition I would need from you:

- a short, two-sentence summary of the manuscript
- two to three bullet points highlighting the key findings of your study

Summary:

This study shows that IRGM/Irgm1 is a master negative regulator of the interferon response by controlling the mitophagy and the selective autophagy of nucleic acid sensors.

Highlights:

- **IRGM/Irgm1 is a master switch that suppresses the interferon responses under steady-state conditions, and its deficiency results in robust and systemic induction of type 1 IFN response.**
- **IRGM suppresses interferon signaling by mediating p62-dependent autophagic degradation of cGAS, RIG-I, and TLR3.**
- **IRGM/Irgm1 deficiency results in defective mitophagy and enhanced mitochondrial DAMPs that stimulate cGAS-STING and RIG-I-MAVS axis to drive the interferon response.**

I am very thankful to you for the consideration!

Sincerely

Dr. Santosh Chauhan
Senior Scientist
Wellcome-DBT and EMBO GIN Fellow
Institute of Life Sciences
Bhubaneswar 751023, Odisha, India.
E.mail: schauhan@ils.res.in; Web: <https://www.autophagylab.com/>

Dear Santosh,

Thank you for the submission of your revised manuscript to our editorial offices. I have a couple of further editorial requests that need to be addressed before we can proceed with formal acceptance:

- The scale bars are still not in a publishable state. In the Appendix some microscopic images still have no scale bars, and all the magnification boxes throughout the paper are missing scale bars. Please add these. Please take care to use a uniform style for the scale bars, using clearly visible black or white bars (depending on the background), without any writing on or near the bars in the image (some scale bars still have that in the present manuscript). Please define their size only in the respective figure legends.
- Please check that where applicable the number "n" for how many independent experiments (biological or technical replicates - please clearly indicate the nature of the replicate) were performed, the bars and error bars (e.g. SEM, SD) and the test used to calculate p-values are indicated in the figure legends of Appendix. This is partly missing in the Appendix, in particular the indication for biological or technical replicates.
- For some of the Western blots you indicate n=2 or n=3, which does not make much sense. Fig. 5F e.g. states '2 replicates shown, n=3'. Do you mean that the experiments were done 2 or three times, and one representative experiment is shown? Please clarify, and change the legends accordingly.
- For Figs 5C/D and EV4CD the labels/numbers on the axes of the diagrams (line profile) shown are not legible. Please find a way to show these in a way that the number can be read.
- Please go through the figures and label panels where data from one mouse is shown 'mouse#x', not 'mice#x', like in Fig. 5G. See e.g. Fig. 2D or 2N.
- You added colour scale bars to the heat maps in Fig. EV1, however there are no values. Please put numbers to indicate the z-scores (i.e. in Figs. EV1D, H, I, J).

Kind regards,

Achim

Achim Breiling
Editor
EMBO Reports

June 27, 2020

To

Editor

EMBO Reports

Dear Dr. Achim Breiling

We are submitting a revised version of our manuscript (EMBOR-2020-50051V2) entitled "***Autoimmunity Gene IRGM Suppresses cGAS-STING and RIG-I-MAVS Signaling to control Interferon Response***" for consideration in publication in *EMBO Reports*.

Thanks a lot for the concerns. We are uploading the manuscript and figures with the changes. Please see below answer to your concerns/queries in blue fonts.

- The scale bars are still not in a publishable state. In the Appendix, some microscopic images still have no scale bars, and all the magnification boxes throughout the paper are missing scale bars. Please add these. Please take care to use a uniform style for the scale bars, using clearly visible black or white bars (depending on the background), without any writing on or near the bars in the image (some scale bars still have that in the present manuscript). Please define their size only in the respective figure legends.

- The scale bars are now added in all main Appendix figures.
- The magnification boxes or zoom boxes are digital magnification (not microscopic magnifications), its not appropriate to add scale bars to digital magnifications, it will be inaccurate if we add it. Now in legends of the figures, we have mentioned this- "zoom images are digital magnifications".
- Our some of the images contain embedded scale bars with size. That was the reason we didn't hide them earlier. Now, we have a uniform style throughout the figures. Now, No writing is there in figure and size of scale bars are mentioned in the figure legends.

Thanks!!

- Please check that where applicable the number "n" for how many independent experiments (biological or technical replicates - please clearly indicate the nature of the replicate) were performed, the bars and error bars (e.g. SEM, SD) and the test used to calculate p-values are indicated in the figure legends of Appendix. This is partly missing inf the Appendix, in particular the indication for biological or technical replicates.

We have now added the details as per the figures. Thanks!!

- For some of the Western blots you indicate $n=2$ or $n=3$, which does not make much sense. Fig. 5F e.g. states '2 replicates shown, $n=3$ '. Do you mean that the experiments were done 2 or three times, and one representative experiment is shown? Please clarify, and change the legends accordingly.

We have now corrected it. Please note that for mice, we are indicating the number of mice (eg. $n=2$ mice or $n=3$ mice). For cell culture, we are indicating whether western blot shown in the figure are from 2 biological replicates or 3 biological replicates only if all the replicates are shown in figures itself (eg, 2 biological replicates shown or 3 biological replicates shown). We have removed any confusing statements now.

- For Figs 5C/D and EV4CD the labels/numbers on the axes of the diagrams (line profile) shown are not legible. Please find a way to show these in a way that the number can be read.

We have improved on the figure axes. Actually, the line profile is indicating the degree of colocalization. The numbers on axes are not at all important for the conclusion of the results. There are so many numbers in axes, if we add all in big fonts, nothing will be visible. So we added the numbers in the beginning, middle, and end in bigger fonts. I hope this is fine.

- Please go through the figures and label panels where data from one mouse is shown 'mouse#x', not 'mice#x', like in Fig. 5G. See e.g. Fig. 2D or 2N.

We have corrected this in Figure 2. Thanks!!

- You added colour scale bars to the heat maps in Fig. EV1, however, there are no values. Please put numbers to indicate the z-scores (i.e. in Figs. EV1D, H, I, J).

We have corrected this. Thanks!!

Summary:

This study shows that IRGM/Irgm1 is a master negative regulator of the interferon response by controlling the mitophagy and the selective autophagy of nucleic acid sensors.

Highlights:

- **IRGM/Irgm1 is a master switch that suppresses the interferon responses under steady-state conditions, and its deficiency results in robust and systemic induction of type 1 IFN response.**

- **IRGM suppresses interferon signaling by mediating p62-dependent autophagic degradation of cGAS, RIG-I, and TLR3.**
- **IRGM/Irgm1 deficiency results in defective mitophagy and enhanced mitochondrial DAMPs that stimulate cGAS-STING and RIG-I-MAVS axis to drive the interferon response.**

I am very thankful to you for the consideration!

Sincerely

Dr. Santosh Chauhan
Senior Scientist
Wellcome-DBT and EMBO GIN Fellow
Institute of Life Sciences
Bhubaneswar 751023, Odisha, India.
E.mail: schauhan@ils.res.in; Web: <https://www.autophagylab.com/>

Dr. Santosh Chauhan
Institute of Life Sciences
Cell Biology
Nalco square rd
Chandrasekharpur
Bhubaneswar, Odisha 751023
India

Dear Dr. Chauhan,

I am very pleased to accept your manuscript for publication in the next available issue of EMBO reports. Thank you for your contribution to our journal.

At the end of this email I include important information about how to proceed. Please ensure that you take the time to read the information and complete and return the necessary forms to allow us to publish your manuscript as quickly as possible.

As part of the EMBO publication's Transparent Editorial Process, EMBO reports publishes online a Review Process File to accompany accepted manuscripts. As you are aware, this File will be published in conjunction with your paper and will include the referee reports, your point-by-point response and all pertinent correspondence relating to the manuscript.

If you do NOT want this File to be published, please inform the editorial office within 2 days, if you have not done so already, otherwise the File will be published by default [contact: emboreports@embo.org]. If you do opt out, the Review Process File link will point to the following statement: "No Review Process File is available with this article, as the authors have chosen not to make the review process public in this case."

Should you be planning a Press Release on your article, please get in contact with emboreports@wiley.com as early as possible, in order to coordinate publication and release dates.

Thank you again for your contribution to EMBO reports and congratulations on a successful publication. Please consider us again in the future for your most exciting work.

Yours sincerely,

Achim Breiling
Editor
EMBO Reports

THINGS TO DO NOW:

You will receive proofs by e-mail approximately 2-3 weeks after all relevant files have been sent to our Production Office; you should return your corrections within 2 days of receiving the proofs.

Please inform us if there is likely to be any difficulty in reaching you at the above address at that time. Failure to meet our deadlines may result in a delay of publication, or publication without your corrections.

All further communications concerning your paper should quote reference number EMBOR-2020-50051V4 and be addressed to emboreports@wiley.com.

Should you be planning a Press Release on your article, please get in contact with emboreports@wiley.com as early as possible, in order to coordinate publication and release dates.

Corresponding Author Name: Dr. Santosh Chauhan

Journal Submitted to: EMBO REPORTS

Manuscript Number: EMBOR-2020-50051V3